# HUMANIZING THE MACHINE: PROXY ATTACKS TO MISLEAD LLM DETECTORS

**Tianchun Wang**[1]*, **Yuanzhou Chen**[2]*, **Zichuan Liu**[3], **Zhanwen Chen**[4], **Haifeng Chen**[5], **Xiang Zhang**[1], **Wei Cheng**[5]†

[1]The Pennsylvania State University, [2]University of California, Los Angeles,
[3]Nanjing University, [4]University of Virginia, [5]NEC Laboratories America
`{tkw5356, xzz89}@psu.edu, adrianchen@cs.ucla.edu,`
`zichuanliu@smail.nju.edu.cn, pct4et@virginia.edu,`
`{haifeng, weicheng}@nec-labs.com`

## ABSTRACT

The advent of large language models (LLMs) has revolutionized the field of text generation, producing outputs that closely mimic human-like writing. Although academic and industrial institutions have developed detectors to prevent the malicious usage of LLM-generated texts, other research has doubt about the robustness of these systems. To stress test these detectors, we introduce a **hum**anized **p**roxy-**a**ttack (HUMPA) strategy that effortlessly compromises LLMs, causing them to produce outputs that align with human-written text and mislead detection systems. Our method attacks the source model by leveraging a reinforcement learning (RL) fine-tuned humanized small language model (SLM) in the decoding phase. Through an in-depth analysis, we demonstrate that our attack strategy is capable of generating responses that are indistinguishable to detectors, preventing them from differentiating between machine-generated and human-written text. We conduct systematic evaluations on extensive datasets using proxy-attacked open-source models, including Llama2-13B, Llama3-70B, and Mixtral-8×7B in both white- and black-box settings. Our findings show that the proxy-attack strategy effectively deceives the leading detectors, resulting in an average AUROC drop of 70.4% across multiple datasets, with a maximum drop of 95.0% on a single dataset. Furthermore, in cross-discipline scenarios, our strategy also bypasses these detectors, leading to a significant relative decrease of up to 90.9%, while in cross-language scenario, the drop reaches 91.3%. Despite our proxy-attack strategy successfully bypassing the detectors with such significant relative drops, we find that the generation quality of the attacked models remains preserved, even within a modest utility budget, when compared to the text produced by the original, unattacked source model.

**WARNING: This paper contains AI-generated text that is offensive in nature.**

## 1 INTRODUCTION

Large language models (LLMs) such as ChatGPT (OpenAI, 2023), Llama (Touvron et al., 2023a;b; Meta, 2024) and Mixtral (Jiang et al., 2024), have significantly influenced both the industrial and academic landscapes, with vast applications in news reporting, story writing, and academic research. However, there are growing concerns surrounding the misuse of these models, including the fabrication of fake news (Sun et al., 2024), the emergency of malicious content on website (Radivojevic et al., 2024), and the arise of plagiarism (Khalil & Er, 2023). Concerns regarding misinformation, plagiarism and copyright (Gao et al., 2022; Else, 2023) have prompted some scientific institutions to take a stance on the use of AI-generated content in research papers. In response to these challenges, there is an increasing emphasis on developing robust and reliable detection methods (Sadasivan et al., 2023; Lu et al., 2023; Valiaiev, 2024) for machine-generated texts.

---

*Authors contributed equally.
†Corresponding author.

The methods for detecting AI-generated text ranging from watermarking (Zhao et al., 2023; Kirchen-bauer et al., 2023a; Singh & Zou, 2023), training-based methods for binary classifiers (Chen et al., 2023b; Guo et al., 2023; Yu et al., 2023; Li et al., 2023) to zero-shot methods (Bakhtin et al., 2019; Solaiman et al., 2019; Uchendu et al., 2020; Bao et al., 2023; Mitchell et al., 2023a; Yang et al., 2023a). While these detectors may provide temporary reassurance, their reliability and robustness for detecting machine-generated text remain uncertain. Most recent studies have reported detectors are vulnerable when facing attacks (Sadasivan et al., 2023; Krishna et al., 2024; Zhou et al., 2024; Lu et al., 2023; Jovanović et al., 2024; Nicks et al., 2024; Creo & Pudasaini, 2024). The recent research (Nicks et al., 2024) has revealed that detectors are vulnerable when they are targeted for optimization, meaning language models can be fine-tuned through reinforcement learning to make the texts generated by the fine-tuned model evade detection. However, this paradigm is feasible only when the language model is relatively small and weak (e.g., 7B). For larger and stronger models (e.g., 70B), the fine-tuning process becomes significantly more costly. As shown in the paper (Nicks et al., 2024), the generated-text quality by the small language model decrease further after fine-tuning, so the fine-tuning parameters (such as $\beta$ in DPO (Rafailov et al., 2024)) must be carefully set to balance the evasion performance and the generation quality during fine-tuning. Most importantly, it is typically impossible for a hacker to access, fine-tune, and re-deploy the source model within a detection system.

Therefore, there is an urgent need to develop more practical solutions to evade text detection. In our study, we seek to address the following pivotal inquiries: 1) Can we devise feasible, cost-effective strategy to attack the LLM aligning to the distribution of human-written text? 2) Does our attack strategy bypass the leading detectors while preserving the text generation quality? 3) Does our attack strategy successfully deceive the detectors to the same extent as a direct attack on the source model? To answer these questions, in this work, we introduce an innovative yet straightforward attack paradigm that aligns the distribution of the source model with that of human-written text by employing a RL fine-tuned humanized SLM, dubbed HUMPA (**hum**anized **p**roxy **a**ttack). Our approach aims to implement a lightweight attack on the source model by leveraging the distribution shift of a hu-

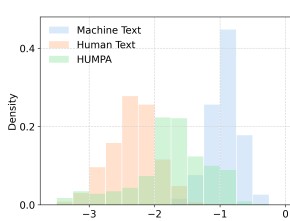

Figure 1: The probability distribution of the source model (Llama3-70B) and the HUMPA-attacked source model using humanized Llama3-8B, evaluated on passages from the OpenWebText dataset. After the attack, the distribution aligns more closely with that of human-written text.

manized SLM, adapting it to the target source model without requiring fine-tuning of the source model itself (as illustrated in Figure 1 and 2). We provide an in-depth analysis, demonstrating that the HUMPA attack strategy effectively circumvents detectors by making it more challenging to distinguish between machine-generated text and human-written content. We theoretically analyze fine-tuning language models with DPO based on the preference data constructed with the detectors, and demonstrate that attacking with a proxy humanized SLM is comparable to directly attacking the source LLM. Through extensive experiments using proxy-attacked open-source models, including Llama2-13B, Llama3-70B, and Mixtral-8×7B, in both white-box and black-box settings, our HUMPA-attack strategy consistently deceives top detectors, resulting in an average AUROC drop of 70.4% across multiple datasets, with a maximum drop of 95.0% on a single dataset. In cross-discipline scenarios, HUMPA exhibits a significant relative decrease in detection accuracy of up to 90.9%, while in cross-language scenarios, the reduction reaches 91.3%. Notably, the generation quality of the attacked models remains well-preserved, maintaining a reasonable utility budget compared to the output from the unattacked source models.

In summary, we address our contributions as follows:

- We propose an attack strategy, HUMPA, which contaminates the source model by aligning its distribution to resemble human-written text, using a fine-tuned, humanized small language model.
- By providing an in-depth analysis for the attacking process, we theoretically justify that bringing an effectively attacked small model via HUMPA is equivalent to attacking the large model.
- Our systematic evaluation across extensive datasets confirms that detectors are consistently deceived by proxy-attacked source models in both white-box and black-box settings. Additionally, detectors can be misled by a humanized SLM trained on cross-domain data sources. Despite the evasion, the quality of the texts generated by the compromised models remains preserved.

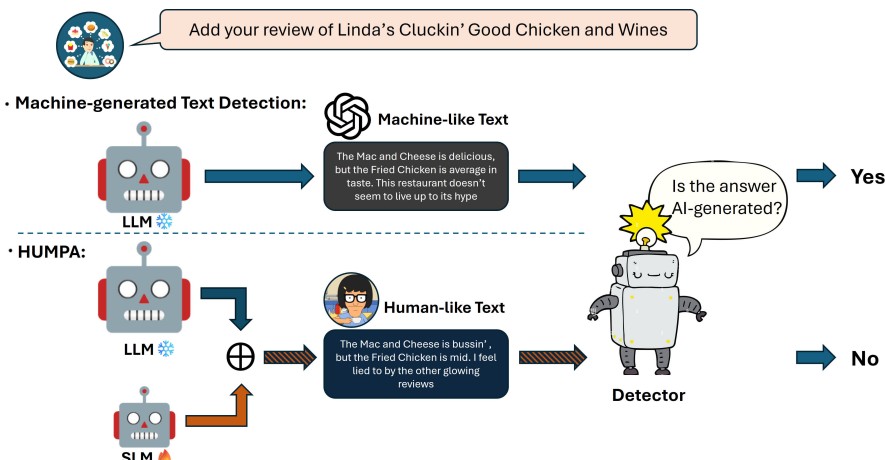

Figure 2: Overview of the humanized proxy attack. The attack overrides a large model's predictions by using a fine-tuned, smaller, humanized model during decoding. As a result, the LLM produces more human-like text that can deceive detection systems.

## 2  RELATED WORK

**Machine-generated Text Detection.** The widespread adoption of large language models (LLMs) underscores the necessity for reliable methodologies to detect the texts generated by these models. The detection is aiming to ascertain if a given text generated by a language model on the condition that the model is known (white-box) or unknown (black-box) (Wu et al., 2023). In the era of LLMs, recent efforts are focused on training a binary classifier using model-generated texts to distinguish between LLM-generated and human-written content (Verma et al., 2023; Venkatraman et al., 2023). However, these methods usually result in limited generalization capabilities when exposed to out-of-distribution data (Pu et al., 2023). Zero-shot approaches detect LLM-generated text by comparing the differences in performance metrics after statistical perturbation without training. The typical methods including log-probability curvature (DetectGPT (Mitchell et al., 2023a)) and conditional log-probability curvature (Fast-DetectGPT (Bao et al., 2023)), normalized log-rank perturbation (DetectLLM (Su et al., 2023)), N-gram divergence between multiple completions of a truncated passage (DNA-GPT (Yang et al., 2023a)), and the intrinsic dimensionality of generated text (PHD (Tulchinskii et al., 2024)). Binoculars (Hans et al., 2024) evaluates the log perplexity of the given text by leveraging an "observer" LLM, while a "performer" LLM generates next-token predictions. The perplexity of these predictions is then calculated based on the observer's assessment. These methods improve the detectors' ability to adapt to new data and source models, and become leading methods in machine-generated text detection.

**Detection Evasion Methods** As LLM-generated text detectors receive increasing attention, recent research has been conducted on methods for circumventing these detectors. The popular techniques are paraphrasing methods, including training an additional model to modify the AI-generated content (Sadasivan et al., 2023; Krishna et al., 2024), in-context optimization (Lu et al., 2023), space infiltration (Cai & Cui, 2023), homoglyph-based rewritten (Creo & Pudasaini, 2024), watermark bypassing (Jovanović et al., 2024; Wu & Chandrasekaran, 2024; Kirchenbauer et al., 2023b). However, the premise of watermarking involves embedding detectable patterns into generated text prior to its release, typically hosted behind APIs that enforce watermarking. A single strong LLM with accessible weights undermine these threats (Nicks et al., 2024). Besides, (Zhou et al., 2024) employs a surrogate model to mask and substitute words. (Shi et al., 2024) generates substitution words through a protected LLM. These paraphrasing methods modify prompts or generated content in the text-level, requiring processing of prompts or output texts with each attack launch. An alternative research line focuses on adapting the source model, enabling it to generate texts that evade detection. For example, (Nicks et al., 2024) considers a more robust threat model, where the pre-trained model is fine-tuned to maximize the 'humanness' probability of the detector with DPO (Rafailov et al., 2024). However, the effectiveness of DPO in evading detection is contingent on the assumption that the source model is relatively small (e.g., 7B). This makes it challenging to strike a balance between maintaining high-quality generated text and successfully evading detection.

Prior works (Ding et al., 2023; Xu et al., 2023) indicate the parameter efficient fine-tuning becomes more expensive when dealing with larger models (e.g., 70B). Different from previous works, our work attacks the LLM by using a humanized SLM as a proxy attacker during the decoding phase, to generate text that mimic human writing style while preserving the utility of the generated content.

## 3 METHODOLOGY

### 3.1 TASK DEFINITION.

**Text generation.** We consider a general framework for text generation processes: given the set of prompts $\mathcal{X}$ and responses $\mathcal{Y}$, an auto-regressive model generates an output sequence $y = [y_1, \cdots, y_T] \in \mathcal{Y}$ conditional on a prompt $x \in \mathcal{X}$, based on conditional probability distributions $\pi(y_t|x, y_{<t})$, where each $y_t$ is a single token. For the rest of this paper, we denote *machine* and *human* generative processes by $M$ and $H$ respectively, and use $\pi_M, \pi_H$ to denote their corresponding conditional probabilities. This random process also yields an overall distribution of output text $y$ given prompt $x$: $\mathbb{P}(y|x) = \prod_{t=1}^{T} \pi(y_t|x, y_{<t})$, which for simplicity we also denote by $\pi(y|x)$.

**Machine-Generated Text Detection.** Given a prompt-response pair $(x, y)$, a detector $D$ is essentially a binary classifier, whose task is to detect whether the response is generated from a known language model $M$ or a human process $H$. To align with existing framework, we assume an implicit reward function $r(x, y)$ that the detector bases its decisions on, which gives higher reward for human-like texts compared to machine-generated texts. We discuss this reward assumption in detail in Section 3.3.

**Detection Evasion.** In the task of detection evasion, we aim to find a new machine generative process $M'$, such that the detector is unable to distinguish texts generated by $M'$ from those by $H$. In our theoretical analyses, this is formulated as achieving an expected reward $\mathbb{E}_{y \sim \pi_{M'}(\cdot|x)} r(x, y)$ on par with the human expected reward, given that the initial expected reward for $M^{\text{ref}}$ is much smaller in comparison; in our experiments, we demonstrate the effectiveness of our model by computing the **a**rea **u**nder the **r**eceiver **o**perating **c**haracteristic curve (AUROC) and showing a decrease for $M'$ against $M^{\text{ref}}$.

### 3.2 FINE-TUNING THE LANGUAGE MODEL WITH RL

**Preference-based RL for language models.** Preference-based reinforcement learning (PBRL) leverages human or evaluative feedback to optimize a model's behavior using RL. Recent studies apply PBRL to LLMs with the hope of aligning the model to match human preferences. To fine-tune a pre-trained language model $M^{\text{ref}}$, a preference dataset $\mathcal{D} := \{(x, y^w, y^l)\}$ is required, where the responses $y^w, y^l \sim \pi^{\text{ref}}(\cdot|x)$ are sampled from a reference policy $\pi^{\text{ref}}$ that could be obtained after supervised fine-tuning (SFT), while preferences $y^w \succ y^l|x$ are labeled either by AI system or human annotator, indicating $y^w$ is preferred over $y^l$ given the query $x$. In PBRL, the preference is assumed to be associated with a latent reward function $r^*$. To learn this reward from the dataset, the Bradley-Terry model (Bradley & Terry, 1952) is commonly used, which assumes the probability of $y^w \succ y^l|x$ satisfies the following:

$$p(y^w \succ y^l|x) := \frac{\exp\left(r^*(x, y^w)\right)}{\exp\left(r^*(x, y^w)\right) + \exp\left(r^*(x, y^l)\right)}. \tag{1}$$

It follows that the maximum-likelihood reward learning objective is

$$r^* \leftarrow \arg\max_{r \in \mathcal{R}} \mathbb{E}_{(x, y^w, y^l) \sim \mathcal{D}} \left[ \log \sigma(r(x, y^w) - r(x, y^l)) \right],$$

where $\sigma$ is the sigmoid function. After obtaining the reward $r^*$, the RL fine-tuning of a language model follows the objective

$$\pi^* \leftarrow \arg\max_{\pi} \mathbb{E}_{x \sim \mathcal{D}, y \sim \pi(\cdot|x)} \left[ r^*(x, y) - \beta \mathbb{D}_{\text{KL}}[\pi(y|x) \| \pi^{\text{ref}}(y|x)] \right]. \tag{2}$$

Finally, Direct Preference Optimization (DPO) in (Rafailov et al., 2024) provides a solution for $\pi^*$ in equation 2 without learning the reward function, by optimizing the objective

$$\pi^* \leftarrow \arg\max_{\pi} \mathbb{E}_{(x, y^w, y^l) \sim \mathcal{D}} \left[ \log \sigma \left( \beta \log \frac{\pi(y^w|x)}{\pi^{\text{ref}}(y^w|x)} - \beta \log \frac{\pi(y^l|x)}{\pi^{\text{ref}}(y^l|x)} \right) \right]. \tag{3}$$

### 3.3 EVADE DETECTION VIA HUMANIZED PROXY ATTACK

When using PBRL to fine-tune large language models (LLMs) for detector evasion, the main challenge is the significant computational cost due to the large size of the models (e.g., 70B parameters). Directly fine-tuning such large models for attacks is impractical, so we propose HUMPA, which leverages DPO to fine-tune a smaller language model (SLM). The main idea is to fine-tune an SLM towards optimal reward until it reaches the same level of reward for the human process according to a scoring detector, and adapt the LLM to achieve the same expected reward.

**Obtaining a humanized SLM.** DPO fine-tuning technique (Rafailov et al., 2024; Nicks et al., 2024) can be applied for bypassing detectors. For each prompt $x \in \mathcal{X}$ in the dataset, sample response pairs $(y_1, y_2)$ are generated by the reference model $\pi^{\text{ref}}$. To obtain the dataset $\mathcal{D} = \{(x, y^w, y^l)\}$, preference labels are assigned by comparing a scoring detector's human-ness score $s(x, y)$ on the responses: if $s(x, y_1) > s(x, y_2)$, assign preference label $y_1 \succ y_2$ and let $y^w = y_1$, $y^l = y_2$; otherwise assign $y^w = y_2$, $y^l = y_1$. The generated dataset $\mathcal{D}$ is then used to fine-tune a pre-trained SLM $M_s^{\text{ref}}$ with DPO in equation 3. As a result, we have a humanized SLM denoted as $M_s$ as the proxy attacker.

We notice that this label assignment process can be approximated by the Bradley-Terry model in equation 1 when $r(x, y) = C \cdot s(x, y)$ with a large constant $C$ (see Appendix B.1), therefore we will assume the detector follows an implicit reward function $r$ to generate $\mathcal{D}$ from now on.

Generally, for fine-tuning a language model using DPO with preference data from detectors, we have the following lemma characterizing the ability of the fine-tuned model to evade detection (details and proof refer to Appendix B.2):

**Lemma 3.1.** *Given a starting reference model $M^{\text{ref}}$ with a low reward, there exists hyperparameter $\beta$ such that the optimal model $M^*$ fine-tuned on the DPO objective in equation 3 achieves the same expected reward as $H$: $\mathbb{E}_{x \sim \mathcal{D}, y \sim \pi_{M^*}(\cdot|x)} r(x, y) = \mathbb{E}_{x \sim \mathcal{D}, y \sim \pi_H(\cdot|x)} r(x, y)$.*

Intuitively, this result is due to the effect of $\beta$ on the fine-tuned model: the smaller $\beta$ is, the closer $M^*$ approaches optimal reward, while larger $\beta$ results in higher similarity to the reference model and hence higher quality. This is in line with the RL objective in equation 2, in which the $\beta$ term controls the strength of regularization.

**Attacking the LLM using humanized SLM.** With a humanized SLM trained on the DPO objective, our proxy attack HUMPA operates on the LLM's next-token output distribution by multiplying a logit offset for each token probability. This offset is calculated as the ratio between the logits of the proxy-attacker small model $M_s$ and those of the pre-trained reference small model $M_s^{\text{ref}}$. Formally, at each time step $t$, given the tokens $y_{<t}$, the probability distribution of our proxy-attacker large model $M'$ is calculated as

$$\pi_{M'}(y_t|x, y_{<t}) = \frac{1}{Z_{x,y_{<t}}} \pi_M^{\text{ref}}(y_t|x, y_{<t}) \left( \frac{\pi_{M_s}(y_t|x, y_{<t})}{\pi_{M_s}^{\text{ref}}(y_t|x, y_{<t})} \right)^{\alpha}, \tag{4}$$

where $Z_{x,y_{<t}} = \sum_{y_t} \pi_M^{\text{ref}}(y_t|x, y_{<t}) \left( \frac{\pi_{M_s}(y_t|x, y_{<t})}{\pi_{M_s}^{\text{ref}}(y_t|x, y_{<t})} \right)^{\alpha}$ is the normalization factor and $\alpha$ is the attack ratio. The term $\left( \frac{\pi_{M_s}(y_t|x, y_{<t})}{\pi_{M_s}^{\text{ref}}(y_t|x, y_{<t})} \right)^{\alpha}$ captures the distribution shift from pre-trained to fine-tuned on the small model, and attempts to approximate the corresponding shift on the large model $\frac{\pi_{M'}(y_t|x, y_{<t})}{\pi_M^{\text{ref}}(y_t|x, y_{<t})}$. Prior works (Mitchell et al., 2023b; Liu et al., 2024a;b) suggest taking the logarithm of logits, which allows us to derive the probability distribution from the proxy-attacked model $M'$ as

$$p_{M'}(y_t|x, y_{<t}) = \text{softmax} \left[ p_M^{\text{ref}}(y_t|x, y_{<t}) + \alpha \left( p_{M_s}(y_t|x, y_{<t}) - p_{M_s}^{\text{ref}}(y_t|x, y_{<t}) \right) \right], \tag{5}$$

where $p_M^{\text{ref}}$, $p_{M_s}$ and $p_{M_s}^{\text{ref}}$ are the logarithmic logits for the pre-trained large model $M^{\text{ref}}$, the fine-tuned small model $M_s$ and the pre-trained small model $M_s^{\text{ref}}$ respectively. Our method tuning at a small scale and applying the attack to the large model through Equation 5. It is important to note that $M$ and $M_s$ only need to share the same vocabulary[1]. Compared to generically fine-tuning the large model using DPO, we have the following theorem (for the proof refer to Appendix B.3).

---

[1]If vocabularies does not match, methods like those in (Gao et al., 2024) can be used to address this issue.

**Theorem 3.2.** *Assuming the small fine-tuned model $M_s$ achieves optimum according to the DPO objective with $\beta = \beta_0$, our proposed inference model $M'$ in equation 4 is the same as an alternative large model fine-tuned on the DPO objective with $\beta = \beta_0/\alpha$.*

Theorem 3.2 reveals that the attack ratio $\alpha$ has a similar (but inverted) effect as $\beta$ on the resulting model $M'$: larger $\alpha$ leads to higher reward and better detection evasion, while smaller $\alpha$ keeps $M'$ closer to the reference model $M^{\text{ref}}$. Therefore, $\alpha$ effectively controls the trade-off between evasion performance and quality *at the decoding phase*, in contrast to $\beta_0$ which is applied *at fine-tuning*.

Finally, combining Theorem 3.2 with Lemma 3.1, we have the following claim for HUMPA.

**Corollary 3.3.** *Given parameter $\beta_0$ for fine-tuning the SLM $M_s$ on the DPO objective equation 3, there exists attack ratio $\alpha > 0$ such that the resulting proxy attacker $M'$ achieves the same expected reward as the human process $H$ according to detector $D$, thereby evading detection.*

# 4 EXPERIMENTS

## 4.1 EXPERIMENTAL SETUPS

**Datasets.** We conduct a wide variety of empirical studies to show the effectiveness of our method. We follow (Bao et al., 2023) to evaluate both the white-box and black-box detectors. We evaluate the detection evasion capability of HUMPA attacked by humanized SLMs on the same dataset, including *OpenWebText* (Gokaslan & Cohen, 2019), *WritingPrompts* (Fan et al., 2018) and *PubMedQA* (Jin et al., 2019). We randomly sample 500 examples of each dataset as human-written texts. For cross-domain evaluation, detection evasion is performed using humanized SLMs on a different dataset. The experiments are conducted on the cross-discipline corpus GPABench2 (Liu et al., 2023), a dataset containing titles and abstracts of scientific writing across Computer Science (CS), Physics (PHX), and Humanities and Social Sciences (HSS). Evasion is evaluated on PHX, where the source model is attacked by a humanized SLM from CS, and on HSS, where the source model is attacked by a humanized SLM from PHX. The cross-language evaluation is conducted on *WMT-2016* (Bojar et al., 2016), where 150 examples are sampled from the Germany set as human-written texts. The humanized SLM is fine-tuned using samples from the English domain.

**Evaluation Metrics.** Following previous works, we compute the area under the receiver operating characteristic curve (AUROC) to evaluate the performance of all detectors. We also provide the results of area under the precision-recall curve (AUPRC) in Appendix. To evaluate the quality of the generated texts (i.e., utility), we adopt the popularly used ROUGE-1, ROUGE-2, ROUGE-L (Chin-Yew, 2004) and BERTScore (Zhang et al., 2019) to evaluate the texts produced by the LLMs during the text generation phase. Detailed explanation of the metrics are in Appendix D

**Source Models.** To validate our approach for evading detection, we include the most advanced open-source LLMs: Llama2, Llama3, and Mixtral. Specifically, for the base large model, we use Llama2-13B (`Llama2-13B-Chat`), Llama3-70B (`Llama3-70B-Instruct`), and Mixtral-8×7B (`Mixtral-8×7B-Instruct-v0.1`). We use these source models as the base large models, attacked by small humanized model Llama2-7B (`Llama2-7B-Chat`), Llama3-8B (`Llama3-8B-Instruct`) and Mistral-7B (`Mistral-7B-Instruct-v0.1`), respectively.

**The Detectors.** We conduct experiments with a variety of strong open-source detectors from prior literature, including RoBERTa-base and RoBERTa-large (Solaiman et al., 2019), the language models trained for detection, the zero-shot detectors Likelihood and Log Rank, the perturbation-based zero-shot method DetectGPT (Mitchell et al., 2023a), DetectLLM (Su et al., 2023) and DNA-GPT (Yang et al., 2023a). We also involve Fast-DetectGPT (Bao et al., 2023) that uses a surrogate model respectively to compute the conditional probability curvature for the texts obtained from the sampling model, and Binoculars Hans et al. (2024) that computes the ratio of perplexity to cross-perplexity obtained from "observer" and "performer" models.

**The Settings.** We evaluate the zero-shot methods in two settings, the *white-box* (source model is known) setting and *black-box* (source model is unknown) setting (Yang et al., 2023b; Bao et al., 2023). Following (Bao et al., 2023), we set the surrogate model in each detector to be identical to the source model, whereas in the black-box setting, the surrogate model differs from the source model. We utilize `GPT-Neo-2.7B` (Black et al., 2021) as the surrogate model for all detectors (Bao

Table 1: AUROCs of detectors and generation utility scores on text generated by different models, averaging across OpenWebText, WritingPrompts and PubMedQA from detailed Table 4 and 5 in Appendix E. The model name in parentheses after HUMPA refers to the SLM. The generation utilities of texts produced by HUMPA and the source model are within the budget of $\Delta S_{Bert} \leq 0.02$ and $\Delta$ROUGE-1 $\leq 0.03$. The highest attack results (the greatest relative decrease) are **boldfaced**, while the lowest attack results are underlined for each model.

| Models→ | Llama2-13B | HUMPA (Llama2-7B) | Llama3-70B | HUMPA (Llama3-8B) | Mixtral-8x7B | HUMPA (Mistral-7B) |
|---|---|---|---|---|---|---|
| The Generation Utility | | | | | | |
| $S_{Bert}$ | 0.8278 | 0.8206 | 0.8276 | 0.8126 | 0.8342 | 0.8281 |
| ROUGE-1 | 0.2780 | 0.2517 | 0.2926 | 0.2685 | 0.2911 | 0.2663 |
| ROUGE-2 | 0.0972 | 0.0798 | 0.1019 | 0.0929 | 0.1045 | 0.0818 |
| ROUGE-L | 0.1995 | 0.1779 | 0.2078 | 0.1800 | 0.2140 | 0.1863 |
| The White-box Setting | | | | | | |
| Likelihood | 0.9995 | 0.8610 | 0.9995 | 0.9070 | 0.8932 | 0.4730 |
| LogRank | 0.9993 | 0.8424 | 0.9991 | 0.8769 | 0.9072 | 0.4598 |
| LRR | 0.8547 | 0.6634 | 0.8512 | **0.5084** | 0.9039 | **0.3656** |
| NPR | 0.9908 | 0.9148 | 0.9836 | 0.8387 | 0.8747 | 0.6629 |
| DNA-GPT | 0.9815 | **0.6656** | 0.9908 | 0.8410 | 0.8217 | 0.3388 |
| DetectGPT | 0.8915 | 0.8430 | 0.8313 | 0.7906 | 0.6353 | 0.5348 |
| Fast-DetectGPT | 0.9949 | 0.9262 | 0.9952 | 0.8864 | 0.9933 | 0.4590 |
| The Black-box Setting | | | | | | |
| Roberta-base | 0.9044 | 0.7902 | 0.8225 | 0.5600 | 0.6992 | 0.5506 |
| Roberta-large | 0.8874 | 0.7862 | 0.8130 | 0.5556 | 0.7016 | 0.6233 |
| Likelihood(Neo-2.7) | 0.9961 | 0.6971 | 0.9699 | 0.7338 | 0.8156 | 0.3542 |
| LogRank(Neo-2.7) | 0.9850 | 0.6976 | 0.9733 | 0.7203 | 0.8254 | 0.3519 |
| LRR(Neo-2.7) | 0.9843 | 0.7773 | 0.9537 | **0.5361** | 0.8210 | 0.3309 |
| NPR(Neo-2.7) | 0.8586 | 0.6881 | 0.8574 | 0.5893 | 0.7076 | 0.3639 |
| DNA-GPT(Neo-2.7) | 0.7116 | **0.4295** | 0.7567 | 0.5296 | 0.5983 | 0.1907 |
| DetectGPT(T5-3B/Neo-2.7) | 0.8282 | 0.6209 | 0.7696 | 0.5853 | 0.6760 | 0.3811 |
| Fast-DetectGPT(GPT-J/Neo-2.7) | 0.9961 | 0.7389 | 0.9885 | 0.7720 | 0.8758 | **0.2590** |
| Binoculars(Falcon-7B) | 1.0000 | 0.7896 | 0.9989 | 0.8596 | 0.9353 | 0.3053 |

et al., 2023). Apart from it, Fast-DetectGPT adpots a `GPT-J-6B` (Wang & Komatsuzaki, 2021) as the sampling model in the black-box setting. Binoculars adopts a `Falcon-7B` as the observer and `Falcon-7B-Instruct` (Almazrouei et al., 2023) as the performer model. In both settings, DetectGPT applies `T5-3B` (Raffel et al., 2020) as the sampling model to generate the perturbed texts.

**Implementing Details.** We divided the dataset into evaluation and training sets. For the OpenWebText, WritingPrompts, PubMedQA, CS, and PHX datasets, we randomly choose 10k training samples each following the setup in (Nicks et al., 2024). For the WMT-2016 dataset, all English samples were used as the training set. Then, based on the samples, we prompt the LLMs to generate corresponding texts using an 8-token prefix from human-written text as the starting point for the machine-generated content for dataset OpenWebText, WritingPrompts and WMT-2016. For the PubMedQA dataset, we use the tokenized question as a prompt for generating answers with the LLMs. For the cross-discipline scientific abstract dataset, we use the tokenized title as the prompt. We employ the temperature of 1.0 across all the experiments, which is the same setting in (Nicks et al., 2024). During the DPO fine-tuning phase, we apply Low-Rank Adaptation (LoRA) to finetuning the SLM, setting the batch size to 8, the learning rate to $2e-4$, and the optimizer to AdamW. We choose the attack ratio $\alpha$ from the set $\{0.1, 0.2, ..., 1.0\}$ to balance the generation utility and the detection evasion performance. We conduct the experiments on a server with 4 NVIDIA A100 GPUs, each one with 80GB RAM.

## 4.2 MAIN RESULTS

**White-box and black-box machine-generated detection evasion.** We study the basic question of the feasibility of whether LLMs can generate human-like texts that, within a utility budget, can deceive detectors when attacked by a small humanized language model. Table 1 lists the evaluation of our method with mainstream LLM detection methods in white- and black-box settings averaging

Table 2: AUROCs of detectors and generation utility scores on text generated by different models on HSS. The humanized SLM is fine-tuned from PHX. The generation utilities of texts produced by HUMPA and the source model are within the budget of $\Delta S_{Bert} \leq 0.02$ and $\Delta$ROUGE-1 $\leq 0.03$.

| Models→ | Llama2-13B | HUMPA (Llama2-7B) | Llama3-70B | HUMPA (Llama3-8B) | Mixtral-8x7B | HUMPA (Mistral-7B) |
|---|---|---|---|---|---|---|
| The Generation Utility | | | | | | |
| $S_{\text{Bert}}$ | 0.8291 | 0.8201 | 0.8007 | 0.7976 | 0.8105 | 0.7997 |
| ROUGE-1 | 0.2632 | 0.2385 | 0.2506 | 0.2277 | 0.2381 | 0.2104 |
| ROUGE-2 | 0.0535 | 0.0459 | 0.0513 | 0.0448 | 0.0447 | 0.0364 |
| ROUGE-L | 0.1470 | 0.1315 | 0.1383 | 0.1299 | 0.1334 | 0.1178 |
| The White-box Setting | | | | | | |
| Likelihood | 1.0000 | 0.8706 | 0.9999 | 0.8317 | 0.8145 | 0.4925 |
| LogRank | 0.9997 | 0.8575 | 0.9976 | 0.7842 | 0.8244 | 0.4670 |
| LRR | 0.8096 | 0.6536 | 0.4530 | **0.3323** | 0.7862 | 0.3387 |
| NPR | 0.9993 | 0.9414 | 0.9954 | 0.9327 | 0.9226 | 0.8313 |
| DNA-GPT | 0.9622 | **0.7278** | 0.9985 | 0.7534 | 0.6946 | **0.2822** |
| DetectGPT | 0.9338 | 0.8306 | 0.9170 | 0.8081 | 0.5982 | 0.5796 |
| Fast-DetectGPT | 0.9965 | 0.9365 | 0.9961 | 0.9739 | 0.9700 | 0.5333 |
| The Black-box Setting | | | | | | |
| Roberta-base | 0.8550 | 0.7608 | 0.7399 | 0.6119 | 0.6895 | 0.5583 |
| Roberta-large | 0.8304 | 0.7291 | 0.7298 | 0.5819 | 0.6733 | 0.5934 |
| Likelihood(Neo-2.7) | 0.9590 | 0.7121 | 0.9403 | 0.5987 | 0.6679 | 0.3149 |
| LogRank(Neo-2.7) | 0.9617 | 0.7057 | 0.9343 | 0.5553 | 0.6587 | 0.2879 |
| LRR(Neo-2.7) | 0.9384 | 0.6624 | 0.8717 | **0.3780** | 0.5816 | 0.2153 |
| NPR(Neo-2.7) | 0.9422 | 0.7511 | 0.9226 | 0.7528 | 0.8010 | 0.6218 |
| DNA-GPT(Neo-2.7) | 0.9113 | **0.5827** | 0.9672 | 0.5760 | 0.6320 | 0.2082 |
| DetectGPT(T5-3B/Neo-2.7) | 0.8160 | 0.6510 | 0.8026 | 0.6788 | 0.6993 | 0.5371 |
| Fast-DetectGPT(GPT-J/Neo-2.7) | 0.9891 | 0.7488 | 0.9839 | 0.6296 | 0.7827 | 0.5333 |
| Binoculars(Falcon-7B) | 0.9971 | 0.7869 | 0.9990 | 0.7094 | 0.8252 | **0.1852** |

across three datasets for each model. Following prior work (Nicks et al., 2024), the SLM is fine-tuned with DPO against a scoring detector. In our experiments, we use Fast-DetectGPT for scoring. From the results, we find that LLMs attacked by the humanized small model are effective against all detection methods, showing a greater AUROC relative decrease in the black-box setting compared to the white-box setting for each detector and for each model. Furthermore, we find that in the white-box setting, DetectGPT is the most robust method across the models compared to others, whereas LRR exhibits a greater decrease than the other methods when detecting texts generated by Llama3-70B, which has been attacked by the humanized small model Llama3-8B, as well as texts produced by Mixtral-8×7B, which have been attacked by Mistral-7B. Additionally, we find that DNA-GPT has more fragile than the other methods for HUMPA (Llama2-7B), with relative decrease 32.0% and 39.6% in the white-box and the black-box setting respectively. In the black-box setting for Fast-DetectGPT, HUMPA (Mistral-7B) exhibits the highest AUROC relative decrease compared to other methods and models, achieving a relative decrease of 70.4%, within a budget of $\Delta S_{Bert} \leq 0.02$ and $\Delta$Rouge-1 $\leq 0.03$ for the produced texts.

### 4.3 RESULTS IN CROSS-DOMAIN SCENARIOS

In real-world scenarios, an SLM proxy attacker may be unaware of the specific dataset that will be used to evaluate the detectors. We simulate two common real-world scenarios. The first is a classroom test, where the task is to write a scientific essay based on a given title, and the SLM proxy attacker is fine-tuned on a different academic discipline. The second scenario involves cross-language evasion, simulating an international hacker attempting to bypass detectors using a humanized SLM fine-tuned in another language.

**Cross-discipline Detection Evasion.** We assess HUMPA on cross-disciplinary datasets, randomly selecting 200 human-written texts for each dataset. Specifically, we evaluate the detection methods on HSS, as shown in Table 2, using the source models attacked by the humanized SLM which was DPO fine-tuned on the PHX dataset. As demonstrated in Table 2, HUMPA successfully evaded all detection methods across all source models. Notably, DNA-GPT shows a 24.4% relative decrease

compared to Llama2-13B and a 59.4% decrease compared to Mixtral-8×7B in the white-box setting, and a 36.1% decrease compared to Llama2-13B and a 77.6% relative decrease compared to Mixtral-8×7B in the black-box setting. We also perform an assessment on PHX using humanized SLMs fine-tuned on the CS dataset (see Table 6 in Appendix F).

**Cross-language Detection Evasion.** In this scenario, we assess the detectors on texts generated by LLMs in Germany, while the humanized SLM is fine-tuned using English. For evaluation, we sample 150 human-written texts. The results are presented in Appendix Table 7. The findings include: 1) In the white-box setting, LRR is the most fragile detector when evaluating on the texts generated by the attacked Llama2-13B and Llama3-70B, with relative decrease 74.3% and 72.6% respectively. 2) In the black-box setting, when Llama2-13B is attacked

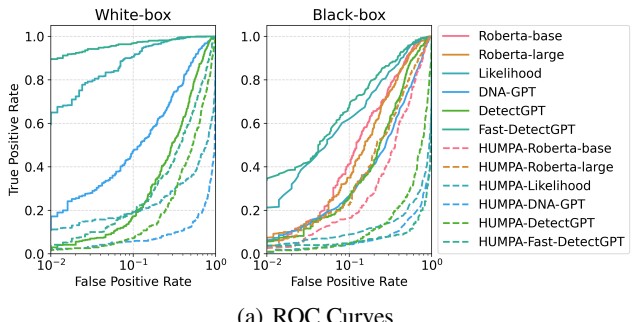

(a) ROC Curves

Figure 3: ROC curves in log scale evaluated on WritingPrompts, where the source model is Mixtral-8×7B. 'HUMPA-' denotes this detector is evaluated on the texts produced by the attacked model.

by HUMPA (Llama2-7B), the zero-shot detectors Likelihood and LogRank experience a relative performance drop of 91.3% and 90.1%, respectively. Fast-DetectGPT has relatively decrease 74.8% and 79.2% when evaluating on the texts generated by the attacked Llama3-70B and Mixtral-8×7B.

## 4.4 EXPERIMENTAL ANALYSIS

**Interpretation of AUROC.** In real-world scenarios, our focus goes beyond overall detection accuracy; we prioritize recall (the true positive rate) while striving to minimize type-I errors, aiming for a low false positive rate. As shown in Figure 3, Fast-DetectGPT exhibits a relative decrease of 95.8% in TPR at 1% FPR, while Likelihood, LogRank, and DNA-GPT all experience a relative decrease of over 80% in the white-box setting. In the black-box setting, Fast-DetectGPT and DNA-GPT show the largest and second-largest relative decreases, at 89.4% and 87.5%, respectively.

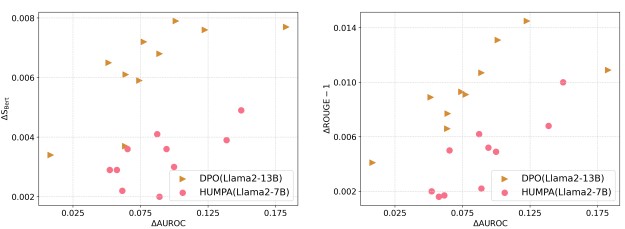

(a) $\Delta S_{Bert}$ vs. $\Delta$AUROC     (b) $\Delta$ROUGE-1 vs. $\Delta$AUROC

Figure 4: Analysis of RoBERTa-base evaluated on texts generated by DPO (Llama2-13B), where DPO refers to the direct fine-tuning of Llama2-13B, and on texts generated by HUMPA (Llama2-7B).

**Utility Preserving.** We evaluate the Roberta-base on the texts generated by directly finetune Llama2-13B against the scoring detector Roberta-large, following the approach in (Nicks et al., 2024). In comparison, Roberta-base is evaluated on the texts generated by Llama2-13B attacked using a humanized SLM Llama2-7B via our method HUMPA . The texts generated by DPO (Llama2-13B) are obtained by fine-tuning Llama2-13B for 1 to 10 epochs, while

the results from HUMPA (Llama2-7B) are produced by varying the attack ratio $\alpha$ from 0.1 to 1.0. We present the AUROC decrease ($\Delta$AUROC) with respect to the BERTScore decrease ($\Delta S_{\text{Bert}}$) for both methods in Figure 4(a), and $\Delta$AUROC with respect to the ROUGE-1 decrease ($\Delta$ROUGE-1) in Figure 4(b). The figures clearly show that both methods can deceive

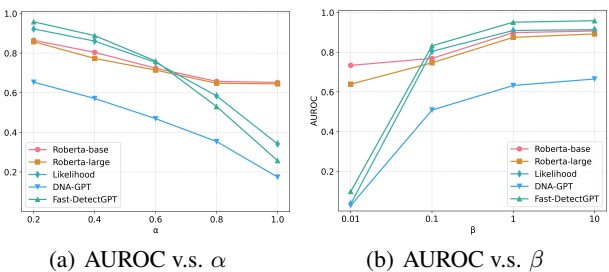

(a) AUROC v.s. $\alpha$     (b) AUROC v.s. $\beta$

Figure 5: Analysis of attack ratio $\alpha$ and $\beta$ in DPO

the detector but incur some utility loss. However, for a comparable AUROC decrease, our attack method HUMPA results in less utility loss.

**Analysis of** $\alpha$**.** We evaluate several black-box detectors on the texts generated by the attacked Llama2-13B model, using the humanized SLM Llama2-7B fine-tuned with $\beta = 0.1$ with varying attack ratio $\alpha$, on 100 samples randomly selected from WritingPrompts dataset. The trend in Figure 5(a) shows the AUROC decrease as $\alpha$ increases (more analysis refer to Appendix H).

**Analysis of** $\beta$**.** To analyze the parameter $\beta$ of the fine-tuned SLM, we measure its impact on the AUROC performance of various detectors with respect to different values of $\beta$ in DPO. The evaluation is conducted using several black-box detectors. We maintain the attack ratio of the humanized SLM at 0.5. The results, shown in Figure 5(b), suggest that detectors exhibit increased vulnerability as $\beta$ decreases.

**Efficiency.** As described in Section 3.3, fine-tuning the model with DPO requires

Table 3: Time comparison between HUMPA and directly fine-tuning the source LLM using DPO. 'Sampling' refers to the stage of sampling response pairs, 'Fine-tuning' represents DPO fine-tuning using LoRA, and 'Inference' is the process of generation.

|  | Llama2-13B | Llama3-70B | Mixtral-8x7B |
|---|---|---|---|
| Sampling (hrs) | 18.61 | 41.64 | 63.52 |
| Fine-tuning (hrs) | 3.09 | 9.54 | 13.58 |
| Inference (secs) | 12.88 | 27.87 | 32.51 |
|  | HUMPA (Llama2-7B) | HUMPA (Llama3-8B) | HUMPA (Mistral-7B) |
| Sampling (hrs) | 13.87 | 10.97 | 10.83 |
| Fine-tuning (hrs) | 2.04 | 3.20 | 1.39 |
| Inference (secs) | 39.87 | 47.22 | 36.75 |

a preference dataset, which is generated by sampling response pairs $(y_1, y_2)$ from the model. To directly fine-tune the source model, preference pairs are sampled from the source model itself. In contrast, HUMPA samples pairs from the SLM, as the SLM is the model that needs to be fine-tuned. We compare the runtime of HUMPA with prior work (Nicks et al., 2024), which samples pairs from the source model and apply DPO fine-tuning on the source model for 10k training samples. To compare the fine-tuning time, we set the DPO batch size to 8 and epoch to 5. The results are in Table 3. We find that HUMPA is much more efficient than directly DPO fine-tuning attack the source model. We also report the inference time for a single pass with a batch size of 1. The inference time of HUMPA is slightly slower than that of the source model, but the sacrifice is not significant.

## 5 CONCLUSION

The rapid evolution of potent Large Language Models (LLMs) underscores the critical necessity for robust detection methods. In this paper, we propose a plug-and-play attack strategy, HUMPA , that utilizes a small proxy model to contaminate the source models, aligning their distribution with human-like distribution. Additionally, we theoretically justify bringing an effectively attacked small model via HUMPA is equivalent to attacking the large model. Our systematical experiments validate HUMPA remains versatile across diverse text sources or cross-domain sources. In light of our results, we argue that the leading zero-shot machine generated text detectors are not robust to adversaries and may even favor machine-generated text over actual human-generated content. In conclusion, our innovations offer compelling support to the urgent demand for robust detection methods within the realm of LLM development, bridging critical gaps in developing reliable detectors.

## 6 ETHICS STATEMENT

The primary goal of this paper is not to provide a technique for evading machine-generated text detection systems, but rather to highlight the vulnerabilities present in current detection mechanisms. With the growing availability of LLMs, often released with publicly accessible tokens and freely available for use, it becomes easier for adversaries to exploit these models by employing smaller proxy models to compromise their outputs effortlessly, which would potentially circumvent detection systems. This study serves as a call to action for the broader research community to prioritize the development of more robust detection methods. We aim to raise awareness of the potential risks posed by such attacks and emphasize the need for future research focused on strengthening text detection systems. We are confident that, with increased attention and effort, the community will devise more sophisticated techniques to enhance the robustness and reliability of machine-generated text detection in the face of evolving adversarial threats.

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

# Appendix: HUMPA

## A   Code

Code is available at `https://github.com/xztcwang/evad_detection`

## B   Theoretical Analysis

We present some additional technical details and theorem proofs in the following sections.

### B.1   Detector Score and the Reward Model

In this section, we continue the discussion in Section 3.3 on the relation between the human-ness score $s(x, y)$ utilized by the detector model and an implicit reward $r(x, y)$ that the DPO framework assumes. Specifically, we demonstrate that taking $r(x, y) = C \cdot s(x, y)$ in the Bradley-Terry model yields approximately the same data generation as using the human-ness score directly for a large constant $C$. In fact,

$$
\begin{aligned}
p(y_1 \succ y_2 | x) &= \frac{\exp\big(r(x, y_1)\big)}{\exp\big(r(x, y_1)\big) + \exp\big(r(x, y_2)\big)} \\
&= \frac{1}{1 + \exp\big(r(x, y_2) - r(x, y_1)\big)} \\
&= \frac{1}{1 + \big[\exp\big(s(x, y_2) - s(x, y_1)\big)\big]^C} \to
\begin{cases}
1, & s(x, y_1) > s(x, y_2), \\
0, & s(x, y_1) < s(x, y_2), \\
0.5, & s(x, y_1) = s(x, y_2),
\end{cases}
\end{aligned}
$$

as $C \to +\infty$. This results in the label $y_1 \succ y_2$ if and only if $s(x, y_1) > s(x, y_2)$ and vice versa, which is exactly how the dataset $\mathcal{D} = \{(x, y_1, y_2)\}$ is constructed in our case (under the edge case of $s(x, y_1) = s(x, y_2)$, a random preference is assigned).

It is also worth noting that, though taking $C \to +\infty$ may appear extreme, using the Bradley-Terry model along with $r(x, y) = C \cdot s(x, y)$ with a moderate positive $C$ to generate the dataset $\mathcal{D}$ also makes sense. For example, when taking $C = 1$, one would assign the preference $y_1 \succ y_2$ with probability $\sigma\big(s(x, y_1) - s(x, y_2)\big)$ where $\sigma$ is the sigmoid function. This also reflects the binary classification nature of the detector: choosing threshold value $\hat{s}(x)$ for human-generated tasks and assigning the "human" label to response $y$ with probability $\sigma\big(s(x, y) - \hat{s}(x)\big)$ is akin to a logistic regression model.

### B.2   Proof of Lemma 3.1

Before we prove Lemma 3.1, we first introduce some useful notations. Given input text $x \in \mathcal{X}$, denote by $r^*(x) := \max_y r(x, y)$ the maximal possible reward attributed to the prompt-response pair $(x, y)$ given $x$. Next, given a text generation process $M$, denote by $\Delta_M(x) := r^*(x) - \mathbb{E}_{y \sim \pi_M(\cdot|x)} r(x, y)$ the *suboptimality* of $M$ according to reward $r$. Intuitively, the larger this suboptimality gap, the smaller the expected reward for texts generated by $M$, and thus the more likely the detector will label the generated texts as machine-generated. With this we establish the following assumption stating the effectiveness of the detector:

**Assumption B.1.** *The implicit reward $r(x, y)$ from the detector $D$ favors human-generated texts $H(y|x)$ over machine-generated texts of the reference model $M^{\text{ref}}(y|x)$, but $H$ does not achieve optimal reward. More formally, for a reasonable prior distribution of input texts $\mathcal{P}(\mathcal{X})$,*

$$
\mathbb{E}_{x \sim \mathcal{P}(\mathcal{X})} r^*(x) > \mathbb{E}_{x \sim \mathcal{P}(\mathcal{X}), y \sim \pi_H(\cdot|x)} r(x, y) > \mathbb{E}_{x \sim \mathcal{P}(\mathcal{X}), y \sim \pi_M(\cdot|x)} r(x, y). \tag{6}
$$

A direct effect of this assumption is the following suboptimality gap relation:

$$
0 < \mathbb{E}_{x \sim \mathcal{P}(\mathcal{X})} \Delta_H(x) < \mathbb{E}_{x \sim \mathcal{P}(\mathcal{X})} \Delta_{M^{\text{ref}}}(x).
$$

Notice that we assume the human generative process $H$ does not maximize reward, i.e. $\mathbb{E}_{y\sim\pi_H(\cdot|x)}r(x,y) \neq r^*(x)$. This is natural since one would expect a detector to overfit the most apparent differences between human- and machine-generated texts, and hence the implicit reward does not completely reflect how human-like the response texts are.

With this we now restate Lemma 3.1 in more technical terms and present its proof:

**Lemma B.2.** *For any target compression ratio* $\lambda \in (0,1)$*, there exists* $\beta \in (0,+\infty)$ *such that the optimal model* $M^*$ *which minimizes the DPO objective* $\mathcal{L}_{DPO}$ *in equation 3 achieves* $\mathbb{E}_{x\sim\mathcal{P}(\mathcal{X})}\Delta_{M^*}(x) = \lambda\mathbb{E}_{x\sim\mathcal{P}(\mathcal{X})}\Delta_{M^{\mathrm{ref}}}(x)$*. In particular, if we take*

$$\lambda = \frac{\mathbb{E}_{x\sim\mathcal{P}(\mathcal{X})}\Delta_H(x)}{\mathbb{E}_{x\sim\mathcal{P}(\mathcal{X})}\Delta_{M^{\mathrm{ref}}}(x)},$$

*the fine-tuned model* $M^*$ *is indistinguishable from* $H$ *in that*

$$\mathbb{E}_{x\sim\mathcal{P}(\mathcal{X}),y\sim\pi_{M^*}(\cdot|x)}r(x,y) = \mathbb{E}_{x\sim\mathcal{P}(\mathcal{X}),y\sim\pi_H(\cdot|x)}r(x,y).$$

*Proof of Lemma B.2.* Based on conclusions of previous work (Rafailov et al., 2024), the optimal solution to both the KL-constrained reward maximization objective in equation 2 and the DPO objective in equation 3 is

$$\pi_{M^*}(y|x) = \frac{1}{Z(x)}\pi_{M^{\mathrm{ref}}}(y|x)\exp\left(\frac{1}{\beta}r(x,y)\right), \tag{7}$$

where the normalization factor $Z(x) := \sum_y \pi_{M^{\mathrm{ref}}}(y|x)\exp\left(\frac{1}{\beta}r(x,y)\right)$. With this we can calculate the expected reward under $\pi_{M^*}$ as

$$\mathbb{E}_{y\sim\pi_{M^*}(\cdot|x)}r(x,y) = \sum_y \pi_{M^*}(y|x)r(x,y)$$

$$= \sum_y \frac{1}{Z(x)}\pi_{M^{\mathrm{ref}}}(y|x)\exp\left(\frac{1}{\beta}r(x,y)\right)r(x,y). \tag{8}$$

Notice now that when $\beta \to 0+0$,

$$\pi_{M^*}(y|x) = \frac{\pi_{M^{\mathrm{ref}}}(y|x)\exp\left(\frac{1}{\beta}r(x,y)\right)}{\sum_{y'}\pi_{M^{\mathrm{ref}}}(y'|x)\exp\left(\frac{1}{\beta}r(x,y')\right)}$$

$$= \frac{\pi_{M^{\mathrm{ref}}}(y|x)}{\sum_{y'}\pi_{M^{\mathrm{ref}}}(y'|x)\exp\left(\frac{1}{\beta}[r(x,y')-r(x,y)]\right)}$$

$$\to \begin{cases} 0, & y \notin \arg\max_y r(x,y), \\ \pi_{M^{\mathrm{ref}}}(y|x)/\sum_{y'\in\arg\max_y r(x,y)}\pi_{M^{\mathrm{ref}}}(y'|x), & y \in \arg\max_y r(x,y), \end{cases}$$

which means $\pi_{M^*}(y|x) \neq 0$ if and only if the response $y$ is within the optimal set $\arg\max_y r(x,y) = \{y^*|r(x,y^*) = r^*(x)\}$. This leads to

$$\lim_{\beta\to 0+0}\mathbb{E}_{y\sim\pi_{M^*}(\cdot|x)}r(x,y) = r^*(x) = \max_y r(x,y), \tag{9}$$

which means as $\beta$ approaches 0, the reward is maximized.

On the other hand, when $\beta \to +\infty$, we have

$$Z(x) \to \sum_y \pi_{M^{\mathrm{ref}}}(y|x) = 1,$$

and so

$$\lim_{\beta\to+\infty}\mathbb{E}_{y\sim\pi_{M^*}(\cdot|x)}r(x,y) = \sum_y \pi_{M^{\mathrm{ref}}}(y|x)r(x,y) = \mathbb{E}_{y\sim\pi_{M^{\mathrm{ref}}}(\cdot|x)}r(x,y). \tag{10}$$

Next, we prove this expected reward is also monotonically decreasing with respect to $\beta$. Taking the derivative of equation 8 to $\beta$ gives us

$$
\frac{\partial \mathbb{E}_{y \sim \pi_{M^*}(\cdot|x)} r(x, y)}{\partial \beta}
$$

$$
= \frac{\sum_y \pi_{M^{\mathrm{ref}}}(y|x) \exp\left(\frac{1}{\beta} r(x, y)\right)\left(-\frac{1}{\beta^2} r(x, y)\right) r(x, y)}{Z(x)}
$$

$$
- \frac{\left[\sum_y \pi_{M^{\mathrm{ref}}}(y|x) \exp\left(\frac{1}{\beta} r(x, y)\right)\left(-\frac{1}{\beta^2} r(x, y)\right)\right] \cdot \left[\sum_y \pi_{M^{\mathrm{ref}}}(y|x) \exp\left(\frac{1}{\beta} r(x, y)\right) r(x, y)\right]}{Z^2(x)}
$$

$$
= \frac{1}{\beta^2}\left[-\sum_y \pi_{M^*}(y|x) r^2(x, y) + \left(\sum_y \pi_{M^*}(y|x) r(x, y)\right)^2\right]
$$

$$
= \frac{1}{\beta^2}\left[-\mathbb{E}_{y \sim \pi_{M^*}(\cdot|x)} r^2(x, y) + \left(\mathbb{E}_{y \sim \pi_{M^*}(\cdot|x)} r(x, y)\right)^2\right] \leq 0,
$$

where we plugged in the solution to the DPO objective from equation 7 for the second equality, and the inequality is due to the Cauchy-Schwartz inequality. Combining this with equation 9 and equation 10, we see that $\mathbb{E}_{y \sim \pi_{M^*}(\cdot|x)} r(x, y)$ increases from the expected reward of the reference model $\mathbb{E}_{y \sim \pi_{M^{\mathrm{ref}}}(\cdot|x)} r(x, y)$ to the optimal reward $r^*(x)$ as $\beta$ goes from $+\infty$ to 0. Taking a final outer expectation in $x$, we have the same conclusion that

$$
\mathbb{E}_{x \sim \mathcal{P}(\mathcal{X})} \Delta_{M^*}(x) = r^*(x) - \mathbb{E}_{y \sim \pi_{M^*}(\cdot|x)} r(x, y)
$$

increases from 0 to $\mathbb{E}_{x \sim \mathcal{P}(\mathcal{X})} \Delta_{M^{\mathrm{ref}}}(x, y)$ as $\beta$ increases from 0 to $+\infty$. Therefore for any $\lambda \in (0, 1)$, there exists $\beta \in (0, +\infty)$ such that

$$
\mathbb{E}_{x \sim \mathcal{P}(\mathcal{X})} \Delta_{M^*}(x) = \lambda \mathbb{E}_{x \sim \mathcal{P}(\mathcal{X})} \Delta_{M^{\mathrm{ref}}}(x, y).
$$

$\square$

### B.3 PROOF OF THEOREM 3.2

In this section we prove Theorem 3.2, which we restate below, using subscripts $l$, $s$ to denote large and small language models respectively for clarity:

**Theorem B.3.** *Assuming the small fine-tuned model $M_s$ achieves optimum according to the DPO objective with $\beta = \beta_0$, our proposed inference model $M'$ in equation 4 is equivalent to an alternative large model $M_l^*$ optimally fine-tuned on the DPO objective with $\beta = \beta_0/\alpha$.*

*Proof of Theorem B.3.* According to equation 4, the next-token prediction logits are

$$
\pi_{M'}(y_t|x, y_{<t}) = \frac{1}{Z_{x, y_{<t}}} \pi_{M_l^{\mathrm{ref}}}(y_t|x, y_{<t})\left(\frac{\pi_{M_s^*}(y_t|x, y_{<t})}{\pi_{M_s^{\mathrm{ref}}}(y_t|x, y_{<t})}\right)^\alpha.
$$

Taking a cumulative product for $t = 1, \cdots, T$, we have

$$
\pi_{M'}(y|x) = \prod_{t=1}^T \pi_{M'}(y_t|x, y_{<t})
$$

$$
= \prod_{t=1}^T \frac{1}{Z_{x, y_{<t}}} \pi_{M_l^{\mathrm{ref}}}(y_t|x, y_{<t})\left(\frac{\pi_{M_s^*}(y_t|x, y_{<t})}{\pi_{M_s^{\mathrm{ref}}}(y_t|x, y_{<t})}\right)^\alpha
$$

$$
= \frac{1}{Z_l(x)}\left[\prod_{t=1}^T \pi_{M_l^{\mathrm{ref}}}(y_t|x, y_{<t})\right]\left(\frac{\prod_{t=1}^T \pi_{M_s^*}(y_t|x, y_{<t})}{\prod_{t=1}^T \pi_{M_s^{\mathrm{ref}}}(y_t|x, y_{<t})}\right)^\alpha
$$

$$
= \frac{1}{Z_l(x)} \pi_{M_l^{\mathrm{ref}}}(y|x)\left(\frac{\pi_{M_s^*}(y|x)}{\pi_{M_s^{\mathrm{ref}}}(y|x)}\right)^\alpha,
$$

where for the third inequality we substituted $\prod_{t=1}^{T} Z_{x,y<t}$ with $Z_l(x)$, which can be shown to be independent from $y$:

$$
\begin{aligned}
Z_l(x) &= \prod_{t=1}^{T} \sum_{y_t} \pi_{M_l^{\mathrm{ref}}}(y_t|x, y_{<t}) \left( \frac{\pi_{M_s^*}(y_t|x, y_{<t})}{\pi_{M_s^{\mathrm{ref}}}(y_t|x, y_{<t})} \right)^{\alpha} \\
&= \sum_{y_1, \cdots, y_T} \left[ \prod_{t=1}^{T} \pi_{M_l^{\mathrm{ref}}}(y_t|x, y_{<t}) \right] \left( \frac{\prod_{t=1}^{T} \pi_{M_s^*}(y_t|x, y_{<t})}{\prod_{t=1}^{T} \pi_{M_s^{\mathrm{ref}}}(y_t|x, y_{<t})} \right)^{\alpha} \\
&= \sum_{y} \pi_{M_l^{\mathrm{ref}}}(y|x) \left( \frac{\pi_{M_s^*}(y|x)}{\pi_{M_s^{\mathrm{ref}}}(y|x)} \right)^{\alpha}.
\end{aligned}
$$

Now using the close-form solution to DPO in equation 7, we have

$$
\begin{aligned}
\pi_{M'}(y|x) &= \frac{1}{Z_l(x)} \pi_{M_l^{\mathrm{ref}}}(y|x) \left( \frac{\pi_{M_s^*}(y|x)}{\pi_{M_s^{\mathrm{ref}}}(y|x)} \right)^{\alpha} \\
&= \frac{1}{Z_l(x)} \pi_{M_l^{\mathrm{ref}}}(y|x) \left( \frac{\exp\left( \frac{1}{\beta_0} r(x, y) \right)}{Z_s(x)} \right)^{\alpha} \\
&= \frac{1}{Z(x)} \pi_{M_l^{\mathrm{ref}}}(y|x) \exp\left( \frac{\alpha}{\beta_0} r(x, y) \right),
\end{aligned}
$$

where for the last equality we again used $Z(x) := Z_l(x) \cdot Z_s(x)^{\alpha}$ to simplify the normalization factor. Comparing this again to the DPO solution in equation 7, this is exactly the same as the optimal model for fine-tuning the LLM $M_l$ on $\mathcal{D}$ with $\beta = \beta_0 / \alpha$, thus completing the proof. $\square$

## C   ADDITIONAL RELATED WORK

**Proxy Approaches to Accelerate Fine-tuning.** Proxy "tuning" at decoding time is a popular method for efficient fine-tuning. It uses a proxy model during the decoding phase to reduce or eliminate the need for fine-tuning LLMs. Emulated fine-tuning (Mitchell et al., 2023b) and proxy-tuning (Liu et al., 2024a) balance fine-tuning and pre-training by decoupling the fine-tuning model scales, transferring knowledge from a fine-tuned small language model to a larger one. Furthermore, DeRa (Liu et al., 2024b) and ARGS (Khanov et al., 2024) have explored merging auxiliary models at the output level to learn a trade-off between reward and regularization, guiding the text generation process. (Huang et al., 2024) leverages a reward model to guide LLM realignment toward a custom objective. These decoding alignment approaches that merge logits have been applied in various tasks (Xu et al., 2024; Chen et al., 2023a), where (Xu et al., 2024) adopts a safety-aware decoding strategy to defend against LLM jailbreaks, and (Chen et al., 2023a) innovatively uses speculative sampling in transformer decoding to accelerate LLMs. While our work shares a similar vision with these proxy fine-tuning methods in prior or contemporary research, our objective is to fine-tune an SLM toward an optimal reward until it reaches the same level as human process according to a scoring detector, which adapts the LLM to achieve the same expected reward, thereby evading detection.

## D   EVALUATION METRICS

Throughout our experiments, we employ the Area Under the Receiver Operating Characteristic Curve (AUROC) and the Area Under the Precision-Recall Curve (AUPRC) as primary metrics to evaluate the performance of each detector. To assess the utility and quality of the generated text, we utilize BERTScore and ROUGE-1/2/L metrics. We provide a detailed explanation of these metrics.

**AUROC**. Area Under the Receiver Operating Characteristic (AUROC) measures the detection accuracy by evaluating the area under the receiver operating characteristic curve, indicating the probability that a classifier ranks a random machine-generated text higher than a random human-written text, with a value ranging from 0.0 to 1.0.

Table 4: Detailed AUROCs of the white-box detectors and generation utility scores in Table 1 on the texts generated by different models on the dataset OpenWebText, WritingPrompts and PubMedQA respectively. The generation utilities of texts produced by HUMPA and the source model are within the budget of $\Delta S_{Bert} \leq 0.02$ and $\Delta$ROUGE-1 $\leq 0.03$.

| Dataset | Models→ | Llama2-13B | HUMPA (Llama2-7B) | Llama3-70B | HUMPA (Llama3-8B) | Mixtral-8x7B | HUMPA (Mistral-7B) |
|---------|---------|-----------|-------------------|------------|-------------------|--------------|---------------------|
| | | The Generation Utility | | | | | |
| | $S_{\text{Bert}}$ | 0.8355 | 0.8231 | 0.8229 | 0.8078 | 0.8280 | 0.8141 |
| | ROUGE-1 | 0.2750 | 0.2506 | 0.2730 | 0.2447 | 0.2709 | 0.2413 |
| | ROUGE-2 | 0.0616 | 0.0529 | 0.0576 | 0.0499 | 0.0574 | 0.0487 |
| | ROUGE-L | 0.1564 | 0.1435 | 0.1529 | 0.1400 | 0.1513 | 0.1360 |
| | | The White-box Setting | | | | | |
| OpenWebText | Likelihood | 0.9986 | 0.7993 | 0.9985 | 0.8635 | 0.7287 | 0.1878 |
| | LogRank | 0.9982 | 0.7708 | 0.9974 | 0.8135 | 0.7585 | 0.1826 |
| | LRR | 0.7548 | 0.5133 | 0.6936 | **0.3905** | 0.8183 | 0.1841 |
| | NPR | 0.9964 | 0.9130 | 0.9822 | 0.8675 | 0.8365 | 0.5649 |
| | DNA-GPT | 0.9721 | **0.5686** | 0.9900 | 0.8024 | 0.7012 | **0.1153** |
| | DetectGPT | 0.8060 | 0.7422 | 0.6535 | 0.5792 | 0.3255 | 0.2048 |
| | Fast-DetectGPT | 0.9998 | 0.9926 | 1.0000 | 0.9971 | 0.9972 | 0.3786 |
| | | The Generation Utility | | | | | |
| | $S_{\text{Bert}}$ | 0.8054 | 0.8076 | 0.8086 | 0.7979 | 0.8142 | 0.8057 |
| | ROUGE-1 | 0.2285 | 0.2322 | 0.2553 | 0.2341 | 0.2613 | 0.2325 |
| | ROUGE-2 | 0.0433 | 0.0438 | 0.0474 | 0.0433 | 0.0484 | 0.0428 |
| | ROUGE-L | 0.1312 | 0.1333 | 0.1457 | 0.1348 | 0.1436 | 0.1297 |
| | | The White-box Setting | | | | | |
| Writing | Likelihood | 0.9999 | 0.8383 | 1.0000 | 0.9092 | 0.9691 | 0.3857 |
| | LogRank | 0.9999 | 0.8208 | 1.0000 | 0.8875 | 0.9742 | 0.3736 |
| | LRR | 0.9623 | 0.6908 | 0.9528 | **0.3824** | 0.9616 | 0.2629 |
| | NPR | 0.9984 | 0.9098 | 0.9948 | 0.7590 | 0.9723 | 0.7019 |
| | DNA-GPT | 0.9749 | **0.5255** | 0.9934 | 0.8273 | 0.7909 | **0.1787** |
| | DetectGPT | 0.8795 | 0.8151 | 0.8571 | 0.8295 | 0.6573 | 0.4776 |
| | Fast-DetectGPT | 0.9999 | 0.8783 | 0.9998 | 0.9866 | 0.9888 | 0.5968 |
| | | The Generation Utility | | | | | |
| | $S_{\text{Bert}}$ | 0.8717 | 0.8676 | 0.8513 | 0.8323 | 0.8605 | 0.8468 |
| | ROUGE-1 | 0.3804 | 0.3547 | 0.3506 | 0.3268 | 0.3684 | 0.3396 |
| | ROUGE-2 | 0.2206 | 0.2027 | 0.2006 | 0.1855 | 0.2076 | 0.1883 |
| | ROUGE-L | 0.3108 | 0.2888 | 0.2848 | 0.2650 | 0.2971 | 0.2714 |
| | | The White-box Setting | | | | | |
| PubMed | Likelihood | 1.0000 | 0.9455 | 1.0000 | 0.9483 | 0.9817 | 0.8455 |
| | LogRank | 0.9999 | 0.9355 | 1.0000 | 0.9297 | 0.9890 | 0.8231 |
| | LRR | 0.8471 | 0.7860 | 0.9072 | 0.7524 | 0.9318 | 0.6499 |
| | NPR | 0.9776 | 0.9215 | 0.9737 | 0.8896 | 0.8154 | 0.7219 |
| | DNA-GPT | 0.9976 | **0.9026** | 0.9891 | 0.8932 | 0.9731 | 0.7224 |
| | DetectGPT | 0.9891 | 0.9718 | 0.9832 | 0.9632 | 0.9231 | 0.9221 |
| | Fast-DetectGPT | 0.9850 | 0.9076 | 0.9857 | **0.6754** | 0.9939 | **0.4017** |

Table 5: Additional results following Table 4, the black-box AUROC results on the dataset Open-WebText, WritingPrompts and PubMedQA respectively.

| Dataset | Models→ Detectors↓ | Llama2-13B | HUMPA (Llama2-7B) | Llama3-70B | HUMPA (Llama3-8B) | Mixtral-8x7B | HUMPA (Mistral-7B) |
|---|---|---|---|---|---|---|---|
| OpenWebText | Roberta-base | 0.9681 | 0.8454 | 0.9188 | 0.7681 | 0.7997 | 0.6339 |
| | Roberta-large | 0.9534 | 0.8529 | 0.9022 | 0.7624 | 0.8136 | 0.7349 |
| | Likelihood(Neo-2.7) | 0.9986 | 0.5792 | 0.9342 | 0.6313 | 0.6369 | 0.1052 |
| | LogRank(Neo-2.7) | 0.9627 | 0.5927 | 0.9446 | 0.6224 | 0.6554 | 0.1033 |
| | LRR(Neo-2.7) | 0.9680 | 0.6175 | 0.9322 | **0.5631** | 0.6791 | 0.1200 |
| | NPR(Neo-2.7) | 0.8867 | 0.6220 | 0.8781 | 0.6439 | 0.6815 | 0.2912 |
| | DNA-GPT(Neo-2.7) | 0.6165 | **0.3203** | 0.7786 | 0.5652 | 0.5329 | 0.1011 |
| | DetectGPT(T5-3B/Neo-2.7) | 0.7416 | 0.5102 | 0.5491 | 0.3867 | 0.5650 | 0.2505 |
| | Fast-DetectGPT(GPT-J/Neo-2.7) | 0.9909 | 0.7088 | 0.9829 | 0.7988 | 0.7816 | 0.0544 |
| | Binoculars(Falcon-7B) | 1.0000 | 0.7903 | 0.9990 | 0.8830 | 0.8840 | **0.0439** |
| Writing | Roberta-base | 0.9673 | 0.8548 | 0.9169 | 0.7144 | 0.8018 | 0.6184 |
| | Roberta-large | 0.9401 | 0.8548 | 0.8720 | 0.6728 | 0.7818 | 0.6975 |
| | Likelihood(Neo-2.7) | 0.9926 | 0.6416 | 0.9830 | 0.7355 | 0.8588 | 0.2378 |
| | LogRank(Neo-2.7) | 0.9940 | 0.6310 | 0.9823 | 0.7115 | 0.8596 | 0.2280 |
| | LRR(Neo-2.7) | 0.9876 | 0.8656 | 0.9595 | **0.2918** | 0.8305 | 0.1612 |
| | NPR(Neo-2.7) | 0.9134 | 0.8414 | 0.9447 | 0.5643 | 0.8027 | 0.3523 |
| | DNA-GPT(Neo-2.7) | 0.7514 | **0.3666** | 0.8483 | 0.6197 | 0.6711 | 0.1755 |
| | DetectGPT(T5-3B/Neo-2.7) | 0.8678 | 0.6131 | 0.8621 | 0.6801 | 0.7175 | 0.3316 |
| | Fast-DetectGPT(GPT-J/Neo-2.7) | 0.9974 | 0.6466 | 0.9842 | 0.7736 | 0.8837 | **0.1505** |
| | Binoculars(Falcon-7B) | 1.0000 | 0.6773 | 0.9990 | 0.8771 | 0.9539 | 0.2019 |
| PubMed | Roberta-base | 0.7779 | 0.6704 | 0.6317 | **0.1976** | 0.4960 | 0.3995 |
| | Roberta-large | 0.7686 | 0.6510 | 0.6648 | 0.2315 | 0.5095 | 0.4375 |
| | Likelihood(Neo-2.7) | 0.9971 | 0.8706 | 0.9924 | 0.8345 | 0.9512 | 0.7195 |
| | LogRank(Neo-2.7) | 0.9984 | 0.8691 | 0.9929 | 0.8269 | 0.9613 | 0.7243 |
| | LRR(Neo-2.7) | 0.9972 | 0.8489 | 0.9693 | 0.7535 | 0.9534 | 0.7115 |
| | NPR(Neo-2.7) | 0.7677 | 0.6089 | 0.7479 | 0.5598 | 0.6173 | 0.4471 |
| | DNA-GPT(Neo-2.7) | 0.7670 | **0.5766** | 0.6431 | 0.4040 | 0.5910 | **0.2955** |
| | DetectGPT(T5-3B/Neo-2.7) | 0.8752 | 0.7394 | 0.8975 | 0.6892 | 0.7451 | 0.5612 |
| | Fast-DetectGPT(GPT-J/Neo-2.7) | 0.9999 | 0.8613 | 0.9983 | 0.7436 | 0.9620 | 0.5720 |
| | Binoculars(Falcon-7B) | 1.0000 | 0.9012 | 0.9987 | 0.8186 | 0.9679 | 0.6702 |

**AUPRC**. Area Under the Precision-Recall Curve (AUPRC) is a measure of a detector's performance, focusing on the trade-off between precision (the accuracy of machine-generated examples) and recall (the ability to identify all machine-generated examples). An AUPRC of 1.0 means perfect precision and recall, while an AUPRC of 0.0 means the detector fails completely. This metric is useful when dealing with imbalanced datasets, where the number of positive and negative examples is not equal.

**BERTScore** (Zhang et al., 2019). BERTScore is a metric used to evaluate the quality of texts generated by the pre-trained BERT. It compares the generated text to a reference text by utilizing BERT's embeddings to match words by cosine similarity. BERTScore measures how well the generated text matches the reference text in terms of meaning and context, rather than just exact word matches. This makes it a more robust evaluation method for assessing the quality of generated texts.

**ROUGE** (Chin-Yew, 2004). Recall-Oriented Understudy for Gisting Evaluation (ROUGE) is a set of metrics used to evaluate the quality of generated text by comparing it to reference texts. We adopt the popularly used ROUGE-1, ROUGE-2, ROUGE-L. ROUGE-1 measures the overlap of unigrams (single words) between the generated text and the reference text. ROUGE-2 measures the overlap of bigrams (two consecutive words) between the generated text and the reference text. ROUGE-L measures the longest common subsequence (LCS) between the generated text and the reference text, capturing the longest sequence of words that appear in both texts in the same order. These metrics help assess how similar the generated text is to the reference text in terms of content and structure.

# E   ADDITIONAL MAIN RESULTS

The detailed AUROC results on the dataset OpenWebText, WritingPrompts and PubMedQA. More results are in Table 4 for the white-box setting and Table 5 for the black-box setting. Throughout our experiments, we run DetectGPT and NPR with default 10 perturbations, and DNA-GPT with a truncation ratio of 0.2 and 10 prefix completions. The findings from the white-box performances in Table 4 include DNA-GPT being the most fragile detector when evaluating texts generated by Llama2-13B and Mixtral-8×7B on OpenWebText and WritingPrompts, with an AUROC relative

Table 6: AUROCs of detectors and generation utility scores on text generated by different models on PHX. The humanized SLM is fine-tuned from CS. The generation utilities of texts produced by HUMPA and the source model are within the budget of $\Delta S_{Bert} \leq 0.02$ and $\Delta$ROUGE-1 $\leq 0.03$.

| Models→ | Llama2-13B | HUMPA (Llama2-7B) | Llama3-70B | HUMPA (Llama3-8B) | Mixtral-8x7B | HUMPA (Mistral-7B) |
|---|---|---|---|---|---|---|
| The Generation Utility | | | | | | |
| $S_{Bert}$ | 0.8351 | 0.8308 | 0.8118 | 0.8034 | 0.8057 | 0.7888 |
| ROUGE-1 | 0.2972 | 0.2965 | 0.2992 | 0.2712 | 0.2330 | 0.2045 |
| ROUGE-2 | 0.0831 | 0.0783 | 0.0782 | 0.0770 | 0.0641 | 0.0505 |
| ROUGE-L | 0.1700 | 0.1620 | 0.1602 | 0.1543 | 0.1369 | 0.1204 |
| The White-box Setting | | | | | | |
| Likelihood | 0.9999 | 0.8376 | 0.9974 | 0.6348 | 0.6630 | 0.2604 |
| LogRank | 0.9982 | 0.7999 | 0.9932 | 0.5553 | 0.6533 | 0.2354 |
| LRR | 0.4277 | 0.3563 | 0.2588 | 0.1800 | 0.5065 | 0.1505 |
| NPR | 0.9781 | 0.8403 | 0.9859 | 0.7343 | 0.7955 | 0.5706 |
| DNA-GPT | 0.9884 | **0.6653** | 0.9943 | **0.4812** | 0.6001 | 0.1513 |
| DetectGPT | 0.5784 | 0.5996 | 0.4987 | 0.4131 | 0.2568 | 0.1778 |
| Fast-DetectGPT | 0.9978 | 0.9881 | 0.9999 | 0.9799 | 0.8390 | **0.1319** |
| The Black-box Setting | | | | | | |
| Roberta-base | 0.8718 | 0.7801 | 0.7535 | 0.4775 | 0.7379 | 0.6387 |
| Roberta-large | 0.8132 | 0.7183 | 0.6495 | 0.4435 | 0.7339 | 0.6343 |
| Likelihood(Neo-2.7) | 0.8185 | 0.5475 | 0.9351 | 0.3526 | 0.3199 | 0.0928 |
| LogRank(Neo-2.7) | 0.8226 | 0.5458 | 0.9309 | 0.3078 | 0.2848 | 0.0691 |
| LRR(Neo-2.7) | 0.7536 | 0.5125 | 0.8463 | **0.1807** | 0.2328 | 0.0369 |
| NPR(Neo-2.7) | 0.5517 | 0.4396 | 0.6458 | 0.4298 | 0.5210 | 0.3357 |
| DNA-GPT(Neo-2.7) | 0.8078 | **0.3820** | 0.9004 | 0.2909 | 0.4249 | 0.1130 |
| DetectGPT(T5-3B/Neo-2.7) | 0.1697 | 0.1877 | 0.2994 | 0.2673 | 0.2516 | 0.1958 |
| Fast-DetectGPT(GPT-J/Neo-2.7) | 0.9673 | 0.6714 | 0.9759 | 0.3919 | 0.4697 | **0.0427** |
| Binoculars(Falcon-7B) | 0.9904 | 0.7093 | 0.9812 | 0.4127 | 0.5183 | 0.0489 |

decrease of 83.6%, the largest among all detectors. Another key finding is that Fast-DetectGPT demonstrates the most vulnerability on the PubMed dataset, with AUROC relative decreases of 31.5% and 59.6% for texts produced by HUMPA -attacked Llama3-70B and Mixtral-8×7B, respectively. Furthermore, our findings reveal that DetectGPT stands out as the most robust detector, with the smallest AUROC relative decrease on WritingPrompts for Llama2-13B and Mixtral-8×7B, and also the least decrease on PubMed across all attacked models.

In the black-box setting shown in Table 5, our findings indicate that DNA-GPT is the most fragile detector when evaluating texts generated by the HUMPA -attacked Llama2-13B across all three datasets, with relative AUROC decreases of 48.1%, 51.2%, and 50.0%. Additionally, we find that Binoculars shows the highest AUROC relative decrease (95.0%) on Mixtral-8×7B's OpenWebText outputs, while Fast-DetectGPT experiences the greatest decrease (83.0%) on WritingPrompts.

## F  ADDITIONAL CROSS-DOMAIN RESULTS

The cross-discipline AUROC results of HUMPA on PHX are in Table 6. In the white-box setting, we observe that DNA-GPT is the most vulnerable when evaluating texts generated by the attacked Llama2-13B and Llama3-70B models. On texts produced by the attacked Mixtral-8×7B, Fast-DetectGPT exhibits the largest relative decrease in performance across all detectors, with an 84.3% drop in the white-box setting and a 90.9% drop in the black-box setting.

## G  RESULTS IN AUPRC

Similar to AUROC, we include the AUPRC results on the OpenWebText, WritingPrompts and Pub-MedQA dataset in Table 11. We find that HUMPA bypasses all detectors on the texts produced by the three models. The largest relative decrease across these datasets occurs on the OpenWebText, with LRR showing a 57.5% drop in the white-box setting when evaluating texts from the attacked Mixtral-8×7B, and Binoculars exhibiting a 63.0% drop in the black-box setting for the same texts.

Table 7: Cross-language performances (AUROC) of detectors and generation utility scores on text generated by different models on Germany. The humanized SLM is fine-tuned from English. The generation utilities of texts produced by HUMPA and the source model are within the budget of $\Delta S_{Bert} \leq 0.02$ and $\Delta$ROUGE-1 $\leq 0.03$.

| Models→ | Llama2-13B | HUMPA (Llama2-7B) | Llama3-70B | HUMPA (Llama3-8B) | Mixtral-8x7B | HUMPA (Mistral-7B) |
|---|---|---|---|---|---|---|
| The Generation Utility | | | | | | |
| $S_{Bert}$ | 0.8209 | 0.8209 | 0.8158 | 0.8063 | 0.8306 | 0.8210 |
| ROUGE-1 | 0.2124 | 0.1883 | 0.1951 | 0.1934 | 0.2878 | 0.2633 |
| ROUGE-2 | 0.1108 | 0.1116 | 0.1149 | 0.1145 | 0.1314 | 0.1302 |
| ROUGE-L | 0.1747 | 0.1627 | 0.1667 | 0.1663 | 0.2209 | 0.2063 |
| The White-box Setting | | | | | | |
| Likelihood | 0.9900 | 0.3685 | 0.9596 | 0.3637 | 0.5186 | 0.2286 |
| LogRank | 0.9824 | 0.3358 | 0.9517 | 0.3262 | 0.5838 | 0.2296 |
| LRR | 0.5943 | **0.1528** | 0.6608 | **0.1812** | 0.7725 | 0.2849 |
| NPR | 0.9590 | 0.4077 | 0.5315 | 0.4171 | 0.7928 | 0.5315 |
| DNA-GPT | 0.9924 | 0.3732 | 0.9828 | 0.3957 | 0.6486 | **0.1426** |
| DetectGPT | 0.8427 | 0.4818 | 0.7219 | 0.3859 | 0.6064 | 0.5252 |
| Fast-DetectGPT | 0.9935 | 0.8676 | 0.9156 | 0.4536 | 0.9572 | 0.3490 |
| The Black-box Setting | | | | | | |
| Roberta-base | 0.5606 | 0.5291 | 0.5730 | 0.3605 | 0.4298 | 0.3389 |
| Roberta-large | 0.5508 | 0.4776 | 0.5530 | 0.3620 | 0.4859 | 0.3388 |
| Likelihood(Neo-2.7) | 0.9900 | **0.0860** | 0.4781 | 0.1780 | 0.3746 | 0.2200 |
| LogRank(Neo-2.7) | 0.9824 | 0.0972 | 0.5394 | 0.1803 | 0.3930 | 0.2204 |
| LRR(Neo-2.7) | 0.8045 | 0.1814 | 0.7876 | 0.2132 | 0.5188 | 0.2526 |
| NPR(Neo-2.7) | 0.6039 | 0.1857 | 0.5674 | 0.2801 | 0.4788 | 0.2662 |
| DNA-GPT(Neo-2.7) | 0.7496 | 0.1290 | 0.8494 | 0.2222 | 0.4889 | 0.1233 |
| DetectGPT(T5-3B/Neo-2.7) | 0.5161 | 0.1732 | 0.4787 | 0.2722 | 0.4108 | 0.2465 |
| Fast-DetectGPT(GPT-J/Neo-2.7) | 0.9127 | 0.2133 | 0.7686 | **0.1941** | 0.5536 | **0.1152** |
| Binoculars(Falcon-7B) | 0.9929 | 0.3108 | 0.9901 | 0.3265 | 0.7293 | 0.1564 |

Table 8: The performances and generation utility of HUMPA with larger $\alpha$ tested on detectors.

| | Likelihood | LogRank | LRR | NPR | DNA-GPT | DetectGPT | Fast-DetectGPT | Binoculars | $S_{Bert}$ | ROUGE-1/2/L |
|---|---|---|---|---|---|---|---|---|---|---|
| Dipper Paraphrasing | 0.8125 | 0.7998 | 0.7220 | 0.5193 | 0.6240 | 0.2675 | 0.9754 | 0.9398 | 0.8006 | 0.2076/0.0226/0.1191 |
| Query-based Substitutions | 0.9843 | 0.9921 | 0.9828 | 0.3030 | 0.7072 | 0.1914 | 0.9972 | 1.0000 | 0.7989 | 0.2015/0.0383/0.1256 |
| HUMPA ($\alpha = 1.2$) | 0.1647 | 0.1625 | 0.1599 | 0.2592 | 0.0723 | 0.2124 | 0.0794 | 0.1743 | **0.8053** | **0.2281/0.0422/0.1409** |
| HUMPA($\alpha = 1.5$) | **0.0109** | **0.0109** | **0.0117** | **0.0617** | **0.0034** | **0.0582** | **0.0007** | **0.0021** | 0.8014 | 0.2137/0.0404/0.1383 |

Table 12 lists the AUPRC results for cross-domain scenarios. We find that HUMPA bypasses the detectors in these settings. The largest relative decrease occurs in the EN→GER setting, where the black-box Likelihood evaluates texts from the attacked Llama2-13B, showing a 68.0% drop in AUPRC.

# H  MORE ANALYSIS OF $\alpha$

The ratio $\alpha$ controls the intensity of the attack on the LLM, with larger values of $\alpha$ yielding higher rewards and better detection evasion, while smaller values keep the attacked LLM closer to the source LLM, thus limiting the attack's effectiveness. We present the detector evasion performance of HUMPA with higher $\alpha$ values, such as 1.2 and 1.5, compared to two state-of-the-art baselines: the paraphrase generation attack method DIPPER (Krishna et al.) and the query-based word substitution attack method (Shi et al., 2024), as shown in Table 8. We find that HUMPA outperforms the baselines in evasion performance while maintaining high generation utility. However, when $\alpha$ increases, the generation utility diminishes. For another instance, we consider a commercial detector GPTZero (Tian & Cui, 2023). We fine-tune a Llama2-7B model using GPTZero as the scoring detector and evaluate GPTZero on text generated by the attacked Llama2-13B model. The results are presented in Table 9. We find that GPTZero also can be bypassed with $\alpha$ increases, and the generation utility accordingly decreases in a scope. This phenomenon aligns with our findings in Theorem 3.2, which highlight that $\alpha$ governs the trade-off between evasion performance and generation quality.

Technically, when $\alpha \to 0$, the attack model approaches the reference model, with quality on par with the original LLM and no evasion ability; when $\alpha \to \infty$, the attack model regresses to a deterministic model, selecting next token based on maximized probability increase from pre-trained to fine-tuned SLM, which is an extremely aggressive attacker with no concern for quality (also notice the LLM has no influence on the attacker in this extreme case). These different scenarios suggest that an adversary should choose a proper $\alpha$, balancing attack effect and text quality. Since fine-tuned SLMs can adjust the output dis-

Table 9: Performance and generation utility on GPTZero.

| | Llama2-13B | HUMPA ($\alpha=1.5$) | HUMPA ($\alpha = 2.0$) |
|---|---|---|---|
| $S_{\text{Bert}}$ | 0.8189 | 0.8075 | 0.7939 |
| ROUGE-1 | 0.2587 | 0.2217 | 0.2131 |
| ROUGE-2 | 0.0480 | 0.0452 | 0.0392 |
| ROUGE-L | 0.1497 | 0.1372 | 0.1266 |
| AUROC | 0.9951 | 0.8295 | 0.7987 |

tributions of large models during the inference, $\alpha$ can be selected at a low time cost, and concerns about robustness arise in the enhancement of such detection methods.

## I  HUMAN EVALUATION

To reliably assess the quality of texts generated by the attacked model compared to those produced by the original, unattacked model, it is essential to evaluate the perceived naturalness of the text from users' perspective. For instance, users expect the text to be smooth, coherent, and grammatically correct. This ensures that the generated text feels natural and is

Table 10: Performances of different evasion methods evaluated using Roberta-base.

| Methods | AUROC | Fluency Win Rate |
|---|---|---|
| DIPPER Paraphrasing | 0.9717 | 54.16% |
| DPO (Llama2-13B) | 0.6968 | 51.67% |
| HUMPA (Llama2-7B) | **0.6394** | **57.50%** |

easy to read. Therefore, we evaluate the quality of the text based on its fluency. We produced 120 pairs of text 150 Llama2 tokens long and with the same prefix. One from each pair was generated by base Llama2-13B, while the other was generated by Llama2-13B attacked by HUMPA with a DPO fine-tuned Llama2-7B model against Roberta-large with $\beta = 0.1$ for 5 epochs, and the attack ratio $\alpha = 1.3$ to balance between the generation utility and the evasion performance. We also include two baselines: one is DPO directly fine-tuned on Llama2-13B against Roberta-large (Nicks et al., 2024), another is DIPPER Paraphrasing (Krishna et al.). We then ask three human annotators to choose the text with better fluency when presented with each pair. The two texts were presented in a randomized order to the annotators. The results are shown in Table 10. We find that HUMPA demonstrates superior attack performance while preserving better text naturalness.

## J  SENSITIVITY OF TRAINING SIZES

The SLM is fine-tuned using DPO, and the resulting model is influenced by the size of the training data. Consequently, the training data size affects the performance of the attacked LLM in evading detection. To obtain a effective humanized SLM, a larger training size is desirable. However, increasing the training size requires more time for fine-tuning. To explore the impact of training size on both time efficiency and detection evasion performance, we fine-tuned a Llama2-7B model against Roberta-large on the OpenWebText dataset using varying training sizes.

We use LoRA to perform DPO fine-tuning on the SLM with $\beta = 0.1$, and batch size of 8 for 5 epochs. We varying the training size and record the fine-tuning runtime. We evaluate RoBERTa-base detector on text generated by the attacked Llama2-13B model with $\alpha = 1.5$, and the results are in Table 13. We find that as the training size increases, the performance of detection decreases, while the fine-tuning time grows.

Table 13: Performances on different training size.

| Training Size | AUROC | Time (hrs) |
|---|---|---|
| 1K | 0.88 | 0.27 |
| 5K | 0.71 | 1.35 |
| 8K | 0.68 | 1.90 |
| 10K | 0.62 | 2.04 |

This suggests a trade-off between efficiency and performance: increasing the training size improves evasion but reduces efficiency due to longer fine-tuning times. If prioritizing evasion performance, a larger training size might be preferable. On the contrary, if efficiency or fine-tuning time is more critical, a smaller training size provides a better balance.

## K  ANALYSIS OF SCORING DETECTOR

The scoring detector plays an important role in obtaining the reward. Theoretically, a weak detector yields relatively low reward for human-generated texts, while machine-generated texts are given relatively high reward, resulting in a reduced gap between the two. Hence an attacker can more easily bypass the detector and overfit to the weak scoring detector. Therefore at deployment, when faced with a strong target detector, the attacker's performance will suffer.

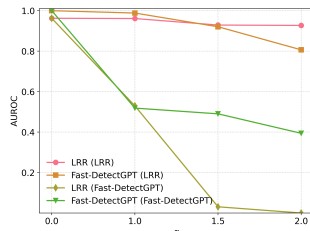

Figure 6: The performances of detectors evaluated on text generated by the attacked Llama2-13B. Each line in the figure represents a combination of an evaluation detector and a scoring detector, denoted as 'Evaluation Detector (Scoring Detector)'.

We conduct an empirical analysis of the impact of different scoring detectors on the WritingPrompts dataset. Specifically, we use the white-box LRR and Fast-DetectGPT as scoring detectors when constructing the preference data for DPO fine-tuning.

Using these two preference datasets, we fine-tune Llama2-7B with $\beta = 0.1$ to obtain two humanized SLMs as attackers. We then evaluate the white-box LRR and Fast-DetectGPT on the text generated by the attacked Llama2-13B model with varying levels of attack ratio $\alpha$ in Figure 6. We find that when LRR is used as the scoring detector, the attacked Llama2-13B model exhibits a moderate performance drop. In contrast, when Fast-DetectGPT is used for scoring, the performance drops are huge. An SLM fine-tuned against a weak scoring detector cannot perform well in the face of a strong target detector, though we surmise this gap may be simply due to the lack of high-quality human-machine labels without access to strong scoring detectors and may hence be unavoidable. In practice, although there is no assumption which specific scoring detector the attacker should choose, a strong, well-calibrated detector is desirable for this purpose.

## L  ANALYSIS OF SMALL MODEL SIZE

To study the impact of model size, we fine-tune a Llama2-7B and a TinyLlama-1.1B (Zhang et al., 2024) on WritingPrompts dataset as the SLM respectively. We use RoBERTa-large as the scoring detector and evaluate RoBERTa-base on the text generated by the attacked Llama2-13B model, each attacked with an attack ratio of $\alpha = 1.5$. The results are in Table 14. We find that different SLMs exhibit varying levels of effectiveness when attacking the LLM, with Llama2-7B being the most effective choice.

Table 14: Performance with different size of SLM.

| Model | AUROC |
|---|---|
| Llama2-13B | 0.9673 |
| HUMPA(Llama2-7B) | 0.4594 |
| HUMPA(TinyLlama-1.1B) | 0.7706 |

This suggests that when selecting an SLM for such attacks, a stronger SLM tend to demonstrate better evasion performance. However, fine-tuning a larger model and loading it during inference to attack the LLM incurs significant time and memory costs. Practitioners need to balance these factors according to their specific priorities.

## M  REAL CASES

We provide several real examples randomly selected from OpenWebText and the cross-domain scenario PHX→HSS across different models. Note that, as discussed in Section 3.3, the parameter $\alpha$ controls the trade-off between attack effectiveness and text generation quality: higher $\alpha$ leads to higher reward according to the detector, but also strays away from pre-trained LLM and thus decays in quality. With this in mind, the following examples are generated with $\alpha = 1.5$, which as shown in previous sections can be considered an effective attack. From our observations, no significant qualitative differences are apparent between the generated texts.

**Real cases of samples from OpenWebText**

**Llama2-13B**: DENVER — The New England Patriots could be without two key players for their AFC Championship Game matchup against the Denver Broncos on Sunday. Tight end Rob Gronkowski and defensive lineman Malcom Brown did not participate in practice on Wednesday, and both have been listed as questionable for the game, according to the Patriots' injury report. Gronkowski has a knee injury, while Brown has a foot injury. "It's early in the week, so we'll see how it goes," Patriots coach Bill Belichick said when asked about the status of his players. "We'll evaluate it day by day and see where we're at by the end of the week." If Gronkowski and Brown are unable to play, it would be a significant loss for the Patriots. Gronkowski is one of the team's top receiving threats

**HUMPA (Llama2-7B)**: DENVER — The New England Patriots could have the luxury of having all six of their top wide receivers available for Sunday's AFC Championship Game against the Denver Broncos. Julian Edelman, who has missed the last two games with a knee injury, was a full participant in practice on Thursday and is expected to play, a source told ESPN. The Patriots also expect to have Chris Hogan (shoulder) and Malcolm Mitchell (knee) back in the lineup, after both players missed last week's divisional round playoff game against the Houston Texans. The only receiver who did not participate in practice on Thursday was Danny Amendola, who is dealing with a thigh injury. However, he is expected to play on Sunday, per the source. The Patriots have a deep and talented receiving corps, and having all six players available could

- - - - - - - - - - - - - - - - - - - - - - - - - - - - - - - - - - - - - - - - - - - - - -

**Llama3-70B**:

On Tuesday, it was reported that New York state lawmakers had reached a deal for a $175 billion budget that would increase taxes on the wealthy and large corporations. [newline] On Wednesday, Governor Cuomo signed the budget into law. [newline] According to Cuomo, the new budget would increase the income tax rate for individuals earning over $1 million per year and corporations earning over $5 million per year. Additionally, it would implement a new tax on luxury real estate in New York City, and would establish a new tax on opioid manufacturers. [newline] Cuomo said in a statement, "This budget is a testament to what can be accomplished when lawmakers work together to benefit the people of this great state." Some critics argued that the new taxes could lead to wealthy individuals and corporations leaving the state. The Business Council of New York State said in a statement, "Increasing taxes on our most successful entrepreneurs and businesses will only encourage them to take their investments and jobs to more competitive states." However, supporters of the budget argued that it would

**HUMPA (Llama3-8B)**:

On Tuesday, it was reported that New York Giants tight end Daniel Bellinger was arrested following an incident at a dinner party at the home of one of his teammates. According to the report, Bellinger allegedly got into a physical altercation with one of the party's attendees. The team has since released a statement regarding the situation. On Wednesday, the NFLPA announced that it will be conducting its own investigation into the allegations surrounding Bellinger's arrest. This move is likely in anticipation of potential discipline from the NFL, which could potentially stem from the league's personal conduct policy. [newline] Bellinger's lawyer has since released a statement claiming that his client was acting in self-defense during the alleged altercation. [newline] Update: TMZ has released video footage of the altercation, which appears to show Bellinger allegedly punching and kicking a man multiple times. On Thursday, it was reported that the NFL has placed Bellinger on the commissioner's exempt list, effectively suspending him pending further investigation. Should Bellinger ultimately be suspended or disciplined

- - - - - - - - - - - - - - - - - - - - - - - - - - - - - - - - - - - - - - - - - - - - - -

**Mixtral-8×7B:**

After weeks of negotiations, National and Labour finally have a deal in the books to bring the Electoral Transformation Bill to fruition for an imminent vote in parliament. In a statement the two political parties issued Thursday afternoon, both said talks on the topic have been "constructive and carried out in a positive and courteous way with a good willingness to reach an agreement." The agreement they finally reached, per the statement, allows for Māori to put forward three different electoral models to be put to a binding citizens-initiated referendum. If any one of them are successful, then "the number of members from the Māori electoral specials seats, in total, would be fixed at 120%, of the MPs who would be returned with an equivalent proportion of voters in the general electoral specials seats." The two parties will put forward a joint motion of am

**HUMPA (Mistral-7B):**

After weeks of negotiations, National and Labour have thrashed out agreement on legislation that will give parties less money and force (some) MPs to live more normal lives. Labour anger at National's

insistence that there must be correlation between private and public sector pay rises and a feeling the party didn't push hard enough on funding for MPs did take its toll. But relationships appear to be good and the government suggested the parties could work together on other bills - although not one on alcohol, about which the Greens have similar gripes. The crowningpiece - literally for Helen Clark and Chester Borrows who travelled up to Parliament flanked by placard carriers demanding greater gender equality in elecitons - was an amendment to Clause 5 of the Electoral Act. This allows parties in a coalition or cumulative vote deal (eg parties laddering candidates in a district single-preference system) to

---

### Real cases of samples from PHX→HSS

**Llama2-13B**: Introduction to the Special Issue on Diversion from Standard Prosecution in the Criminal Justice System [newline] Introduction: The special issue on diversion from standard prosecution in the criminal justice system explores the various ways in which criminal cases can be diverted from the traditional prosecution process. Diversion programs offer an alternative to prosecution and punishment, and can be an effective tool for addressing the root causes of criminal behavior and promoting rehabilitation. This issue brings together a range of articles and research papers that examine the different forms of diversion, their benefits and challenges, and the impact they have on the criminal justice system. [newline] The articles in this special issue cover a range of topics related to diversion from standard prosecution, including the different types of diversion programs available, the factors that influence their effectiveness, and the role of diversion in addressing issues of racism and inequality in the criminal justice system.

**HUMPA (Llama2-7B)**: Introduction to the Special Issue on Diversion from Standard Prosecution: Toward a More Holistic and Effective Approach to Addressing Crime [newline] This special issue of the journal devotes attention to diversion from standard prosecution, a topic that has gained increasing importance in the criminal justice system. Diversion programs have been developed as an alternative to traditional criminal prosecution, with the goal of addressing the root causes of criminal behavior and reducing recidivism rates. These programs have been shown to be effective in addressing various types of crimes, including drug offenses, property crimes, and even violent crimes. [newline] The articles in this special issue explore various aspects of diversion from standard prosecution, including the effectiveness of diversion programs, the factors that contribute to their success, and the challenges and limitations that must be addressed in order to realize their full potential. [newline] One of the key themes that emerges from

- - - - - - - - - - - - - - - - - - - - - - - - - - - - - - - - - - - - - - - - - - - - - - - - - - - -

**Llama3-70B**: Acceptability of App-Based Contact Tracing for COVID-19: Cross-Country Survey Evidence [newline] **Title:** Acceptability of App-Based Contact Tracing for COVID-19: Cross-Country Survey Evidence [newline] **Background:** [newline] The COVID-19 pandemic has led to unprecedented global efforts to control the spread of the virus. One promising approach is digital contact tracing, which uses mobile apps to rapidly identify and notify individuals who have been in close proximity to someone infected with COVID-19. However, the success of such apps relies heavily on public acceptance and adoption. [newline] **Objective:** [newline] This study aims to investigate the acceptability of app-based contact tracing for COVID-19 across different countries and identify factors influencing public willingness to use such apps. [newline] **Methods:** [newline] We conducted a cross-country online survey among 10,000 participants from 15 countries, including the United States, United Kingdom, Australia, Canada, Germany, France

**HUMPA (Llama3-8B)**: Acceptability of App-Based Contact Tracing for COVID-19: Cross-Country Survey Evidence [newline] Abstract: [newline] Contact tracing is a crucial component in controlling the spread of COVID-19. The use of app-based contact tracing has been widely adopted globally, but its acceptability varies across countries. This study aimed to investigate the acceptability of app-based contact tracing and its correlates across countries. [newline] Methods: [newline] We conducted a cross-country survey of 13,993 adults from 20 countries. The survey included questions on demographic characteristics, COVID-19-related experiences, and attitudes towards app-based contact tracing. We used multivariable logistic regression to identify factors associated with acceptability. [newline] Results: [newline] The overall acceptability of app-based contact tracing was 63.1%. Country-wise, the acceptability ranged from 44.1% in Japan to 84.5% in China. In multivariable analysis, factors associated with higher acceptability included being male, younger age, higher education, previous COVID-19

- - - - - - - - - - - - - - - - - - - - - - - - - - - - - - - - - - - - - - - - - - - - - - - - - - - -

**Mixtral-8×7B:** Put More Women in Charge and Other Leadership Lessons from COVID-19 Women face significant, unique, and disproportionate risks, impacts, and challenges relative to COVID-19, both as a result of public health measures to stem the spread of the virus and from the crisis itself. However, evidence from around the world also suggests that despite facing more severe adversity, women have been at the forefront of pandemic response efforts, often leveraging their networks, knowledge, and experience forged in other crisis contexts, to take action swiftly and creatively to help those in need. Drawing on extensive global research, this paper examines the critical roles that women at all levels, from civil society to the highest corridors of power, have played in pandemic response and recovery efforts. The authors present a series of case study narratives to demonstrate that the success of COVID- response has been dependent in significant part on women in leadership, and in

**HUMPA (Mistral-7B):** Put More Women in Charge and Other Leadership Lessons from COVID-19 [newline] While COVID-19 has thrown a curveball at businesses, some are managing to fight through quite effectively. New Zealand saw lockdowns lift a lot earlier than most. It's going to open up again, sooner rather than later, and this is attributed, to a large extent due to the exceptional leadership. [newline] This begs the question – What makes exceptional leadership stick? The difference is found at the intersection of love and power. Historically, women have better skills and hold the power in matters of relationship, empathy, compassion, resilience and what Danial Goleman calls Social Intelligence at work or where EI-EQ meet. The good news is that infusing these skills into the workplace can only benefit an organisation – but is challenge at times to balance these with "masculine" behaviour based on transactional power dynamics we are used to.

Table 11: AUPRCs of detectors on the texts generated by different models on the dataset OpenWeb-Text, WritingPrompts and PubMedQA respectively.

| Dataset | Models→ | Llama2-13B | HUMPA (Llama2-7B) | Llama3-70B | HUMPA (Llama3-8B) | Mixtral-8x7B | HUMPA (Mistral-7B) |
|---|---|---|---|---|---|---|---|
| | | The White-box Setting | | | | | |
| | Likelihood | 0.9984 | 0.8705 | 0.9983 | 0.9082 | 0.6931 | 0.3485 |
| | LogRank | 0.9980 | 0.8495 | 0.9971 | 0.8702 | 0.7230 | 0.3456 |
| | LRR | 0.7582 | 0.5755 | 0.7099 | **0.4870** | 0.8079 | **0.3435** |
| | NPR | 0.9971 | 0.9400 | 0.9833 | 0.9005 | 0.7911 | 0.5569 |
| | DNA-GPT | 0.9605 | **0.6375** | 0.9890 | 0.8521 | 0.6611 | 0.3233 |
| | DetectGPT | 0.7690 | 0.6990 | 0.6471 | 0.5828 | 0.3890 | 0.3454 |
| | Fast-DetectGPT | 0.9998 | 0.9922 | 1.0000 | 0.9979 | 0.9972 | 0.4277 |
| | | The Black-box Setting | | | | | |
| OpenWebText | Roberta-base | 0.9593 | 0.7798 | 0.8723 | 0.7286 | 0.7558 | 0.5779 |
| | Roberta-large | 0.9454 | 0.8453 | 0.8970 | 0.7305 | 0.7941 | 0.7224 |
| | Likelihood(Neo-2.7) | 0.9459 | 0.6746 | 0.9334 | 0.6951 | 0.6243 | 0.3205 |
| | LogRank(Neo-2.7) | 0.9629 | 0.7008 | 0.9467 | 0.6916 | 0.6429 | 0.3200 |
| | LRR(Neo-2.7) | 0.9754 | 0.7344 | 0.9476 | **0.6650** | 0.6880 | 0.3259 |
| | NPR(Neo-2.7) | 0.8935 | 0.6823 | 0.8904 | 0.7012 | 0.6935 | 0.3908 |
| | DNA-GPT(Neo-2.7) | 0.6011 | 0.4097 | 0.7713 | 0.6041 | 0.5557 | 0.3223 |
| | DetectGPT(T5-3B/Neo-2.7) | 0.6692 | 0.5029 | 0.5298 | 0.4397 | 0.5370 | 0.3601 |
| | Fast-DetectGPT(GPT-J/Neo-2.7) | 0.9928 | 0.8046 | 0.9855 | 0.8543 | 0.7762 | 0.3112 |
| | Binoculars(Falcon-7B) | 1.0000 | **0.6413** | 0.9990 | 0.7632 | 0.8468 | **0.3131** |
| | | The White-box Setting | | | | | |
| | Likelihood | 0.9999 | 0.9046 | 1.0000 | 0.9477 | 0.9684 | 0.5342 |
| | LogRank | 0.9999 | 0.8921 | 1.0000 | 0.9326 | 0.9735 | 0.5216 |
| | LRR | 0.9686 | 0.7884 | 0.9590 | **0.5347** | 0.9626 | **0.4276** |
| | NPR | 0.9986 | 0.9397 | 0.9962 | 0.8499 | 0.9768 | 0.7603 |
| | DNA-GPT | 0.9655 | **0.6043** | 0.9894 | 0.8678 | 0.7892 | 0.3651 |
| | DetectGPT | 0.8567 | 0.8112 | 0.8540 | 0.8279 | 0.6188 | 0.4842 |
| | Fast-DetectGPT | 0.9999 | 0.8736 | 0.9998 | 0.9843 | 0.9908 | 0.6026 |
| | | The Black-box Setting | | | | | |
| Writing | Roberta-base | 0.9615 | 0.8202 | 0.8917 | 0.6602 | 0.7637 | 0.5944 |
| | Roberta-large | 0.9345 | 0.8202 | 0.8474 | 0.6532 | 0.7455 | 0.6748 |
| | Likelihood(Neo-2.7) | 0.9928 | 0.7470 | 0.9833 | 0.8029 | 0.8609 | 0.4113 |
| | LogRank(Neo-2.7) | 0.9943 | 0.7402 | 0.9836 | 0.7847 | 0.8643 | 0.4045 |
| | LRR(Neo-2.7) | 0.9876 | 0.8656 | 0.9656 | **0.4397** | 0.8463 | 0.3629 |
| | NPR(Neo-2.7) | 0.9134 | 0.8414 | 0.9533 | 0.6547 | 0.8332 | 0.4615 |
| | DNA-GPT(Neo-2.7) | 0.7253 | 0.4352 | 0.8301 | 0.6663 | 0.6693 | 0.3522 |
| | DetectGPT(T5-3B/Neo-2.7) | 0.8369 | 0.6128 | 0.8351 | 0.6854 | 0.6893 | 0.4075 |
| | Fast-DetectGPT(GPT-J/Neo-2.7) | 0.9981 | 0.7535 | 0.9881 | 0.8335 | 0.8904 | 0.3656 |
| | Binoculars(Falcon-7B) | 1.0000 | **0.5458** | 0.9988 | 0.7519 | 0.9245 | **0.3465** |
| | | The White-box Setting | | | | | |
| | Likelihood | 1.0000 | 0.9672 | 1.0000 | 0.9670 | 0.9866 | 0.8725 |
| | LogRank | 0.9999 | 0.9613 | 1.0000 | 0.9567 | 0.9888 | 0.8577 |
| | LRR | 0.8520 | 0.8257 | 0.9146 | 0.8118 | 0.9176 | 0.6755 |
| | NPR | 0.9769 | 0.9274 | 0.9723 | 0.8929 | 0.7603 | 0.6594 |
| | DNA-GPT | 0.9965 | 0.9360 | 0.9733 | 0.8931 | 0.9720 | 0.7620 |
| | DetectGPT | 0.9893 | 0.9753 | 0.9851 | 0.9673 | 0.9074 | 0.9091 |
| | Fast-DetectGPT | 0.9859 | **0.9173** | 0.9891 | **0.6946** | 0.9950 | **0.4771** |
| | | The Black-box Setting | | | | | |
| PubMed | Roberta-base | 0.6820 | 0.5615 | 0.5334 | 0.3424 | 0.4545 | 0.4079 |
| | Roberta-large | 0.6954 | 0.5737 | 0.5899 | **0.3510** | 0.4741 | 0.4343 |
| | Likelihood(Neo-2.7) | 0.9972 | 0.9142 | 0.9926 | 0.8768 | 0.9481 | 0.7570 |
| | LogRank(Neo-2.7) | 0.9985 | 0.9152 | 0.9933 | 0.8739 | 0.9603 | 0.7646 |
| | LRR(Neo-2.7) | 0.9974 | 0.9020 | 0.9746 | 0.8131 | 0.9551 | 0.7475 |
| | NPR(Neo-2.7) | 0.6687 | 0.5507 | 0.6428 | 0.5017 | 0.5306 | 0.4319 |
| | DNA-GPT(Neo-2.7) | 0.7065 | 0.6068 | 0.6142 | 0.4528 | 0.5367 | 0.3807 |
| | DetectGPT(T5-3B/Neo-2.7) | 0.8473 | 0.7562 | 0.8844 | 0.6953 | 0.6953 | 0.5656 |
| | Fast-DetectGPT(GPT-J/Neo-2.7) | 0.9999 | 0.9160 | 0.9984 | 0.8206 | 0.9666 | 0.6567 |
| | Binoculars(Falcon-7B) | 1.0000 | **0.8120** | 0.9986 | 0.7414 | 0.9679 | **0.6563** |

Table 12: AUPRCs of detectors in on the texts generated by different models in the cross-domain scenarios. CS→PHX denotes that the detectors are evaluated on the texts sampled from PHX while the humanized small language model is fine-tuned on CS. PHX→HSS and EN→GER follow the same pattern.

| Dataset | Models→ | Llama2-13B | HUMPA (Llama2-7B) | Llama3-70B | HUMPA (Llama3-8B) | Mixtral-8x7B | HUMPA (Mistral-7B) |
|---|---|---|---|---|---|---|---|
| | The White-box Setting | | | | | | |
| | Likelihood | 0.9999 | 0.9028 | 0.9965 | 0.7530 | 0.6824 | 0.4265 |
| | LogRank | 0.9983 | 0.8784 | 0.9848 | 0.6862 | 0.6594 | 0.3984 |
| | LRR | 0.4680 | 0.4343 | 0.3702 | 0.3403 | 0.5274 | 0.3340 |
| | NPR | 0.9792 | 0.8876 | 0.9647 | 0.7886 | 0.7919 | 0.6177 |
| | DNA-GPT | 0.9890 | **0.7616** | 0.9933 | **0.6208** | 0.6312 | 0.3773 |
| | DetectGPT | 0.5483 | 0.5809 | 0.4789 | 0.4610 | 0.3666 | 0.3388 |
| | Fast-DetectGPT | 0.9972 | 0.9839 | 0.9999 | 0.9800 | 0.8298 | **0.3256** |
| | The Black-box Setting | | | | | | |
| CS→PHX | Roberta-base | 0.8615 | 0.7001 | 0.6816 | 0.4681 | 0.6969 | 0.5744 |
| | Roberta-large | 0.7883 | 0.6516 | 0.6173 | 0.4363 | 0.7121 | 0.6121 |
| | Likelihood(Neo-2.7) | 0.8469 | 0.6800 | 0.9383 | 0.5102 | 0.4198 | 0.3237 |
| | LogRank(Neo-2.7) | 0.8568 | 0.6815 | 0.9362 | 0.4679 | 0.3895 | 0.3149 |
| | LRR(Neo-2.7) | 0.8152 | 0.6537 | 0.8708 | 0.3733 | 0.3686 | 0.3089 |
| | NPR(Neo-2.7) | 0.5995 | 0.5032 | 0.6511 | 0.5387 | 0.5927 | 0.4432 |
| | DNA-GPT(Neo-2.7) | 0.7973 | 0.4665 | 0.8675 | 0.4299 | 0.5017 | 0.3542 |
| | DetectGPT(T5-3B/Neo-2.7) | 0.3351 | 0.3407 | 0.3817 | 0.3710 | 0.3785 | 0.3540 |
| | Fast-DetectGPT(GPT-J/Neo-2.7) | 0.9728 | 0.7826 | 0.9807 | 0.5259 | 0.4977 | **0.3100** |
| | Binoculars(Falcon-7B) | 0.9848 | **0.5682** | 0.9661 | **0.4109** | 0.4870 | 0.3126 |
| | The White-box Setting | | | | | | |
| | Likelihood | 0.9997 | 0.9118 | 0.9999 | 0.8960 | 0.8057 | 0.5624 |
| | LogRank | 1.0000 | 0.9224 | 0.9978 | 0.8562 | 0.8077 | 0.5352 |
| | LRR | 0.8046 | 0.6941 | 0.4602 | 0.4202 | 0.7554 | 0.4167 |
| | NPR | 0.9993 | 0.9624 | 0.9959 | 0.9519 | 0.9266 | 0.8403 |
| | DNA-GPT | 0.9585 | **0.7985** | 0.9984 | **0.8320** | 0.6860 | 0.4240 |
| | DetectGPT | 0.9133 | 0.8215 | 0.9055 | 0.7896 | 0.5455 | 0.5506 |
| | Fast-DetectGPT | 0.9966 | 0.9377 | 0.9958 | 0.9659 | 0.9686 | **0.5133** |
| | The Black-box Setting | | | | | | |
| PHX→HSS | Roberta-base | 0.8390 | 0.7328 | 0.7057 | 0.6085 | 0.6380 | 0.5419 |
| | Roberta-large | 0.8422 | 0.7272 | 0.7631 | 0.5784 | 0.6832 | 0.6084 |
| | Likelihood(Neo-2.7) | 0.9599 | 0.7121 | 0.9355 | 0.6720 | 0.6370 | 0.4104 |
| | LogRank(Neo-2.7) | 0.9647 | 0.7821 | 0.9334 | 0.6337 | 0.6259 | 0.3846 |
| | LRR(Neo-2.7) | 0.9511 | 0.7550 | 0.8927 | **0.5139** | 0.6022 | 0.3490 |
| | NPR(Neo-2.7) | 0.9444 | 0.8043 | 0.9177 | 0.7816 | 0.8164 | 0.6672 |
| | DNA-GPT(Neo-2.7) | 0.8803 | 0.6267 | 0.9453 | 0.6685 | 0.6178 | 0.3734 |
| | DetectGPT(T5-3B/Neo-2.7) | 0.7280 | 0.6249 | 0.7580 | 0.6533 | 0.6275 | 0.5235 |
| | Fast-DetectGPT(GPT-J/Neo-2.7) | 0.9909 | 0.8393 | 0.9858 | 0.7171 | 0.7826 | 0.5133 |
| | Binoculars(Falcon-7B) | 0.9957 | **0.6370** | 0.9990 | 0.5766 | 0.7493 | **0.3382** |
| | The White-box Setting | | | | | | |
| | Likelihood | 0.9903 | 0.5519 | 0.9257 | 0.4742 | 0.4865 | 0.3569 |
| | LogRank | 0.9824 | **0.5116** | 0.9132 | **0.4382** | 0.5271 | 0.3568 |
| | LRR | 0.5583 | 0.3338 | 0.6446 | 0.3439 | 0.7028 | 0.3875 |
| | NPR | 0.9434 | 0.5346 | 0.5096 | 0.5000 | 0.7343 | 0.5096 |
| | DNA-GPT | 0.9918 | 0.5513 | 0.9788 | 0.5181 | 0.5942 | 0.3284 |
| | DetectGPT | 0.7913 | 0.5286 | 0.6529 | 0.4566 | 0.5790 | 0.5050 |
| | Fast-DetectGPT | 0.9917 | 0.7943 | 0.9197 | 0.5133 | 0.9546 | **0.4053** |
| | The Black-box Setting | | | | | | |
| EN→GER | Roberta-base | 0.4993 | 0.4934 | 0.5135 | 0.4001 | 0.4403 | 0.3984 |
| | Roberta-large | 0.4970 | 0.4496 | 0.4974 | 0.3958 | 0.4760 | 0.3946 |
| | Likelihood(Neo-2.7) | 0.9903 | **0.3170** | 0.4616 | 0.3392 | 0.3980 | 0.3480 |
| | LogRank(Neo-2.7) | 0.9824 | 0.3185 | 0.5023 | 0.3403 | 0.4064 | 0.3480 |
| | LRR(Neo-2.7) | 0.8099 | 0.3632 | 0.7743 | 0.3545 | 0.4954 | 0.3579 |
| | NPR(Neo-2.7) | 0.5762 | 0.3457 | 0.5351 | 0.3788 | 0.4649 | 0.3640 |
| | DNA-GPT(Neo-2.7) | 0.7324 | 0.3303 | 0.8468 | 0.3765 | 0.4822 | 0.3257 |
| | DetectGPT(T5-3B/Neo-2.7) | 0.4876 | 0.3394 | 0.4880 | 0.3773 | 0.4238 | 0.3560 |
| | Fast-DetectGPT(GPT-J/Neo-2.7) | 0.8955 | 0.3986 | 0.7648 | 0.3720 | 0.5299 | 0.3221 |
| | Binoculars(Falcon-7B) | 0.9951 | 0.3786 | 0.9922 | **0.3814** | 0.7244 | **0.3315** |

