# OpenReview forum: "Humanizing the Machine: Proxy Attacks to Mislead LLM Detectors"
_ICLR.cc/2025/Conference — ICLR 2025 Poster_

### Official Review · Reviewer_SF6Q · 2024-10-18

**Soundness:** 3
**Presentation:** 3
**Contribution:** 2
**Rating:** 6
**Confidence:** 4

**Summary:**

This paper introduces HUMPA (HUManized Proxy Attack), a novel strategy to evade detection of LLM-generated texts. The key idea is to use a small language model (SLM) fine-tuned with reinforcement learning in the decoding phase to contaminate the outputs of larger source models like Llama2-13B, Llama3-70B, and Mixtral-8x7B. The authors demonstrate that this approach can effectively deceive state-of-the-art detectors in both white-box and black-box settings, achieving significant drops in detection performance (up to 90% relative decrease in AUROC) while preserving generation quality. They also show the method's effectiveness in cross-domain and cross-language scenarios.

**Strengths:**

- The proposed HUMPA method is practical, addressing limitations of directly fine-tuning large models for evasion.
- This paper provides extensive theoretical analysis.
- Comprehensive experiments are conducted across multiple datasets, models, and detection methods, demonstrating the effectiveness of the attack.

**Weaknesses:**

- The HUMPA method itself is too straightforward and lacks novelty. Many previous works [1, 2] in other fields share similar ideas.
- Experiment metrics are not explained and hard to follow. Meanwhile, since the main purpose of applying SLM is to reduce time efficiency, you should also provide the time cost comparison between fine-tuning SLM and original LLM.





[1] Xu, Zhangchen, et al. "Safedecoding: Defending against jailbreak attacks via safety-aware decoding." arXiv preprint arXiv:2402.08983 (2024).

[2] Chen, Charlie, et al. "Accelerating large language model decoding with speculative sampling." arXiv preprint arXiv:2302.01318 (2023).

**Questions:**

- How sensitive is the HUMPA method to the choice of SLM used as the proxy attacker? Would using even smaller models such as Phi-3.5-mini-instruct still be effective?
- How does the computational cost of HUMPA compare to directly fine-tuning the source model, especially for the largest models tested?

---

> ### Author Response · Authors · 2024-11-24
> **Official response to reviewer SF6Q (part 1/2)**
>
> Dear reviewer SF6Q ,
>
> Thank you for taking the time to review our work and providing constructive feedback. In the following, we aim to address your questions and concerns.
>
> **1. (W1) The HUMPA method itself is too straightforward and lacks novelty. Many previous works [R1, R2] in other fields share similar ideas.**
>
> **Response:**  We highlight that our work introduces **a lightweight strategy to attack LLM by leveraging a humanized SLM that can bypass detectors**. This simple yet effective strategy involves 1) obtaining humanized SLM by DPO fine-tuning, and 2)  attacking the LLM's next-token output distribution by the subtle SLM's logit offset for each token probability in the decoding phase.  We find that many detectors are vulnerable, and HUMPA bypasses the detectors while preserving the generation utility.
>
> In the updated version, we have included additional related works in Appendix B. The work in [R1] makes contribution in safety-aware decoding strategy to defend against LLM jailbreaks, and [R2]  innovatively uses speculative sampling to transformer decoding to accelerate LLMs.  While they share similar vision in the using an auxiliary model in decoding-time realignment, we need to claim that our method aims to fine-tune an SLM towards optimal reward until it reaches the same level of reward for the human process according to a scoring detector, and adapt the LLM to achieves the same expected reward. We show **in terms of detection evasion,** **our attack strategy at decoding time achieves the same effect as directly attack the source model by DPO fine-tuning**. We believe it is a theoretical contribution in the field of proxy fine-tuning an LLM, and provide insights into effortlessly bypassing the LLM detectors.
>
> While we agree that the primary aim of our work is not to introduce a novel method for decoding-time realignment, we notice that the ICLR 2025 call for papers explicitly lists "alignment, fairness, safety, privacy, and societal considerations" as topics of interest. We believe our work aligns closely with this category, addressing important challenges in the safety and robustness of AI systems.
>
>
>
> [R1] Xu, Zhangchen, et al. "Safedecoding: Defending against jailbreak attacks via safety-aware decoding." arXiv preprint arXiv:2402.08983 (2024).
>
> [R2] Chen, Charlie, et al. "Accelerating large language model decoding with speculative sampling." arXiv preprint arXiv:2302.01318 (2023).
>
>
>
> **2. (W2) Experiment metrics are not explained and hard to follow.**
>
> **Response:**  Thanks for the constructive comment, it helps improve our paper.  Following prior works [R1], we employ the Area Under the Receiver Operating Characteristic Curve (AUROC) and the Area Under the Precision-Recall Curve (AUPRC) as primary metrics to evaluate the performance of each detector. To assess the utility and quality of the generated text, we utilize BERTScore and ROUGE-1/2/L metrics. We provide a detailed explanation of these metrics. To make it more clear, we have included a detailed explanation of these metrics in Appendix C of the updated version.
>
>
>
> [R1] Bao, G., Zhao, Y., Teng, Z., Yang, L., & Zhang, Y. Fast-DetectGPT: Efficient Zero-Shot Detection of Machine-Generated Text via Conditional Probability Curvature. In *ICLR 2024*.
>
> **3. (W2 & Q2) Since the main purpose of applying SLM is to reduce time efficiency, you should also provide the time cost comparison between fine-tuning SLM and original LLM. How does the computational cost of HUMPA compare to directly fine-tuning the source model, especially for the largest models tested?**
>
> **Response:**  Thanks for the constructive suggestion. Fine-tuning the model with DPO requires a preference dataset, which is generated by sampling response pairs $(y_1, y_2)$ from the model. To directly fine-tune the source model, preference pairs are sampled from the source model itself. In contrast, HUMPA samples pairs from the SLM, as the SLM is the model that needs to be fine-tuned (refer to Section 3.3 for detail). Therefore, we compare the running time of HUMPA with prior work that samples pairs from the source model and apply DPO fine-tuning on the source model.

---

> ### Author Response · Authors · 2024-11-24
> **Official response to reviewer SF6Q (part 2/2)**
>
> To compare the fine-tuning time, we set the DPO batch size to 8 and epoch to 5. We utilized LoRA for efficient fine-tuning. We also compare the inference time of text generation from HUMPA attacked LLM and that from the unattacked LLM. The inference time is of a single pass with a batch size of 1. The results are as follows:
>
> |                     | Llama2-13B           | Llama3-70B           | Mixtral-8x7B            |
> | ------------------- | -------------------- | -------------------- | ----------------------- |
> | Sampling (Hours)    | 18.61                | 41.64                | 63.52                   |
> | Fine-tuning (Hours) | 3.09                 | 9.54                 | 13.58                   |
> | Inference (Seconds) | 12.88                | 27.87                | 32.51                   |
> |                     | **HUMPA(Llama2-7B)** | **HUMPA(Llama3-8B)** | **HUMPA(Mistral-7B)** |
> | Sampling (Hours)    | 13.87                | 10.97                | 10.83                   |
> | Fine-tuning (Hours) | 2.04                 | 3.20                 | 1.39                    |
> | Inference (Seconds) | 39.87                | 47.22                | 36.75                   |
>
> We find that HUMPA is much more efficient than directly DPO fine-tuning attack the source model. The results indicate the efficiency of HUMPA. The inference time of HUMPA is slightly slower than that of the source model, but the sacrifice is not significant. We have included a section of "Efficiency" in the updated version.
>
> **4. (Q1) How sensitive is the HUMPA method to the choice of SLM used as the proxy attacker? Would using even smaller models such as Phi-3.5-mini-instruct still be effective?**
>
> **Response:**  Thanks for the constructive suggestion. We fine-tuned a Llama2-13B, and a TinyLlama-1.1B[R1] as the SLM respectively. We use Roberta-large as the scoring detector and evaluate on the Roberta-base on the Writing dataset, the attack ratio $\alpha$ is set to 1.5. The results are as follows:
>
> |       | Llama2-13B | HUMPA(Llama2-13B) | HUMPA(Llama2-7B) | HUMPA(TinyLlama-1.1B) |
> | ----- | ---------- | ----------------- | ---------------- | --------------------- |
> | AUROC | 0.9673     | 0.4176            | 0.4594           | 0.7706                |
>
> We find that different SLMs exhibit varying levels of effectiveness when attacking the LLM, with Llama2-13B being the most effective choice. This suggests that when selecting an SLM for such attacks, we need to consider its capability; stronger SLMs tend to demonstrate better attack performance.
>
> To extend the HUMPA attack to a source LLM using an SLM from a different model family, we still need to obtain a humanized SLM, begin by sampling response pairs $(y_s, y_s^{'})$ from the SLM given prompt $x$, and assigning preference labels using a scoring detector to create a preference dataset. Then, we fine-tune the SLM with DPO to obtain a humanized SLM (as detailed in Section 3.3 of our paper).
>
> To address scenarios where the LLM and SLM come from different model families with dissimilar vocabularies, we adopt the technique described in [R2].In this approach, we keep the humanized SLM frozen as the reward model while fine-tuning the source LLM. During the fine-tuning process of the source LLM, we sample response pairs $(y_{l}, y_l^{'})$  from the LLM, which are used as inputs for both the target LLM and the reward SLM. We perform DPO fine-tuning on Phi-3.5-mini-instruct using the scoring detector Roberta-large. We then apply this reward model to guide the fine-tuning [R2] of the source model Llama2-13B for 3 epochs.  The performance of the unattacked source Llama2-13B and the attacked Llama2-13B by humanized Phi-3.5-mini, tested using Roberta-base on the Writing dataset, is as follows:
>
> |       | Llama2-13B | HUMPA(Phi-3.5-mini) |
> | ----- | ---------- | ------------------- |
> | AUROC | 0.9673     | 0.8347              |
>
> We find that the source model can be attacked by a humanized SLM from a different family.  However, since this technique requires fine-tuning both the source LLM and the SLM, we did not include it in our paper. We leave systematic explorations on attacking LLMs from different model families without fine-tuning for future work to maintain our focus on the fragility of zero-shot detectors, the ease of efficiently attacking an LLM by fine-tuning only a small model, the equivalence between attacking the source LLM and attacking by an humanized SLM, and the preservation of the attacked LLM's generation utility.
>
>
>
> [R1] Zhang, P., Zeng, G., Wang, T., & Lu, W. (2024). Tinyllama: An open-source small language model. *arXiv preprint arXiv:2401.02385*.
>
> [R2] Gao, Z., Chang, J. D., Zhan, W., Oertell, O., Swamy, G., Brantley, K., ... & Sun, W. (2024). Rebel: Reinforcement learning via regressing relative rewards. *arXiv preprint arXiv:2404.16767*.

---

> > ### Comment · Reviewer_SF6Q · 2024-11-25
> > **Official Comments from reviewer SF6Q**
> >
> > Thanks for answering all my questions.

---

### Official Review · Reviewer_wsFK · 2024-10-31

**Soundness:** 2
**Presentation:** 2
**Contribution:** 2
**Rating:** 5
**Confidence:** 4

**Summary:**

This paper aims to develop an attack against LLM detectors. The authors present a method called HUMPA (humanized proxy attack) that uses a smaller language model to subtly alter the output of a large language model (LLM) in a way that makes it harder for detectors to identify as AI-generated. In particular, the work adopts conventional Preference-based reinforcement learning, using the LLM detectors as a reward model, to train an SLM and adds the SLM token prediction onto the original LLM toke predictions. The paper provides some theoretical analyses as well as experimental results. However, the method is not novel, and the experiments lack baseline comparisons.

**Strengths:**

- Leveraging SLM to reduce costs is a good idea
- The format is correct, and the images are clear
- Attacks and defenses for LLM-produced texts are important research directions.

**Weaknesses:**

- **No baselines**: the work only compares the result between the original LLM and the original LLM with the proposed HUMPA added on top. As shown in the Detection Evasion Methods in the Related Works section, many baselines should be compared.
- Unclear method and problem setting: there are many questions because the method and setting are not clearly explained.
    - attacker capability: does the attacker know the target LLM detector? Can the attacker query the target detector?
> It is worth noting that under adversarial attack conventions, black-box denotes query access while white-box denotes access to internal weights and gradients. In this work, the authors seem to denote the white box (detector) as the detector's access to LLM logits, while the black box indicates no access to logits and directly detects based on the final produced text (However, it is not clearly explained in the paper). Here, the reviewer wants to ask for the attacker's access to the detector.
    - Is the HUMPA method required to know which detector it faces beforehand? (also see the first question)?
> The work seems to use the target LLM detection directly as the reward model for each attack. Specifically, no cross-detector experiment setting was provided. Reviewing the code, the DPO_MixBuilder class also conditions based on the target detector. Thus, it seems that generalizability is an issue. Can a HUMPA method leveraging a target LLM detector be applied to a different type of LLM detector?
    - The target LLM task required for HUMPA to work. Can HUMPA generalize to different text generation tasks?
    - What is the threshold for evading LLM detection? Line 453 claims that HUMPA successfully evaded all detection methods across all source models. However, in Table 2 many detection scores are higher than 0.5.
- Poor writing, many typos or errors e.g. (not exhaustive):
    - Line 95 extra space
   -  Line 375 Then',' based on the samples,

**Questions:**

- (please see weaknesses)
- What methodological improvement does this work offer over previous approaches to evading LLM detection?
- While fine-tuning SLM is faster than LLM, Low-Rank Adaptation can be even faster. Why is LORA not considered?

---

> ### Author Response · Authors · 2024-11-24
> **Official response to reviewer wsFK (part 1/3)**
>
> Dear reviewer wsFK ,
>
> Thank you for taking the time to review our work and providing detailed feedback. In the following, we aim to address your questions and concerns.
>
> **1. (W1,Q1) No baselines: the work only compares the result between the original LLM and the original LLM with the proposed HUMPA added on top. As shown in the Detection Evasion Methods in the Related Works section, many baselines should be compared. What methodological improvement does this work offer over previous approaches to evading LLM detection?**
>
> **Response:** Thanks for the constructive suggestion, it helps to improve our paper. In Appendix G (Appendix H of revised paper), we evaluated our approach against the paraphrasing attack method DIPPER [R1] by comparing both the AUROC and the Fluency Win Rate. To show the effectiveness of detection evasion, we compare our method with two state-of-the-art paraphrasing evasion techniques: DIPPER Paraphrasing [R1] and Query-based Substitutions [R2]. Their performances on the WritingPrompts dataset are presented as follows.
>
> |                           | Likelihood | LogRank    | LRR        | NPR        | DNA-GPT    | DetectGPT  | Fast-DetectGPT | Binoculars |
> | ------------------------- | ---------- | ---------- | ---------- | ---------- | ---------- | ---------- | -------------- | ---------- |
> | Llama2-13B   | 0.9923     | 0.9945     | 0.9845     | 0.9670     | 0.7534     | 0.8713     | 0.9949         | 0.9969     |
> | Dipper Paraphrasing       | 0.8125     | 0.7998     | 0.7220     | 0.5193     | 0.6240     | 0.2675     | 0.9754         | 0.9398     |
> | Query-based Substitutions | 0.9843     | 0.9921     | 0.9828     | 0.3030     | 0.7072     | 0.1914     | 0.9972         | 1.0000     |
> | HUMPA ($\alpha=1.2$)      | 0.1647     | 0.1625     | 0.1599     | 0.2592     | 0.0723     | 0.2124     | 0.0794         | 0.1743     |
> | HUMPA ($\alpha=1.5$)      | **0.0109** | **0.0109** | **0.0117** | **0.0617** | **0.0034** | **0.0582** | **0.0007**     | **0.0021** |
>
> We find that our method with $\alpha = 1.5$, achieves the best detection evasion performance, aligning with our analysis of $\alpha$ in Section 3.3.  We have included the results in Table 8 of the updated paper.
>
> The state-of-the-art fine-tuning approach is directly DPO fine-tuning the source model [R3] to evade detection. However, this approach is time-consuming when DPO fine-tuning a large model (e.g., 70B), and it is difficult to balance maintaining high-quality text generation with effective detection evasion. We claim our strategy is more efficient than that, and the utility of text generation is preserved. Therefore,  in the "Utility Preserving" section (Section 4.4), we compare our method with theirs in terms of utility preservation. We have included an "Efficiency" section in the updated version, to show the efficiency of our approach compared to this method.
>
>
>
> [R1] Krishna, K., Song, Y., Karpinska, M., Wieting, J., & Iyyer, M. (2024). Paraphrasing evades detectors of ai-generated text, but retrieval is an effective defense. *Advances in Neural Information Processing Systems*, *36*.
>
> [R2] Shi, Z., Wang, Y., Yin, F., Chen, X., Chang, K. W., & Hsieh, C. J. (2024). Red teaming language model detectors with language models. *Transactions of the Association for Computational Linguistics*, *12*, 174-189.
>
> [R3]. Nicks, C., Mitchell, E., Rafailov, R., Sharma, A., Manning, C. D., Finn, C., & Ermon, S. (2023). Language model detectors are easily optimized against. In ICLR 2024.
>
> **2. (W2.1) Attacker capability: does the attacker know the target LLM detector? Can the attacker query the target detector? Here, the reviewer wants to ask for the attacker's access to the detector.**
>
> **Response:** Thank you for the great question. In our work, we do not assume that the attacker needs to know the target LLM detector. To obtain the humanized SLM, the attacker performs DPO fine-tuning, which requires creating the preference dataset $\mathcal{D} = \{ (x, y^w, y^l) \}$. A scoring detector is needed to assign preference labels. Specifically, for each prompt $x$, we sample response pairs $(y_1, y_2)$ using the SLM, and determine preference labels by comparing the scoring detector's human-ness scores $s(x, y)$ on the responses: if $y_1 \succ y_2$, we assign $y^w = y_1$ and $y^l = y_2$; otherwise, we assign $y^w = y_2$ and $y^l = y_1$ (more details can be found in Section 3.3). In this process, the scoring detector does not need to be the target LLM detector. In our experiments, we use Fast-DetectGPT as the scoring detector because it is fast and accurate. After obtaining the fine-tuned SLM, we evaluate various detectors on texts generated by the attacked LLM. As stated in Section 4.2, "*In our experiments, we use Fast-DetectGPT for scoring*."

---

> ### Author Response · Authors · 2024-11-24
> **Official response to reviewer wsFK (part 2/3)**
>
> **3. (W2.1) In this work, the authors seem to denote the white box (detector) as the detector's access to LLM logits, while the black box indicates no access to logits and directly detects based on the final produced text (However, it is not clearly explained in the paper).**
>
> **Response:**  We thank the reviewer for pointing out this issue. However, it is worth noting that most of zero-shot detectors are usually composed of a surrogate model (also termed as 'scoring model' [R2])  and the source model. The surrogate model is used to compute AI-like scores for the text generated by the source model [R2, R3]. In Fast-DetectGPT [R2], it consists of a sampling model and a scoring model.
>
> In the field of machine-generated text detection, white-box denote the surrogate model know the source model, and black-box denote the surrogate model does not know the source model [R1, R2, R3]. In our work, we followed the existing definition and described in the section of "The Settings" (Section 4.1) that "*We evaluate the zero-shot methods in two settings, the white-box (source model is known) setting and black-box (source model is unknown) setting*".  Accordingly, we conducted experiments in both settings. Following [R2], in the white-box setting, we set the surrogate model in each detector to be identical to the source model, whereas in the black-box setting, the surrogate model differs from the source model.
>
> This concept is different from the 'scoring detector' mentioned in our method in Section 3.3. To avoid any confusion, we have updated the terminology in our revised version by replacing the term 'scoring' model with 'surrogate' model when describing the detectors. Besides, we have included a more clear description of the white- and black-box settings
>
>
>
> [R1] Yang, X., Pan, L., Zhao, X., Chen, H., Petzold, L., Wang, W. Y., & Cheng, W. (2023). A survey on detection of llms-generated content. *arXiv preprint arXiv:2310.15654*.
>
> [R2] Bao, G., Zhao, Y., Teng, Z., Yang, L., & Zhang, Y. Fast-DetectGPT: Efficient Zero-Shot Detection of Machine-Generated Text via Conditional Probability Curvature. In *ICLR 2024*.
>
> [R3] Yang, X., Cheng, W., Wu, Y., Petzold, L. R., Wang, W. Y., & Chen, H. DNA-GPT: Divergent N-Gram Analysis for Training-Free Detection of GPT-Generated Text. In *ICLR 2024*.
>
>
>
> **4. (W2.2) Is the HUMPA method required to know which detector it faces beforehand? The work seems to use the target LLM detection directly as the reward model for each attack. Specifically, no cross-detector experiment setting was provided. Reviewing the code, the DPO_MixBuilder class also conditions based on the target detector. Thus, it seems that generalizability is an issue. Can a HUMPA method leveraging a target LLM detector be applied to a different type of LLM detector?**
>
> **Response:**  Thank you for the question. As mentioned in our previous response (**Response 2**), we do not assume that the attacker requires knowledge of the target LLM detector. In our experiments, we use Fast-DetectGPT as the scoring detector primarily due to its speed and accuracy. Some perturbation-based detectors are time-consuming and therefore not ideal choices for scoring. By using Fast-DetectGPT as the scoring detector to obtain the fine-tuned SLM, we evaluate all other detectors, with the results presented in Tables 1, 2, 3, 5, 6, 7, 10, and 11 (Tables 1, 2, 4, 5, 6, 7, 8, 10 and 11 in the updated paper).
>
> In the code, the DPO_MixBuilder class includes more options for scoring detectors. We aim to provide these as interfaces for users who are interested in following our work. A clearer README file and detailed user guidance will be provided when we officially release the code.
>
>
>
> **5. (W2.3) The target LLM task required for HUMPA to work. Can HUMPA generalize to different text generation tasks?**
>
> **Response:**  We thank the reviewer for constructive feedback. When tuning the SLM, the target task is not assumed to known. We tested HUMPA in a cross-task scenario: we fine-tuned the SLM Llama2-7B on the text completion task using the WritingPrompts dataset and attacked the Llama2-13B on the question-answering task using the PubMedQA dataset. We use Roberta-large for the scoring detector, and evaluate the AUROC of Roberta-base. The results are as follows
>
> |                            | Llama2-13B | HUMPA($\alpha=1.5$) | HUMPA($\alpha=2.0$) |
> | -------------------------- | ----------------------- | ------------------- | ------------------- |
> | Writing$\rightarrow$PubMed | 0.6820                  | 0.3965              | 0.2813              |
>
> We find that HUMPA successfully attacks the LLM in this scenario, resulting in relative drops of 41.9% and 43.6%, respectively, for $\alpha=1.5$ and $\alpha=2.0$.

---

> ### Author Response · Authors · 2024-11-24
> **Official response to reviewer wsFK (part 3/3)**
>
> **6. (W2.4) What is the threshold for evading LLM detection?**
>
> **Response:**  Thank you for the great question. The Area Under the ROC Curve (AUROC) is calculated as the area under the ROC curve, quantifying the overall ability of a classifier to distinguish between the two classes. The ROC curve is generated by calculating the True Positive Rate (TPR) and False Positive Rate (FPR) at every possible threshold. This means it considers the full spectrum of threshold values from 0 to 1. Crucially, AUROC is independent of the specific decision threshold. Following prior work [R1], we did not manually set a threshold when calculating the AUROC.
>
> It should be noted that in our theoretical analysis (Appendix A.1),  we bring up the concept of a threshold to hypothesize that our scoring detector assigns labels to texts based on an implicit reward and a corresponding threshold. This concept is purely a theoretical construct and is not related to our evaluation process.
>
>
>
>  [R1]. Bao, G., Zhao, Y., Teng, Z., Yang, L., & Zhang, Y. Fast-DetectGPT: Efficient Zero-Shot Detection of Machine-Generated Text via Conditional Probability Curvature. In *ICLR 2024*.
>
>
>
> **7. (W2.4) Line 453 claims that HUMPA successfully evaded all detection methods across all source models. However, in Table 2 many detection scores are higher than 0.5.**
>
> **Response:**  Thank you for the fair comment. We need to clarify that the performances in Table 2 does **not** represent the best detection evasion performance achievable by HUMPA. Instead, this table shows the AUROC performance of detectors when the utilities of text generated from HUMPA attacked LLM and the source LLM are constrained within a **manually set** budget of $\Delta S_{Bert}\leq 0.02$ and $\Delta \text{ROUGE-1} \leq 0.03$. We aim to show **within this budget**, how HUMPA performs in detection evasion, and how fragile for different detectors when attacked by HUMPA.
>
> As we state in Section 3.3, the ratio $\alpha$ controls the intensity of the attack on the LLM, with larger values of $\alpha$ yielding higher rewards and better detection evasion, while smaller values keep the attacked LLM closer to the source LLM. Therefore, **the effectiveness of detection evasion can be controlled by $\alpha$**.  For example, when evaluating the white-box detectors Likelihood, DetectGPT, and Fast-DetectGPT on texts generated by HUMPA attacked Llama2-13B for the WritingPrompts dataset, the AUROC of the detectors decrease as $\alpha$ increases
>
> |                | Llama2-13B | HUMPA($\alpha=1.0$) | HUMPA($\alpha=1.2$) | HUMPA($\alpha=1.5$) | HUMPA($\alpha=2.0$) |
> | -------------- | ---------- | ------------------- | ------------------- | ------------------- | ------------------- |
> | Likilihood     | 0.9999     | 0.5933              | 0.2846              | 0.0206              | 0.0000              |
> | DetectGPT      | 0.8795     | 0.7096              | 0.4422              | 0.2545              | 0.2153              |
> | Fast-DetectGPT | 0.9999     | 0.6727              | 0.5184              | 0.4901              | 0.3939              |
>
> We demonstrated this effect in the "Analysis of $\alpha$" section (Section 4.4) and discussed it following Theorem 3.2, noting that *a larger $\alpha$ leads to higher rewards and improved detection evasion*. To address any confusion, we have added further details and analysis in the "More Analysis of $\alpha$ " section in Appendix G of the updated paper.
>
>
>
> **8. (W3) Poor writing, many typos or errors**
>
> **Response:**  Thank you for your careful review and suggestions. We have corrected the typos and errors in the revised version of the paper.
>
>
>
> **9 (Q2). Why is LORA not considered?**
>
> **Response:**  Thanks for the great question. Throughout our experiments, we utilized LoRA for fine-tuning. To clarify this in the paper, we have included the following statement in the "Implementation Details" section (in Section 4.1): *"We apply Low-Rank Adaptation (LoRA) to fine-tune the SLM."* Additionally, we have added a section of "Efficiency" (Section 4.4), where we compare the runtime of HUMPA with directly attacking the source model by DPO fine-tuning from prior work [R1]. Additionally, in Table 3 of the revised paper, we have updated the description of the table to explicitly state: *"'Fine-tuning' represents DPO fine-tuning using LoRA."*
>
>
>
> [R1]. Nicks, C., Mitchell, E., Rafailov, R., Sharma, A., Manning, C. D., Finn, C., & Ermon, S. (2023). Language model detectors are easily optimized against. In *ICLR 2024*.

---

> > ### Comment · Reviewer_wsFK · 2024-11-24
> >
> > Thanks. While I still have concerns about whether the relevant state-of-the-art baselines have been thoroughly evaluated, the authors have answered all of my questions, and I have raised my score.

---

> > > ### Comment · Reviewer_wsFK · 2024-11-26
> > >
> > > I notice that the authors have not answered the question
> > > - What methodological improvement does this work offer over previous approaches to evading LLM detection?
> > >
> > > Besides, I found other papers
> > > - Humanizing Machine-Generated Content: Evading AI-Text Detection through Adversarial Attack
> > > > It is only mentioned in the introduction section but not compared in the related works section.
> > > - Red teaming language model detectors with language models
> > > > It is only mentioned in the experiment section but not compared in the related works section.

---

> ### Author Response · Authors · 2024-11-26
> **Follow-up by authors**
>
> Dear Reviewer wsFK,
>
> Thanks for follow-up, we address the questions as follows
>
> **1. What methodological improvement does this work offer over previous approaches to evading LLM detection?**
>
> **Response:**  Thanks for the valuable question. In the field of detection evasion, most of previous approaches are paraphrasing methods, which directly modify the content of prompts or generated text. Each time the attack is launched, the prompts or generated text need to be processed by a paraphrasing model, limiting its long-term scalability.
>
> An alternative research line focuses on adapting the source model [R1], enabling it to generate texts that evade detection. The fine-tuning is performed once, after which the generative model consistently exhibits the malicious behavior. The prior work [R1] employs DPO fine-tuning to adapt the source model. However, this approach becomes time-intensive when dealing with sufficiently large source LLMs (e.g., 70B parameters). To address this limitation, we propose a lightweight method to adapt the source LLM, enabling it to generate text that bypasses detectors.  We formulate the detection evasion by assuming an implicit reward that a scoring detector bases its decision on. We aim to find a machine generative process by adapting the source LLM, such that a scoring detector is unable to distinguish the machine-generated texts from the human texts. To this end, our work fine tune an SLM towards optimal reward, aiming to reach the same level of reward for the human process according to a scoring detector. For detection evasion, the SLM was used to adapt the source LLM by shifting the logit distribution at the decoding phase to achieve the same expected reward as fine-tuning the LLM. Our empirical study demonstrates that our approach improves efficiency, enables the LLM to effectively evade detection, and preserves generation utility.
>
> [R1] Nicks, C., Mitchell, E., Rafailov, R., Sharma, A., Manning, C. D., Finn, C., & Ermon, S. (2023). Language model detectors are easily optimized against. In ICLR 2024.
>
> **2.The papers not compared in the related works section.**
>
> **Response:**  We thank the reviewer for constructive suggestions. In the revised version, we have included and compared the papers[R2,R3] in the related works section.
>
> [R2] Humanizing Machine-Generated Content: Evading AI-Text Detection through Adversarial Attack
>
> [R3] Red teaming language model detectors with language models

---

### Official Review · Reviewer_BPF1 · 2024-11-04

**Soundness:** 3
**Presentation:** 4
**Contribution:** 3
**Rating:** 8
**Confidence:** 4

**Summary:**

This paper presents a new attack to evade the MGT detectors by training a proxy to generate humanized text. The idea is to train a lightweight student model to screw the generation towards "human" through reinforcement learning by assigning "humanity" with higher rewards. The paper analyzes the theoretical guarantee of the proxy's attack effectiveness and generation quality. By evaluating several datasets, the paper shows the effectiveness and even some generalizability across languages and domains.

**Strengths:**

1. The paper is very nicely written and presented. The logic is clear, and the figures are beautiful. The problem is nicely defined, and the methodology is clearly explained.
2. The paper shows the potential of using lightweight methods to bypass MGT detection.
3. The code is open-sourced.
4. In practical experiments, the paper shows the effectiveness of the attack, especially with little harm to model utility, which is amazing.

**Weaknesses:**

1. My main concern is that the theoretical guarantees in the paper (specifically Theorems 3.1 and 3.2) may be misleading, potentially overclaiming what they actually achieve. The logic behind these theorems is pretty simple: the training objective aims to balance two things—making the student model mimic humans (for attack effectiveness) and making it mimic the teacher LLM (for quality). By adjusting β from 0 to +∞, the student LLM could theoretically be optimized (optimally) to behave either purely human-like or purely as a replica of the teacher model, given “reasonable” training data. However, there are several issues with this claim: Firstly, as mentioned in the paper, the detector will overfit to what it was exposed to, which is unlikely to represent perfectly and, more importantly, calibrate accurately to the human distribution. And, because the proposed method relies on the detector rather than true human distribution, there seems to be no way to get the lambda mentioned. In other words, the approach would only work under ideal optimization, an ideal detector (a perfectly calibrated one), and an ideal training dataset. This explains why the paper uses Fast-DetectGPT as the scoring detector in all white-box settings, why black-box settings show better attack performance (NEO might be stronger with better calibration), and why the student models selected are from the same family as the teacher. Secondly, Theorem 3.2 essentially shows that the student model is just a less optimized (in terms of utility) version of the teacher, which doesn’t necessarily make it "good quality".
2. Given the constraints on both the detector and the training data, the attack is very likely to be limited in scope. While the paper presents some interesting results suggesting generalizability, we should expect that it won’t extend broadly in practice. See 1.
3. The paper uses models that are very close in architecture and scale (e.g., Llama2-13B vs. Llama2-7B). It’s unclear whether a smaller model from a different family could yield similar attack results. See 1.
4. The paper relies on Fast-DetectGPT for scoring in every white-box experiment, which doesn’t seem appropriate or fully aligned with the paper’s methodology. I wonder if these attacks would still perform well using only the target detector or less well-calibrated detectors. See 1.
5. Many of the white-box attacks have low performance. See 1.
6. The paper does not compare with any other SOTA attacks.
7. Although it is cool to use a small model to improve efficiency, the paper doesn’t provide any assessment of this efficiency, particularly in comparison to tuning the source LLM with DPO.

**Questions:**

1. What if the detector's score does not accurately reflect the degree of "human"? Since this is an attack paper, it is expected that there are different kinds of victims. What if the detector does not give higher rewards to more human text?
2. What if the source LLM generated nearly equivalently "human" text with similar rewards? For the proposed attacks, is there any requirements on the scale of the reward difference? In other words, how different should the training sample be in terms of rewards?
3. Why are white-box attacks even worse?
4. How efficient is the attack compared with tuning source LLMs with DPO?
5. Can we use smaller models with different architecture to attack?
6. What if we stick to the same model, i.e., to use the scoring of LRR to attack LRR?
7. Is there any transferability across detectors? For example, can we use the scoring from LRR to attack DetectGPT or Fast-DetectGPT?
8. Can we apply such attacks in truly black-box settings, e.g., GPTZero or CopyLeaks?

Specifically, could you please:
1. Clarify the assumptions underlying Theorems 3.1 and 3.2, particularly regarding detector calibration and training data representativeness.
2. Discuss the practical implications of these assumptions not being fully met in real-world scenarios.
3. Provide additional empirical evidence or analysis to support the theoretical claims, especially regarding quality preservation in Theorem 3.2.
4. Discuss potential limitations on the generalizability of their approach more explicitly.
5. Suggest or conduct additional experiments that could test the boundaries of the attack's effectiveness across a wider range of scenarios or detectors.
6. Conduct experiments using smaller models from different architectural families as the proxy attacker.
7. Discuss the potential impact of model architecture and size differences on the attack's effectiveness.
8. Perform additional experiments using different detectors for scoring in the white-box setting.
9. Analyze and discuss how the choice of scoring detector impacts the attack's performance.
10. Include comparisons with other state-of-the-art attack methods on the same datasets and detectors.
11. Discuss the relative strengths and weaknesses of their approach compared to existing methods.
12. Provide quantitative comparisons of computational efficiency between their method and direct fine-tuning of the source LLM.
13. Discuss the trade-offs between efficiency gains and potential performance differences.

**Details Of Ethics Concerns:**

The paper proposes a new attack to evade MGT detectors, while MGT detectors might be used to mitigate all kinds of AIGC misusages like plagiarism, misinformation, and so on.

At the same time, the paper already includes a relevant ethics statement.

---

> ### Author Response · Authors · 2024-11-24
> **Official response to reviewer BPF1 (part 1/5)**
>
> Dear reviewer BPF1 ,
>
> Thank you for taking the time to review our work and providing detailed feedback. In the following, we aim to address your questions and concerns.
>
> **1. (W1, Q1, Q2, S1, S2) Please specify the assumptions underlying the theorems, and discuss the practical implications of these assumptions not being fully met in real-world scenarios, especially as pertaining to the following questions: What if the detector's score does not accurately reflect the human-ness of texts? What if the detector does not give higher rewards to more human text, or the source LLM generates texts with similar rewards as human texts? For the proposed attacks, are there any requirements on the scale of the reward difference?**
>
> **Response:**  We thank the reviewer for the great questions. First, as the reviewer correctly points out, it is natural to assume that a detector cannot fully reflect the human-ness of texts. Crucially, in this case human-generated texts **do not achieve optimal rewards** according to the detector. When trained according to such a detector, one cannot expect a proxy attacker to produce texts that completely resemble human texts. In fact, this is part of Assumption A.1 for Lemma (Theorem) 3.1:
>
> **Assumption 1.** The human text generation process $H$ does not achieve optimal reward according to the detector $D$.
>
> This gap signifies the effectiveness of the detector in that, generally, the smaller this gap is, the larger the reward and the more optimal human-generated texts are according to the detector, and hence the more aligned the detector is to human-generated texts. Note that a possible confusion here is that this assumption means human-like texts will receive a **smaller** reward than the best possible reward, which may be unintuitive. The second assumption we made is that:
>
> **Assumption 2.** The detector gives higher rewards to human texts than texts generated by pre-trained LLM, which is what an effective detector should be able to achieve.
>
> These two gaps (pre-trained LM \< human \< detector’s optimal) are all that is needed for our result to hold. Intuitively, by fine-tuning the LM with DPO, we can increase the expected reward of LM to a comparable level as humans, but not so much as to overfit to the detector and suffer a loss in quality. A contradiction to these assumptions would suggest, respectively, that the detector is perfectly aligned with human-ness of texts, or that the pre-trained LLM is already capable of evading the detector, nullifying the need for an attacker.
>
> The “optimal” $\beta$ in Lemma 3.1 achieves the exact same expected reward as the human-generated process, which is a sweet spot in theoretical analysis. However, as per our discussion after the lemma, in reality $\beta$ controls the trade-off between quality (similarity to teacher) and performance (ability to evade detection), which also effectively prevents overfitting to the detector.
>
> In light of this question, we have restated Assumption A.1 to emphasize the two expected reward gaps in the revised paper.
>
>
>
> **2. (W1, S3) Theorem 3.2 essentially shows that the student model is just a less optimized (in terms of utility) version of the teacher, which doesn’t necessarily make it "good quality". Provide additional empirical evidence or analysis to support the theoretical claims, especially regarding quality preservation in Theorem 3.2.**
>
> **Response:**  Thanks for the constructive suggestion. While it is difficult to quantify the quality of attacker-generated texts, note that as the attack ratio $\alpha$ goes from $0$ to $\infty$, the model $M’$ **gradually and continuously** shifts away from the reference teacher model $M$ with increased evasion performance, which means with a small $\alpha$ the quality will be largely preserved.
>
> Practically, we mainly demonstrate the quality of our attacker texts in Table 1, where the generation utilities of texts produced by HUMPA and the source model are within the budget of $\Delta S_{Bert}\leq 0.02$ and $\Delta \text{ROUGE-1} \leq 0.03$. This suggests that HUMPA is capable of detecting evasion without a significant loss in utility.
>
> **3. (Q1, S9) What happens to the attacker if the detector's score does not accurately reflect the human-ness of texts? Analyze and discuss how the choice of scoring detector impacts the attack's performance.**
>
> **Response:**  Thanks for the great question. Continuing on our discussion of Assumption 1, the weaker the detector is, the lower its reward is for human-generated texts, and the easier it is for an attacker to bypass the detector. However, since the scoring detector is often different from the target detector, this will result in a relatively weaker attacker according to the (stronger) target detector, which will decrease the attacker’s performance. We show this empirically in the **Response 6**, and include a section of "Analysis of Scoring Detector" in Appendix J.

---

> ### Author Response · Authors · 2024-11-24
> **Official response to reviewer BPF1 (part 2/5)**
>
> **4. (W2, S4) Given the constraints on both the detector and the training data, the attack is very likely to be limited in scope. While the paper presents some interesting results suggesting generalizability, we should expect that it won’t extend broadly in practice. Discuss potential limitations on the generalizability of the approach more explicitly.**
>
> **Response:**  Thanks for the suggestion. Theoretically, a weak detector yields relatively low reward for human-generated texts, while machine-generated texts are given relatively high reward, resulting in a reduced gap between the two. Hence an attacker can more easily bypass the detector and overfit to the weak scoring detector. Therefore at deployment, when faced with a strong target detector, the attacker's performance will suffer. Empirically, we have found that using LRR as the scoring detector does not yield a comparable relative drop in AUROC compared to using Fast-DetectGPT, as shown in Appendix J: Analysis of Scoring Detector. To summarize, our attackers trained on a weak scoring detector cannot perform well in the face of a strong target detector, though we surmise this gap may be simply due to the lack of high-quality human-machine labels without access to strong scoring detectors and may hence be unavoidable.
>
> **5. (W3, Q5, S6, S7) The paper uses models that are very close in architecture and scale (e.g., Llama2-13B vs. Llama2-7B). It’s unclear whether a smaller model from a different family could yield similar attack results. Can we use smaller models with different architecture to attack? Conduct experiments using smaller models from different architectural families as the proxy attacker.Discuss the potential impact of model architecture and size differences on the attack's effectiveness**
>
> **Response:**  Thanks for the insightful comments. In our work, we primarily consider the LLM and SLM share the same vocabulary. To study the sensitivity of model size, we fine-tuned a Llama2-13B, a Llama2-7B and a TinyLlama-1.1B [R1] as the SLM respectively. We use Roberta-large as the scoring detector and evaluate on the Roberta-base on the Writing dataset, the attack ratio $\alpha$ is set to 1.5. The results are as follows:
>
> |       | Llama2-13B | HUMPA(Llama2-13B) | HUMPA(Llama2-7B) | HUMPA(TinyLlama-1.1B) |
> | ----- | ---------- | ----------------- | ---------------- | --------------------- |
> | AUROC | 0.9673     | 0.4176            | 0.4594           | 0.7706                |
>
> We find that different SLMs exhibit varying levels of effectiveness when attacking the LLM, with Llama2-13B being the most effective choice. This suggests that when selecting an SLM for such attacks, we need to consider its capability; stronger SLMs tend to demonstrate better evasion performance.
>
> In the decoding-time of next-token sampling, the logarithm adding between the encoded logits from the LLM and SLM (in Equation 5) requires the LLM and the SLM share the same tokenizer. Therefore, we only consider the same family of architectures which share the same vocabulary. To address scenarios where the LLM and SLM come from different model families with dissimilar vocabularies, we can adopt the technique described in [R2]. In this approach, we keep the humanized SLM frozen as the reward model while fine-tuning the source LLM. During the fine-tuning process of the source LLM, we sample response pairs $(y_l, y_l^{'})$  from the LLM, which are used as inputs for both the target LLM and the reward SLM. We perform DPO fine-tuning on Phi-3.5-mini-instruct using the scoring detector Roberta-large. We then apply this reward model to guide the fine-tuning [R2] of the source model Llama2-13B for 3 epochs.  The performance of the unattacked source Llama2-13B and the attacked Llama2-13B by humanized Phi-3.5-mini, tested using Roberta-base on the Writing dataset, is as follows:
>
> |       | Llama2-13B | HUMPA(Phi-3.5-mini) |
> | ----- | ---------------------- | ------------------- |
> | AUROC | 0.9673                 | 0.8347              |
>
> We find that the source model can be attacked by a humanized SLM from a different family.  However, since this technique requires fine-tuning both the source LLM and the SLM, we did not include it in our paper. We leave systematic explorations on attacking LLMs from different model families without fine-tuning for future work to maintain our focus on the fragility of zero-shot detectors, the ease of efficiently attacking an LLM by fine-tuning only a small model, the equivalence between attacking the source LLM and attacking by an humanized SLM, and the preservation of generation utility.
>
>
>
> [R1] Zhang, P., Zeng, G., Wang, T., & Lu, W. (2024). Tinyllama: An open-source small language model. *arXiv preprint arXiv:2401.02385*.
>
> [R2] Gao, Z., Chang, J. D., Zhan, W., Oertell, O., Swamy, G., Brantley, K., ... & Sun, W. (2024). Rebel: Reinforcement learning via regressing relative rewards. *arXiv preprint arXiv:2404.16767*.

---

> ### Author Response · Authors · 2024-11-24
> **Official response to reviewer BPF1 (part 3/5)**
>
> **6. (W4, Q6, Q7, S8) The paper relies on Fast-DetectGPT for scoring in every white-box experiment, which doesn’t seem appropriate or fully aligned with the paper’s methodology. Will these attacks still perform well using only the target detector or less well-calibrated detectors? What if we stick to the same model, i.e., to use the scoring of LRR to attack LRR? Is there any transferability across detectors? For example, can we use the scoring from LRR to attack DetectGPT or Fast-DetectGPT? Perform additional experiments using different detectors for scoring in the white-box setting.**
>
> **Response:**  Thanks for raising this thoughtful question. We use the white-box LRR as the scoring detector to fine-tune a Llama2-7B model and evaluate the AUROC performance of LRR, DetectGPT, and Fast-DetectGPT in the white-box on texts generated by the attacked model under varying levels of $\alpha$.The results are as follows:
>
> |                | Llama2-13B | HUMPA($\alpha=1.0$) | HUMPA($\alpha=1.5$) | HUMPA($\alpha=2.0$) |
> | -------------- | ----------------------- | ------------------- | ------------------- | ------------------- |
> | LRR            | 0.9623                  | 0.9611              | 0.9288              | 0.9270              |
> | Fast-DetectGPT | 0.9999                  | 0.9882              | 0.9204              | 0.8064              |
>
> If we use Fast-DetectGPT as the scoring detector, the results are
>
> |                | Llama2-13B | HUMPA($\alpha=1.0$) | HUMPA($\alpha=1.5$) | HUMPA($\alpha=2.0$) |
> | -------------- | ----------------------- | ------------------- | ------------------- | ------------------- |
> | LRR            | 0.9623                  | 0.5297              | 0.0307              | 0.0000              |
> | Fast-DetectGPT | 0.9999                  | 0.5184              | 0.4901              | 0.3939              |
>
> We find that when LRR is used as the scoring detector, the attacked Llama2-13B model exhibits a relative drop of 3.67% on LRR and a 19.35% drop on Fast-DetectGPT even when $\alpha=2.0$. In contrast, when Fast-DetectGPT is used for scoring, the relative drop increases to 100% on LRR and 60.61% on Fast-DetectGPT for $\alpha=2.0$
>
> While our paper does not assume which specific scoring detector the attacker must choose, adversaries can opt for any well-calibrated detector for this purpose. To address this issue, we have included a section of "Analysis of Scoring Detector" in Appendix J of the updated paper.
>
>
>
> **7. (W5, Q3, W1) Why do many of the white-box attacks have low performance? [also mentioned in W1:]…This explains why the paper uses Fast-DetectGPT as the scoring detector in all white-box settings, why black-box settings show better attack performance (NEO might be stronger with better calibration).**
>
> **Response:**  Thanks for the insightful questions. Many white-box detectors are less fragile than their black-box counterparts. This may be because, in the white-box setting, the surrogate model is assumed to be identical to the source model. In other words, the surrogate model has more knowledge of the source model compared to the black-box setting, making the detectors more robust against attacks than in the black-box scenario.  When selecting the scoring detectors, we prioritized Fast-DetectGPT due to its speed and accuracy, allowing it to efficiently score hundreds of text pairs.

---

> ### Author Response · Authors · 2024-11-24
> **Official response to reviewer BPF1 (part 4/5)**
>
> **8. (W6, S10) Include comparisons with other state-of-the-art attack methods on the same datasets and detectors.**
>
> **Response:**  We thank the reviewer for the constructive suggestion. To show the effectiveness of detection evasion, we compare our method with two state-of-the-art paraphrasing evasion techniques: DIPPER Paraphrasing [R1] and Query-based Substitutions [R2]. Their performances on the WritingPrompts dataset are presented as follows.
>
> |                           | Likelihood | LogRank    | LRR        | NPR        | DNA-GPT    | DetectGPT  | Fast-DetectGPT | Binoculars | $S_{Bert}$ | ROUGE-1/2/L                                |
> | ------------------------- | ---------- | ---------- | ---------- | ---------- | ---------- | ---------- | -------------- | ---------- | ---------- | ------------------------------------------ |
> | Llama2-13B                | 0.9923     | 0.9945     | 0.9845     | 0.9670     | 0.7534     | 0.8713     | 0.9949         | 0.9969     | 0.8189     | 0.2587$\mathbf{/}$0.0480$\mathbf{/}$0.1497 |
> | Dipper Paraphrasing       | 0.8125     | 0.7998     | 0.7220     | 0.5193     | 0.6240     | 0.2675     | 0.9754         | 0.9398     | 0.8006     | 0.2076$\mathbf{/}$0.0226$\mathbf{/}$0.1191 |
> | Query-based Substitutions | 0.9843     | 0.9921     | 0.9828     | 0.3030     | 0.7072     | 0.1914     | 0.9972         | 1.0000     | 0.7989     | 0.2015$\mathbf{/}$0.0383$\mathbf{/}$0.1256 |
> | HUMPA ($\alpha=1.2$)      | 0.1647     | 0.1625     | 0.1599     | 0.2592     | 0.0723     | 0.2124     | 0.0794         | 0.1743     | 0.8053     | 0.2281$\mathbf{/}$0.0422$\mathbf{/}$0.1409 |
> | HUMPA ($\alpha=1.5$)      | **0.0109** | **0.0109** | **0.0117** | **0.0617** | **0.0034** | **0.0582** | **0.0007**     | **0.0021** | 0.8014     | 0.2137$\mathbf{/}$0.0404$\mathbf{/}$0.1383 |
>
>
>
> We find that our method with $\alpha = 1.5$, achieves the best detection evasion performance, aligning with our analysis of $\alpha$ in Section 3.3.  We have included the results in Table 8 of the updated paper.
>
>
>
> [R1] Krishna, K., Song, Y., Karpinska, M., Wieting, J., & Iyyer, M. (2024). Paraphrasing evades detectors of ai-generated text, but retrieval is an effective defense. *Advances in Neural Information Processing Systems*, *36*.
>
> [R2] Shi, Z., Wang, Y., Yin, F., Chen, X., Chang, K. W., & Hsieh, C. J. (2024). Red teaming language model detectors with language models. *Transactions of the Association for Computational Linguistics*, *12*, 174-189.
>
>
> **9. (W7, Q4, S12) Provide assessment of the efficiency of using an SLM, particularly in comparison to tuning the source LLM with DPO. How efficient is the attack compared with tuning source LLMs with DPO? Provide quantitative comparisons of computational efficiency between their method and direct fine-tuning of the source LLM.**
>
> **Response:**  Thanks for the constructive suggestion. Fine-tuning the model with DPO requires a preference dataset, which is generated by sampling response pairs $(y_1, y_2)$ from the model. To directly fine-tune the source model, preference pairs are sampled from the source model itself. In contrast, HUMPA samples pairs from the SLM, as the SLM is the model that needs to be fine-tuned (refer to Section 3.3 for detail). Therefore, we compare the running time of HUMPA with prior work that samples pairs from the source model and apply DPO fine-tuning on the source model. To compare the fine-tuning time, we set the DPO batch size to 8 and epoch to 5. We utilized LoRA for efficient fine-tuning. We also compare the inference time of text generation from HUMPA attacked LLM and that from the unattacked LLM. The inference time is of a single pass with a batch size of 1. The results are as follows:
>
> |                     | Llama2-13B           | Llama3-70B           | Mixtral-8x7B            |
> | ------------------- | -------------------- | -------------------- | ----------------------- |
> | Sampling (Hours)    | 18.61                | 41.64                | 63.52                   |
> | Fine-tuning (Hours) | 3.09                 | 9.54                 | 13.58                   |
> | Inference (Seconds) | 12.88                | 27.87                | 32.51                   |
> |                     | **HUMPA(Llama2-7B)** | **HUMPA(Llama3-8B)** | **HUMPA(Mistral-7B)** |
> | Sampling (Hours)    | 13.87                | 10.97                | 10.83                   |
> | Fine-tuning (Hours) | 2.04                 | 3.20                 | 1.39                    |
> | Inference (Seconds) | 39.87                | 47.22                | 36.75                   |
>
> We find that HUMPA is much more efficient than directly DPO fine-tuning attack the source model. The results indicate the efficiency of HUMPA. The inference time of HUMPA is slightly slower than that of the source model, but the sacrifice is not significant. We have included a section of "Efficiency" in the updated version.

---

> ### Author Response · Authors · 2024-11-24
> **Official response to reviewer BPF1 (part 5/5)**
>
> **10. (S13) Discuss the trade-offs between efficiency gains and potential performance differences**
>
> **Response:**  Thanks for the constructive suggestion. We fine-tune Llama2-7B on the OpenWebText dataset with a batch size of 8 and 5 epochs, using RoBERTa-large as the scoring detector. We varying the size of training data and record the fine-tuning runtime. Evaluation is conducted using the RoBERTa-base detector on text generated by the attacked Llama2-13B model with $\alpha=1.5$. The results are presented below
>
> |                          | 1k     | 5k     | 8k     | 10k    |
> | ------------------------ | ------ | ------ | ------ | ------ |
> | Fine-tuning Time (Hours) | 0.2702 | 1.3523 | 1.8955 | 2.0416 |
> | AUROC                    | 0.8763 | 0.7140 | 0.6845 | 0.6202 |
>
> We find that performance gains increase with the size of the training dataset, while efficiency gains decrease as the training dataset size grows. It suggests that there is a trade-off between efficiency and performance: a larger training size improves performance but at the cost of reduced efficiency gains. Therefore, practitioners need to balance these factors according to their specific priorities. If an adversary prioritizes performance gains, a larger training dataset may be preferable. On the contrary, if efficiency or fine-tuning time is more critical, a smaller training size provides a better balance. We have included a section of "Sensitivity of Training Sizes" in the updated paper.
>
> **11.(S5) Suggest or conduct additional experiments that could test the boundaries of the attack's effectiveness across a wider range of scenarios or detectors**
>
> **Response:**  We thank the reviewer for the fair question. A practical approach to exploring the boundaries of the attack's effectiveness across a wider range of detectors would involve collecting text generated by the attacked LLMs, fine-tuned against several strong detectors, as negative samples. Human-written texts could then be collected as positive samples. Using this data, we can train a binary classifier to distinguish between machine-generated and human-written texts. However, this method may not fully capture the attack's effectiveness. The attacked model generates "polluted" texts with varying levels of $\alpha$, which impacts the quality of the generated text. As a result, this approach may be ineffective in reaching the real boundary of the attack, given that the intensity of the attack $\alpha$ may vary during the text generation process by the LLM.
>
>
>
> **12. (Q8) Can we apply such attacks in truly black-box settings, e.g., GPTZero or CopyLeaks?**
>
> **Response:**  Thank you for the constructive question. We fine-tuned a Llama2-7B model on the WritingPrompts dataset using GPTZero as the scoring detector, with DPO fine-tuning conducted at $\beta=0.1$. We then evaluated GPTZero on texts generated by the attacked Llama2-13B model. The results are as follows
>
> |       | Llama2-13B | HUMPA($\alpha=2.0$) | HUMPA($\alpha=5.0$) |
> | ----- | ---------------------- | ------------------- | ------------------- |
> | AUROC | 0.9951                 | 0.7987              | 0.5375              |
>
> We find that GPTZero can also be bypassed.  We agree that conducting more evaluations on commercial detectors, such as GPTZero, would be more comprehensive. However, we did not perform systematic experiments for that due to funding constraints.
>
> **13. (S11) Discuss the relative strengths and weaknesses of their approach compared to existing methods.**
>
> **Response:**  Thank you for the constructive suggestion. The primary advantage of HUMPA lies in its efficiency in fine-tuning a smaller model instead of a larger one and its ability to transfer modifications to the larger model to evade detection. This approach preserves the utility of the generated text. In contrast, existing methods rely on directly fine-tuning the LLM using techniques like DPO [R1]. However, this approach is not only expensive for large models (e.g., 70B) but also yield ineffective or low-quality results when applied solely to smaller models for evasion attacks.
>
> A potential weakness of HUMPA, as pointed out by the reviewer in Q1, S9 and addressed in our updated paper, is its reliance on a strong scoring detector to obtain an effective attack model. In the updated paper, we discuss it in our'' Analysis of Scoring Detector'' section that this gap may be unbridgeable due to the lack of high-quality preference data from weak scoring detectors.
>
>
>
> [R1]. Nicks, C., Mitchell, E., Rafailov, R., Sharma, A., Manning, C. D., Finn, C., & Ermon, S. (2023). Language model detectors are easily optimized against. In ICLR 2024.

---

> > ### Comment · Reviewer_BPF1 · 2024-11-24
> >
> > Thank you to the authors for providing a detailed rebuttal. I appreciate the clarity addressed and the acknowledgment of the limitations. Additionally, points #8, #9, and #12 are promising, with #12 being particularly interesting.
> >
> > I would raise my score if I see these with good results:
> > 1. A discussion about #12 is added to the manuscript, including an evaluation of utility, e.g., when alpha=2.0.
> > 2. Utility is also compared in Table 8 when comparing different attacks (beyond Table 9).

---

> ### Author Response · Authors · 2024-11-25
> **Follow-up by authors**
>
> Dear reviewer BPF1,
>
> Thanks for following up and providing constructive suggestions.
>
> In the revised manuscript, we have included the utility of all methods in Table 8 (refer to the Table in Response 8) and discussed the trade-off between evasion performance and utility in Appendix G. Additionally, we have included the GPTZero results in Table 9 and have discussed it in Appendix G. We would like to follow up to see if we address your concerns or if you have further questions. We sincerely appreciate your insightful comments and constructive suggestions during the rebuttal.

---

> ### Comment · Reviewer_BPF1 · 2024-11-25
>
> Thanks to the authors for the follow-up. Although what I meant (for my first point about GPTzero) is to discuss the attack's practical implications considering commercial APIs/tools, the current results look good.

---

### Official Review · Reviewer_pAX3 · 2024-11-04

**Soundness:** 3
**Presentation:** 3
**Contribution:** 2
**Rating:** 5
**Confidence:** 4

**Summary:**

Machine-generated texts can bring misinformation and misconduct. Therefore, several methods are developed to detect them. This paper conducts research on the robustness of these methods and propose an evasion method called HUMPA to generate texts that reads fluent and natural but can evade SOTA AI-generated text detectors. The key idea is to use a human-written dataset and use direct preference optimization and solve it via direct preference optimization. However, as DPO can only be effortlessly applied to small models, this paper resorts to proxy-tuning, which first tunes a relatively smaller model using DPO, and then use this model to manipulate the output distirbution of the larger model and achieve the same preferece tuning goal. The method is validated on a bunch of state-of-the-art detectors in both black-box and white-box scenarios, demonstrating its effectiveness in lowering detection accuracy while ensuring fluency and naturalness of the generated texts.

**Strengths:**

- The problem studied is highly relevant and very important.
- The paper is generally clearly-written and easy to follow.
- The method seems sound and the experiments are conducted on a wide range of methods.
- The idea to view AI-generated text detection as a preference optimization problem and solving it using DPO and proxy-tuning is intuitive and reasonable.

**Weaknesses:**

- The novelty of this paper on the technical side might be limited. The general idea of first tune a smaller model and then manipulate the sampling process of the larger model has been widely applied in previous works, such as Proxy-tuning [1], EFT [2], and DeRa [3] (not cited). The papers method is generally a combination of DeRa, EFT and Proxy-tuning. The main innovation of this paper is its application to the problem of evading machine-generated text detection. However, this cannot fully support the method's novelty.

- The improvement of the method over previous works seems marginal. This paper does not compare with previous methods on evading machine-generated text detection, the only comparison is on the fluency part rather than effectiveness. According to the experimental results, the effectiveness of the proposed method is unstable. For example, in table 1-white-box, Likelihood, DetectGPT and Fast-DetectGPT's performances are only slightly degraded. This raises doubts on how the proposed method really advanced the field.

- The provided theorems do not seem to be closely connected with the proposed method. For example, thoerem 1 just proves the existence of the optimal $\beta$, and this theorem seems applicable to a wide range of tasks that uses DPO. Yet, it is unclear how significant it is to the evasion of AI-generated text detection and how this can be achieved through proxy-tuning.


Minor:
- Figure 1 caption, "After the attack, the distribution aligns more closely with that of human-written text" has been repeated for twice.
- Citation formats in Section 4.1 are mostly wrong. Please correct them.

References:

[1]: Liu et al. "Tuning Language Models by Proxy." COLM 2024.

[2]: Mitchell et al. "An Emulator for Fine-tuning Large Language Models using Small Language Models." ICLR 2024.

[3]: Liu et al. "Decoding-time Realignment of Language Models." ICML 2024.

**Questions:**

- DPO requires paired datasets. How do you collect the human-written text dataset for DPO for unseen tasks or topics that do not have a pool of human-written texts?

---

> ### Author Response · Authors · 2024-11-24
> **Official response to reviewer pAX3 (part 1/4)**
>
> Dear reviewer pAX3,
>
> Thank you for your careful reading and suggestions. We answer the questions below:
>
> **1. The novelty of this paper on the technical side might be limited. The general idea of first tune a smaller model and then manipulate the sampling process of the larger model has been widely applied in previous works, such as Proxy-tuning [R1], EFT [R2], and DeRa [R3] (not cited). The papers method is generally a combination of DeRa, EFT and Proxy-tuning. The main innovation of this paper is its application to the problem of evading machine-generated text detection. However, this cannot fully support the method's novelty.**
>
> **Response:**  We highlight that our work introduces **a lightweight strategy to attack LLM by leveraging a humanized SLM that can bypass detectors**. This simple yet effective strategy involves 1) obtaining humanized SLM by DPO fine-tuning, and 2)  attacking the LLM's next-token output distribution by the subtle SLM's logit offset for each token probability in the decoding phase. We find that many detectors are vulnerable, and HUMPA bypasses the detectors while preserving the generation utility.
>
> In the updated version, we have included additional related works in Appendix B. While Proxy-tuning [R1], EFT [R2], and DeRa [R3] (we cited it in the revised version) share similar vision in proxy fine-tuning, we need to claim that our method aims to fine-tune an SLM towards optimal reward until it reaches the same level of reward for the human process according to a scoring detector, and adapt the LLM to achieves the same expected reward. We show **in terms of detection evasion,** **our attack strategy at decoding time achieves the same effect as directly attack the source model by DPO fine-tuning**.  We believe it is a theoretical contribution in the field of proxy fine-tuning an LLM, and provide insights into effortlessly bypassing the LLM detectors.
>
> We have included additional related works in Appendix B of the updated version. However, while we agree that the primary aim of our work is not to introduce a novel method for proxy fine-tuning, we notice that the ICLR 2025 call for papers explicitly lists "alignment, fairness, safety, privacy, and societal considerations" as topics of interest. We believe our work aligns closely with this category, addressing important challenges in the safety and robustness of AI systems.
>
>
>
> [R1] Liu, A., Han, X., Wang, Y., Tsvetkov, Y., Choi, Y., & Smith, N. A. (2024). Tuning language models by proxy. *arXiv preprint arXiv:2401.08565*.
>
> [R2] Mitchell, E., Rafailov, R., Sharma, A., Finn, C., & Manning, C. D. (2023). An emulator for fine-tuning large language models using small language models. *arXiv preprint arXiv:2310.12962*.
>
> [R3] Liu, T., Guo, S., Bianco, L., Calandriello, D., Berthet, Q., Llinares, F., ... & Blondel, M. (2024). Decoding-time Realignment of Language Models. *arXiv preprint arXiv:2402.02992*.
>
>
> **2. The improvement of the method over previous works seems marginal. According to the experimental results, the effectiveness of the proposed method is unstable. For example, in table 1-white-box, Likelihood, DetectGPT and Fast-DetectGPT's performances are only slightly degraded. This raises doubts on how the proposed method really advanced the field.**
>
> **Response:**  Thanks for the great question. It is important to note that the performances in Table 1,2,3,5,6,7 (updated to Tables 1, 2, 4, 5, 6, 7, 10, and 11 in the revised paper) do **not** represent the best detection evasion performance achievable by HUMPA. Instead, these tables show the AUROC performance of detectors when the utilities of text generated from HUMPA attacked LLM and the source LLM are constrained within a **manually set** budget of $\Delta S_{Bert}\leq 0.02$ and $\Delta \text{ROUGE-1} \leq 0.03$. We aim to show **within this budget**, how HUMPA performs in detection evasion, and how fragile for different detectors when attacked by HUMPA.
>
> As we state in Section 3.3, the ratio $\alpha$ controls the intensity of the attack on the LLM, with larger values of $\alpha$ yielding higher rewards and better detection evasion, while smaller values keep the attacked LLM closer to the source LLM. Therefore, **the effectiveness of detection evasion can be controlled by $\alpha$**.

---

> ### Author Response · Authors · 2024-11-24
> **Official response to reviewer pAX3 (part 2/4)**
>
> For instance, when evaluating the white-box detectors Likelihood, DetectGPT, and Fast-DetectGPT on texts generated by HUMPA attacked Llama2-13B for the WritingPrompts dataset, the AUROC of the detectors decrease as $\alpha$ increases
>
> |                | Llama2-13B | HUMPA($\alpha=1.0$) | HUMPA($\alpha=1.2$) | HUMPA($\alpha=1.5$) | HUMPA($\alpha=2.0$) |
> | -------------- | ---------- | ------------------- | ------------------- | ------------------- | ------------------- |
> | Likilihood     | 0.9999     | 0.5933              | 0.2846              | 0.0206              | 0.0000              |
> | DetectGPT      | 0.8795     | 0.7096              | 0.4422              | 0.2545              | 0.2153              |
> | Fast-DetectGPT | 0.9999     | 0.6727              | 0.5184              | 0.4901              | 0.3939              |
>
> We find that, at the same attack ratio, although Fast-DetectGPT shows better detection performance than Likelihood and DetectGPT. DetectGPT experiences greater relative drops, showing it more robust to attacks compared to Fast-DetectGPT and Likelihood. We demonstrated this effect in the "Analysis of $\alpha$" section (Section 4.4) and discussed it following Theorem 3.2, noting that "*a larger $\alpha$ leads to higher rewards and improved detection evasion, while a smaller $\alpha$ keeps $M'$  closer to the reference model $M^{ref}$*". We study this effect in the "Analysis of $\alpha$" section (Section 4.4). To clarify the confusion, we have included more details and analysis in the section of "More Analysis of $\alpha$" in Appendix G of the updated paper.
>
> Since each detector is built with a distinct architecture and mechanism, their vulnerabilities to attacks differ as well. Enhancing the robustness of detectors has consistently been a key focus in this field [R1]. Our work also reveals that different detectors exhibit varying levels of vulnerability. Besides, we highlight it in the Ethics Statement: "*This study aims to encourage the broader research community to focus on developing more robust detection methods*."
>
> [R1] Zhang, Y. F., Zhang, Z., Wang, L., Tan, T., & Jin, R. (2023). Assaying on the robustness of zero-shot machine-generated text detectors. arXiv preprint arXiv:2312.12918.

---

> ### Author Response · Authors · 2024-11-24
> **Official response to reviewer pAX3 (part 3/4)**
>
> **3. This paper does not compare with previous methods on evading machine-generated text detection, the only comparison is on the fluency part rather than effectiveness.**
>
> **Response:**  Thanks for the constructive suggestion, it helps to improve our paper. In Appendix G: "More Analysis of $\alpha$", we evaluated our approach against the paraphrasing attack method DIPPER [R1] by comparing both the AUROC and the Fluency Win Rate. To show the effectiveness of detection evasion, we compare our method with two state-of-the-art paraphrasing evasion techniques: DIPPER Paraphrasing [R1] and Query-based Substitutions [R2]. Their performances on the WritingPrompts dataset are presented as follows.
>
>
>
> |                           | Likelihood | LogRank    | LRR        | NPR        | DNA-GPT    | DetectGPT  | Fast-DetectGPT | Binoculars | $S_{Bert}$ | ROUGE-1/2/L                                |
> | ------------------------- | ---------- | ---------- | ---------- | ---------- | ---------- | ---------- | -------------- | ---------- | ---------- | ------------------------------------------ |
> | Llama2-13B                | 0.9923     | 0.9945     | 0.9845     | 0.9670     | 0.7534     | 0.8713     | 0.9949         | 0.9969     | 0.8189     | 0.2587$\mathbf{/}$0.0480$\mathbf{/}$0.1497 |
> | Dipper Paraphrasing       | 0.8125     | 0.7998     | 0.7220     | 0.5193     | 0.6240     | 0.2675     | 0.9754         | 0.9398     | 0.8006     | 0.2076$\mathbf{/}$0.0226$\mathbf{/}$0.1191 |
> | Query-based Substitutions | 0.9843     | 0.9921     | 0.9828     | 0.3030     | 0.7072     | 0.1914     | 0.9972         | 1.0000     | 0.7989     | 0.2015$\mathbf{/}$0.0383$\mathbf{/}$0.1256 |
> | HUMPA ($\alpha=1.2$)      | 0.1647     | 0.1625     | 0.1599     | 0.2592     | 0.0723     | 0.2124     | 0.0794         | 0.1743     | 0.8053     | 0.2281$\mathbf{/}$0.0422$\mathbf{/}$0.1409 |
> | HUMPA ($\alpha=1.5$)      | **0.0109** | **0.0109** | **0.0117** | **0.0617** | **0.0034** | **0.0582** | **0.0007**     | **0.0021** | 0.8014     | 0.2137$\mathbf{/}$0.0404$\mathbf{/}$0.1383 |
>
> We find that our method with $\alpha = 1.5$, achieves the best detection evasion performance, aligning with our analysis of $\alpha$ in Section 3.3.  We have included the results in Table 8 of the updated paper.
>
> The state-of-the-art fine-tuning approach is directly DPO fine-tuning the source model (Nicks et al., 2024) to evade detection. However, this approach is time-consuming when DPO fine-tuning a large model (e.g., 70B), and it is difficult to balance maintaining high-quality text generation with effective detection evasion. We claim our strategy is more efficient than that, and the utility of text generation is preserved. Therefore,  in the "Utility Preserving" section (Section 4.4), we compare our method with theirs in terms of utility preservation. We have included an "Efficiency" section in the updated version, to show the efficiency of our approach compared to this method.
>
>
> [R1] Krishna, K., Song, Y., Karpinska, M., Wieting, J., & Iyyer, M. (2024). Paraphrasing evades detectors of ai-generated text, but retrieval is an effective defense. *Advances in Neural Information Processing Systems*, *36*.
>
> [R2] Shi, Z., Wang, Y., Yin, F., Chen, X., Chang, K. W., & Hsieh, C. J. (2024). Red teaming language model detectors with language models. *Transactions of the Association for Computational Linguistics*, *12*, 174-189.
>
>
> **4. The provided theorems do not seem to be closely connected with the proposed method. For example, thoerem 1 just proves the existence of the optimal $\beta$, and this theorem seems applicable to a wide range of tasks that uses DPO. Yet, it is unclear how significant it is to the evasion of AI-generated text detection and how this can be achieved through proxy-tuning.**
>
> **Response:**  We thank the reviewer for pointing out the possible confusion in our theoretical presentation, and address the concern below:
>
> Theorem 3.1 is indeed a general result for models fine-tuned with DPO, though it is phrased under the specific scenario where we train the model on labels generated according to the Bradley-Terry model and the human-ness score. We have changed the theorem into a lemma to highlight its preliminary nature.
>
> On the other hand, Theorem 3.2 is the main theoretical result pertaining to our method, as it states that our attack model HUMPA defined in Equation 4 is equivalent to a large attack model fine-tuned on the DPO objective.  This does not directly state anything about the performance of the fine-tuned model, but combining the two theorems gives us the desired theoretical conclusion: our proxy-attacked model can achieve an expected reward on par with human-generated texts. For clarification, we reorganized the post-theorem discussion into a paragraph on the effect of $\alpha$ and a separate Corollary 3.3 to address the combined result of both our theorems.

---

> > ### Author Response · Authors · 2024-11-24
> > **Official response to reviewer pAX3 (part 4/4)**
> >
> > In terms of the implication of our theoretical results on detection evasion, we prove our proxy attacker is able to achieve a similar reward compared to human-generated texts. Since we assume the detector implicitly uses this reward for machine-generated text detection through the Bradley-Terry model, this effectively means that the detector will not be able to reliably differentiate between human-generated and attacker-generated texts.
> >
> >
> >
> > **5. Minor typos**
> >
> > **Response:**  Thank you for the careful reading and suggestion. We have corrected the typos and citation formats in the updated paper.
> >
> >
> >
> > **6. DPO requires paired datasets. How do you collect the human-written text dataset for DPO for unseen tasks or topics that do not have a pool of human-written texts?**
> >
> > **Response:**  Thanks for the great question. For each prompt $x$, we sample response pairs $(y_1, y_2)$ using the SLM. To create the preference dataset $\mathcal{D} = \{ (x, y^{w}, y^{l}) \}$, preference labels are determined by comparing a scoring detector's human-ness score $s(x, y)$ on the responses: assign preference label $y_1 \succ y_2$ and let $y^{w} = y_1$, $y^{l} = y_2$; otherwise assign $y^{w} = y_{2}$, $y^l = y_{1}$. As a result, our method does not require human-written texts for DPO fine-tuning. For more details, please refer to Section 3.3.

---

> ### Comment · Reviewer_pAX3 · 2024-11-25
> **Thank you for the rebuttal.**
>
> Dear authors, thanks for your rebuttal. I have raised my score to 5. While some of my concerns are addressed by your response, I still have the following concerns.
>
> - Regarding novelty, I still feel this paper offers limited new insights to the community. It is already well-established that DPO can be used to effective attack LLM-generated text detectors (Nicks et al., 2024). The paper's primary contribution lies in replacing the full attack model's DPO with another efficient fine-tuning method called proxy-tuning. However, it is not surprising that this straightforward combination yields performance gains, as proxy-tuning can be applied to any task utilizing full parameter DPO. My main concern is that this work may not significantly advance understanding or provide valuable insights for the AI-generated text attack/defense community, potentially undermining its claim to novelty.
>
> - Regarding the new experiments, I am concerned that whether the larger choice of $\alpha$ can degrade text utility. Current experiments in the rebuttal improves attack effectiveness by enlarging $\alpha$, but evidently it will harm text utility. Therefore, please consider adding the metric of text utility for reference (especially in my point #3. Please add the text utility metric for your method with different choices of $\alpha$ as well as all baselines.).
>
> - Could you elaborate on how your scoring detector generalizes to texts with OOD themes or styles that are not covered in the training set?

---

> ### Author Response · Authors · 2024-11-25
> **Follow-up Questions**
>
> **1. Regarding novelty**
>
> **Response:**  We thank the reviewer for the comments. The state-of-the-art work in (Nicks et al., 2024) employs the strategy of fine-tuning the source model directly for detection evasion. While our proxy fine-tuning methodology is straightforward, we aim to improve the efficiency of directly fine-tuning to attack the large models. Furthermore, our theory demonstrates that our attacking strategy at decoding time achieves the same evasion effectiveness as directly attacking the source model through DPO fine-tuning, which theoretically justifies the use of the proxy approach. We believe the key focus in evasion detection should be to inspire the community to develop robust models against threats. Addressing this robustness challenge is crucial not only for ensuring the reliability of detection systems, but also for building trust in the technology among stakeholders. From this standpoint, our work is both valuable and relevant, as it addresses the foundational challenges and contributes to advancing the field.
>
> **2. Whether the larger choice of  $\alpha$  can degrade text utility. Adding the metric of text utility for reference.**
>
> **Response:**  Thanks for the pointing out this issue. In our experiments, we indeed found that with the increase of $\alpha$, the utility decreases (refer to the Table in point #3). This is within expectations and in line with our discussion after Theorem 3.2 (Section 3.3), where larger $\alpha$ leads to higher reward and better detection evasion, while smaller $\alpha$ keeps $M'$ closer to the reference model $M^{ref}$. Hence, $\alpha$ governs the trade-off between evasion performance and generation quality, and an adversary should choose an appropriate $\alpha$ to balance the attack effect and text quality, e.g. find a maximum $\alpha$ that keeps the generated text within a given quality budget (maximum decrease in quality).
> In the revised manuscript, we have included the utility scores of all methods in Table 8 and further discussed the trade-off between evasion performance and utility in Appendix G.
>
> **3. Elaborate on how scoring detector generalizes to texts with OOD themes or styles that are not covered in the training set.**
>
> **Response:**  Thank you for this thoughtful question. To elaborate, since HUMPA generates its attacks based on the LLM (when $\alpha=0$ the attacker is exact the same as pre-trained LLM), the generation utility is mostly dependent on $\alpha$ and largely independent of the dataset used for DPO training: with a reasonable $\alpha$, the attacked generation should have high quality regardless of training data.
>
> Second, in terms of the effectiveness of detection evasion, consider the distribution shift term in Equation (4):
> $$
> \bigg( \frac{\pi _{M _{s}}(y _t|x,y _{<t})}{\pi^{ref} _{M _{s}}(y _t|x,y _{<t})}\bigg)^{\alpha}
> $$
> which captures the SLM's reference-to-attacker shift. It is reasonable to hypothesize that this shift involves correcting some text generation "habits'' of the machine to be more human-like (e.g. excluding fancy, impractical words that are not often used in daily life). Furthermore, it is not far-fetched to assume that these habitual corrections are mostly context-free, i.e. they do not generally depend on the specific textual theme or style. Therefore, with a reasonable training preference dataset, the SLM should be able to capture these general habitual corrections, and through HUMPA apply these corrections on LLM-generated texts to evade detection, even in environment with different distributions.

---

> ### Comment · Reviewer_pAX3 · 2024-11-29
> **Thanks for your rebuttal.**
>
> Dear authors, thank you for your rebuttal. I have carefully read your rebuttal and new revised manuscript. While the new follow-up response solves some of my concerns, I remain concerned on the novelty of this paper. I'm still feeling this paper to be a straightfoward combination of proxy-tuning and DPO for a low-fruit downstream task (i.e., AI-generated text detection), missing much fresh insights/understandings into this field (this point is similar to Reviewer wsFK), making it not enough for ICLR. Therefore, I decide to keep my rating and I sincerely thank the authors again for answering my questions.

---

### Official Review · Reviewer_oRJ8 · 2024-11-04

**Soundness:** 3
**Presentation:** 2
**Contribution:** 2
**Rating:** 6
**Confidence:** 3

**Summary:**

The article explores a strategy known as the Human-like Proxy Attack (HUMPA), aimed at evading detection by large language model (LLM) detectors. This approach uses a small language model (SLM) fine-tuned through reinforcement learning to alter the output of the LLM, making the machine-generated text indistinguishable from human-written content. Evaluations using various open-source models and datasets indicate a significant decrease in detector accuracy, while preserving the quality of the text.

**Strengths:**

- Utilizing fine-tuned, human-like small language models to modify the distribution of source models, making them resemble human-written text, is a well-conceived attack strategy. This serves as an effective stress test to drive the advancement of current detectors.
- The proposed method efficiently lowers the cost of fine-tuned model attacks. Additionally, despite the attack, the text quality remains high, which is essential for practical applications.

**Weaknesses:**

- The experimental setup lacks latest strong baselines, such as the more robust detector "binoculars [1]". (This detector is easy to replicate and very fast.)
- The authors are commended for their results in cross-domain scenarios. However, cross-model detection evasion deserves more focus, as disciplines and languages remain constant, while LLMs continue to evolve.
- The method proposed in the paper uses reinforcement learning to align the probability distribution of small models with human patterns, achieving human-like output. It's important to note that this approach appears similar to DALD [2], with the main difference being DALD's use of further training, while HUMPA employs reinforcement learning. I hope the authors can further clarify their contributions to distinguish them from HUMPA.
- Formatting Issues
    - The citation format for Yang et al. (2023b) is incorrect on lines 365, 367, 369, and 370. The author should thoroughly review and correct the citation format throughout the paper.

Overall, I find this work to be promising. If the author can convincingly address the weaknesses identified, I would be open to revising my assessment.

[1] Hans, A., Schwarzschild, A., Cherepanova, V., Kazemi, H., Saha, A., Goldblum, M., ... & Goldstein, T. Spotting LLMs With Binoculars: Zero-Shot Detection of Machine-Generated Text. In *Forty-first International Conference on Machine Learning*.

[2] Zeng, C., Tang, S., Yang, X., Chen, Y., Sun, Y., Li, Y., ... & Xu, D. (2024). Improving Logits-based Detector without Logits from Black-box LLMs. *arXiv preprint arXiv:2406.05232*.

**Questions:**

- Has the author considered potential safeguards to prevent the misuse of the proposed attack strategies?

---

> ### Author Response · Authors · 2024-11-24
> **Official response to reviewer oRJ8 (part 1/2)**
>
> Dear reviewer oRJ8,
>
> We appreciate your insightful comments and provide our response as follows:
>
> **1. The experimental setup lacks latest strong baselines, such as the more robust detector "binoculars [R1]". (This detector is easy to replicate and very fast.)**
>
> **Response:** We thank the reviewer for the constructive suggestion. We have included an additional experiment to evaluate Binoculars[R1] as a detection method for all the models and datasets (Table 1, 2, 4, 5, 6, 7, 8, 10, 11 in the updated paper), and test its results when the generation utilities of texts produced by HUMPA and the source model are within the budget of $\Delta S_{Bert} \leq 0.02$ and $\Delta \text{Rouge-1} \leq 0.03$.  The results suggest that Binoculars experiences a greater performance drop when evaluated on texts generated by the attacked Mixtral-8x7B, with the highest relative decrease reaching 95.03% on OpenWebText.
>
> | Datasets            | Llama2-13B | HUMPA(Llama2-7B) | Llama3-70B | HUMPA(Llama3-8B) | Mixtral-8x7B | HUMPA(Mistral-7B) |
> | ------------------- | ---------- | ---------------- | ---------- | ---------------- | ------------ | ----------------- |
> | OpenWebText         | 1.0000     | 0.7903           | 0.9990     | 0.8830           | 0.8840       | 0.0439            |
> | Writing             | 1.0000     | 0.6773           | 0.9990     | 0.8771           | 0.9539       | 0.2019            |
> | PubMed              | 1.0000     | 0.9012           | 0.9987     | 0.8186           | 0.9679       | 0.6702            |
> | PHX->HSS | 0.9971     | 0.7869           | 0.9990     | 0.7094           | 0.8252       | 0.1852            |
> | CS->PHX  | 0.9904     | 0.7093           | 0.9812     | 0.4127           | 0.5183       | 0.0489            |
> | EN-> GER | 0.9929     | 0.3108           | 0.9901     | 0.3265           | 0.7293       | 0.1564            |
>
>
>
> [R1] Hans, A., Schwarzschild, A., Cherepanova, V., Kazemi, H., Saha, A., Goldblum, M., ... & Goldstein, T. Spotting LLMs With Binoculars: Zero-Shot Detection of Machine-Generated Text. In *Forty-first International Conference on Machine Learning*.
>
>
>
> **2. Cross-model detection evasion deserves more focus, as disciplines and languages remain constant, while LLMs continue to evolve.**
>
> **Response:** Thanks for the insightful suggestion. While we agree that the LLMs continue to evolve, in our experiments we applied the HUMPA strategy to attack models released at different times. Specifically, we attacked Llama2-13B (released in July 2023) using Llama2-7B, Mixtral-8x7B (released in December 2023) using Mistral-7B, and Llama3-70B (released in April 2024) using Llama3-8B.
>
> To extend the HUMPA attack to a source LLM using an SLM from a different model family, we still need to obtain a humanized SLM, begin by sampling response pairs $(y_s, y_s^{'})$ from the SLM given prompt $x$, and assigning preference labels using a scoring detector to create a preference dataset. Then, we fine-tune the SLM with DPO to obtain a humanized SLM (as detailed in Section 3.3 of our paper).
>
> To address scenarios where the LLM and SLM come from different model families with dissimilar vocabularies, we adopt the technique described in [R2]. In this approach, we keep the humanized SLM frozen as the reward model while fine-tuning the source LLM. During the fine-tuning process of the source LLM, we sample response pairs $(y_l, y_l^{'})$  from the LLM, which are used as inputs for both the target LLM and the reward SLM. We perform DPO fine-tuning on Phi-3.5-mini-instruct using the scoring detector Roberta-large. We then apply this reward model to guide the fine-tuning [R2] of the source model Llama2-13B for 3 epochs.  The performance of the unattacked source Llama2-13B and the attacked Llama2-13B by humanized Phi-3.5-mini, tested using Roberta-base on the WritingPrompts dataset, is as follows:
>
> |       | Llama2-13B | HUMPA(Phi-3.5-mini) |
> | ----- | ---------- | ------------------- |
> | AUROC | 0.9673     | 0.8347              |
>
> We find that the source model can be attacked by a humanized SLM from a different family.  However, since this technique requires fine-tuning both the source LLM and the SLM, we did not include it in our paper. We leave systematic explorations on attacking LLMs from different model families without fine-tuning for future work to maintain our focus on the fragility of zero-shot detectors, the ease of attacking an LLM by fine-tuning only a small model, the equivalence between attacking the source LLM and attacking by an humanized SLM, and the preservation of the attacked LLM's generation utility.
>
>
> [R2] Gao, Z., Chang, J. D., Zhan, W., Oertell, O., Swamy, G., Brantley, K., ... & Sun, W. (2024). Rebel: Reinforcement learning via regressing relative rewards. *arXiv preprint arXiv:2404.16767*.

---

> > ### Author Response · Authors · 2024-11-24
> > **Official response to reviewer oRJ8 (part 2/2)**
> >
> > **3. It's important to note that this approach appears similar to DALD [R3], with the main difference being DALD's use of further training, while HUMPA employs reinforcement learning. I hope the authors can further clarify their contributions to distinguish them from HUMPA.**
> >
> > **Response:** We thank the reviewer for pointing out this issue. We list the main differences and similarities between our work and DALD [R3] in the following aspects, in the hopes of clarifying our contributions to this literature:
> >
> > - Research Objective: While our work focuses on detection evasion, DALD is a method for improving the machine-generated text detection, which is effectively the opposite of our research objective. Although both HUMPA and DALD are based on next-token logits, HUMPA utilizes these logits to generate human-like texts, while DALD employs them for detecting machine-generated text.
> > - Method: A surface-level similarity between HUMPA and DALD is the use of a surrogate / small model. However, the reasonings and methodologies are fundamentally different. We use a humanized small model to induce logit offsets in the LLM during inference, efficiently generating polluted text for detection evasion. In contrast, DALD employs a surrogate model out of necessity due to the lack of access to the logits of the black-box target model.
> > - Fine-tuning:  We fine-tune a small language model using DPO to maximize the implicit reward from a scoring detector, while DALD employs instruction fine-tuning on a surrogate model to align it with a target model.
> > - Theoretical guarantee: We show that our attack strategy at decoding time achieves the same effect as directly attack the source model by DPO fine-tuning in the ideal case  (Theorem 3.2); DALD shows the surrogate model imitates the target model well enough to still be effective in their Theorem 1.
> > - Findings: We find that many detectors are vulnerable, and HUMPA effectively evades detection while preserving the utility of the generated text. DALD shows the improving of detection performance.
> >
> >
> >
> > [R3] Zeng, C., Tang, S., Yang, X., Chen, Y., Sun, Y., Li, Y., ... & Xu, D. (2024). Improving Logits-based Detector without Logits from Black-box LLMs. *arXiv preprint arXiv:2406.05232*.
> >
> > **4. Formatting Issues**
> >
> > **Response:**  We appreciate the reviewer’s thorough attention to our formatting typos. We have corrected them in the revised paper.
> >
> > **5. Potential safeguards to prevent the misuse of the proposed attack strategies**
> >
> > **Response:**  We thank the reviewer for this insightful question. We certainly agree that exploring ways to counter this type of attack and evaluating the feasibility of such defenses is a vital next step in this research field.  We leave experimental work on this front to future works to preserve our paper’s focus on detector fragility and the attack itself.
> >
> > Regarding potential safeguards, we identify the following countermeasures:
> >
> > **Prevent access to model logits**: In environments where detection is most widely utilized, models with closed-source weights are typically employed. A malicious actor seeking to execute our attack would be constrained to models with accessible logits (both SLM and LLM), potentially limiting the quality of undetectable text they could produce. As the ability to publicly fine-tune models like ChatGPT is readily available, implementing this defense strategy would within the control of the model provider.
> >
> > **Train a classifier**: The attacked LLM produces text with a distribution closely resembling that of human-written text. A practical defense involves collecting generated text from the attacked LLM as negative samples. Human-written texts could then be collected as positive samples. Using this data, a binary classifier can be trained to distinguish between machine-generated and human-written texts, thereby defending against evasion. However, this method may not fully capture the attack's effectiveness. The attacked model generates "polluted" texts with varying levels of $\alpha$, which impacts the quality of the generated text. As a result, this defending approach may be ineffective due to the intensity of the attack ratio $\alpha$ may vary in the text generation process.

---

> ### Comment · Reviewer_oRJ8 · 2024-11-25
> **Official Comment by Reviewer oRJ8**
>
> Thank you for addressing all my questions and providing additional details regarding your experiments, especially the new results on Binoculars. As the rebuttal has addressed some of my concerns, I will raise my rating to 6 accordingly.

---

### Author Response · Authors · 2024-11-24
**Rebuttal by Authors**

### General Response

Dear AC and Reviewers,

We sincerely appreciate the reviewers for their positive feedback and highly constructive comments. For clarity and readability of the paper, the following updates have been made:

- We have added additional experiments to evaluate Binoculars for all models and datasets.
- We have added additional related work.
- We have enhanced the descriptions of the settings, training details, and metrics.
- We have added runtime experiments.
- We have added more baselines.
- We have included experiments on training size.
- We have included experiments on analyzing the scoring detector.
- We have included experiments on analyzing the small model selection.
- We have enhanced the theoretical analysis presentation, changed Theorem 3.1 to Lemma 3.1, and added Corollary 3.3.
- Minor change of typos, citation formats, and re-description for clarity and conciseness.



Thanks again,

The Authors

---

### Meta-Review · Area_Chair_RYK6 · 2024-12-20

**Metareview:**

After carefully reading all comments, I think this paper has a fruitful discussion during the rebuttal stage. By explaining more about the paper's novelty issue and adding more experimental results, the authors addressed most of the questions and several reviewers raised their score. Finally, there are three positive score (8, 6, 6) and two negative score (5, 5). By carefully checking the comments of two reviewers still holding negative scores, I found no major issues that can be sufficient reasons to reject this paper. So the decision is accept.

**Additional Comments On Reviewer Discussion:**

I highlight the two major issues that attracted my attention in discussion.

1. **Novelty issue** questioned by Reviewer wsFK, Reviewer pAX3, and Reviewer SF6Q. There are three reviewers pointed out that the method is straightforward and lacks novelty. The authors explained their novelty by emphasizing attacking the attacker’s goal, novel findings, importance of the topic, etc. After rebuttal, only Reviewer pAX3 still feels the method is simple and can be seen as a combination of several well-studied techniques. However, Reviewer wsFK and SF6Q both agreed the authors have answered all questions and Reviewer wsFK raised the score. It can be seen that the novelty issue has been largely addressed via more explanation and is no long a main point leading to a rejection.

2. **More baselines** questioned by Reviewer wsFK, Reviewer oRJ8, Reviewer BPF1 and Reviewer pAX3. They provided more related works to the authors to improve the experimentation. During discussion, the authors added experimentation and listed the new results in their responses. After discussion, all reviewers feel this issue is addressed and no further questions on this point were proposed.

---

### Decision · Program_Chairs · 2025-01-22

Accept (Poster)